# A Cubic Regularization Approach for Finding Local Minimax Points in Nonconvex Minimax Optimization

**Ziyi Chen**  *u1276972@utah.edu*
*Department of Electrical and Computer Engineering*
*University of Utah*

**Zhengyang Hu**  *datou30@mail.ustc.edu.cn*
*School of Mathematical Sciences*
*University of Science and Technology of China*

**Qunwei Li**  *qunwei.qw@antgroup.com*
*Ant Group*

**Zhe Wang**  *wang.10982@osu.edu*
*JD company*

**Yi Zhou**  *yi.zhou@utah.edu*
*Department of Electrical and Computer Engineering*
*University of Utah*

**Reviewed on OpenReview:** *https://openreview.net/forum?id=jVMMdg31De*

## Abstract

Gradient descent-ascent (GDA) is a widely used algorithm for minimax optimization. However, GDA has been proved to converge to stationary points for nonconvex minimax optimization, which are suboptimal compared with local minimax points. In this work, we develop cubic regularization (CR) type algorithms that globally converge to local minimax points in nonconvex-strongly-concave minimax optimization. We first show that local minimax points are equivalent to second-order stationary points of a certain envelope function. Then, inspired by the classic cubic regularization algorithm, we propose an algorithm named Cubic-LocalMinimax for finding local minimax points, and provide a comprehensive convergence analysis by leveraging its intrinsic potential function. Specifically, we establish the global convergence of Cubic-LocalMinimax to a local minimax point at a sublinear convergence rate and characterize its iteration complexity. Also, we propose a GDA-based solver for solving the cubic subproblem involved in Cubic-LocalMinimax up to certain pre-defined accuracy, and analyze the overall gradient and Hessian-vector product computation complexities of such an inexact Cubic-LocalMinimax algorithm. Moreover, we propose a stochastic variant of Cubic-LocalMinimax for large-scale minimax optimization, and characterize its sample complexity under stochastic sub-sampling. Experimental results demonstrate faster or comparable convergence speed of our stochastic Cubic-LocalMinimax than the state-of-the-art algorithms such as GDA (Lin et al., 2020) and Minimax Cubic-Newton (Luo et al., 2022). In particular, our stochastic Cubic-LocalMinimax was also faster as compared to several other algorithms for minimax optimization on a particular adversarial loss for training a convolutional neural network on MNIST.

## 1 Introduction

Minimax optimization (a.k.a. two-player sequential zero-sum games) is a popular modeling framework that has broad applications in modern machine learning, including game theory (Ferreira et al., 2012), generative

adversarial networks (Goodfellow et al., 2014), adversarial training (Sinha et al., 2017), reinforcement learning (Qiu et al., 2020; Ho and Ermon, 2016; Song et al., 2018), etc. A standard minimax optimization problem is shown below, where $f$ is a smooth bivariate function.

$$\min_{x \in \mathbb{R}^m} \max_{y \in \mathbb{R}^n} \ f(x, y). \tag{P}$$

In the existing literature, many optimization algorithms have been developed to solve different types of minimax problems. Among them, a simple and popular algorithm is the gradient descent-ascent (GDA), which alternates between a gradient descent update on $x$ and a gradient ascent update on $y$ in each iteration. Specifically, the global convergence of GDA has been established for minimax problems under various types of global geometries, such as convex-concave-type geometry ($f$ is convex in $x$ and concave in $y$) (Nedić and Ozdaglar, 2009; Du and Hu, 2019; Mokhtari et al., 2020; Zhang and Wang, 2021), bi-linear geometry (Neumann, 1928; Robinson, 1951) and Polyak-Łojasiewicz geometry (Nouiehed et al., 2019; Yang et al., 2020), yet these geometries are not satisfied by general nonconvex minimax problems in modern machine learning applications. Recently, many studies proved the convergence of GDA in nonconvex minimax optimization for both nonconvex-concave problems (Lin et al., 2020; Nouiehed et al., 2019; Xu et al., 2023) and nonconvex-strongly-concave problems (Lin et al., 2020; Xu et al., 2023; Chen et al., 2021). In these studies, it has been shown that GDA converges sublinearly to a stationary point where the gradient of an envelope-type function $\Phi(x) := \max_y f(x, y)$ vanishes.

Although GDA can find stationary points in nonconvex minimax optimization, the stationary points may include candidate solutions that are far more sub-optimal than global minimax points, (e.g. saddle points of the envelope function $\Phi$) which are known to be the major challenge for training high-dimensional machine learning models (Dauphin et al., 2014; Jin et al., 2017; Zhou and Liang, 2018). However, finding global minimax points is in general NP-hard (Jin et al., 2020). Recently, Jin et al. (2020) proposed a notion of local minimax point that is computationally tractable and is close to global minimax point (see Definition 2.1 for the formal definition) .

In the existing literature, several studies have proposed Newton-type GDA algorithms for finding such local minimax points. Specifically, Wang et al. (2020) proposed a Follow-the-Ridge (FR) algorithm, which is a variant of GDA that applies a second-order correction term to the gradient ascent update. In particular, the authors showed that any strictly stable fixed point of FR is a local minimax point, and vice versa. In another work (Zhang et al., 2021), the authors proposed two Newton-type GDA algorithms that are proven to locally converge to a local minimax point at a linear and super-linear convergence rate, respectively. However, these second-order-type GDA algorithms only have asymptotic convergence guarantees that require initializing sufficiently close to a local minimax point, and they do not have any global convergence guarantees. Therefore, we are motivated to ask the following fundamental question.

- **Q:** *Can we develop globally convergent algorithms that can efficiently find local minimax points in nonconvex minimax optimization? What are their convergence rates and complexities?*

In this work, we provide comprehensive answers to these questions. We develop deterministic and stochastic cubic regularization type algorithms that globally converge to local minimax points in nonconvex minimax optimization, and study their convergence rates, computation complexities and sample complexities.

## 1.1 Our Contributions

We consider the minimax optimization problem (P), where $f$ is twice-differentiable with Lipschitz continuous gradient and Hessian and is nonconvex-strongly-concave. In this setting, we first show that local minimax points of $f$ are equivalent to second-order stationary points of the envelope function $\Phi(x) := \max_{y \in \mathbb{R}^n} f(x, y)$. Then, inspired by the classic cubic regularization algorithm, we propose an algorithm named Cubic-LocalMinimax to find local minimax points. The algorithm uses gradient ascent to update $y$, which is then used to estimate the gradient and Hessian involved in the cubic regularization update for $x$ (see Algorithm 1 for more details).

**Global convergence.** We show that Cubic-LocalMinimax admits an intrinsic potential function $H_t$ (see Proposition 4.1) that monotonically decreases over the iterations. Based on this property, we prove that every limit point of $\{x_t\}_t$ generated by Cubic-LocalMinimax is a local minimax point. Moreover, to achieve

an $\epsilon$-accurate local minimax point, Cubic-LocalMinimax requires $\mathcal{O}\big(L_2\kappa^{1.5}\epsilon^{-3}\big)$ number of cubic updates and $\widetilde{\mathcal{O}}\big(L_2\kappa^{2.5}\epsilon^{-3}\big)$ number of gradient ascent updates, where $\kappa > 1$ denotes the problem condition number.

**GDA-Cubic solver.** The updates of Cubic-LocalMinimax involve a cubic subproblem that has a very special Hessian structure. To solve this subproblem, we reformulate it as a minimax optimization problem, for which we develop a GDA-type solver (see Algorithm 2). We name such a variant of Cubic-LocalMinimax as Inexact Cubic-LocalMinimax. By bounding the approximation error of the cubic solver carefully, we establish a monotonically decreasing potential function and establish the same iteration complexity as that of Cubic-LocalMinimax. Moreover, the total number of Hessian-vector product computations involved in the cubic solver is of the order $\widetilde{O}(L_1\kappa^2\epsilon^{-4})$.

**Sample complexity.** We further develop a stochastic variant of Cubic-LocalMinimax named as Stochastic Cubic-LocalMinimax, which applies stochastic sub-sampling to improve the sample complexity in large-scale minimax optimization. In particular, we adopt time-varying batch sizes in a way such that the induced gradient inexactness and Hessian inexactness are adapted to the optimization increment $\|x_t - x_{t-1}\|$ in the previous iteration. Consequently, to achieve an $\epsilon$-accurate local minimax point, stochastic Cubic-LocalMinimax requires querying $\widetilde{\mathcal{O}}(\kappa^{3.5}\epsilon^{-7})$ number of gradient samples and $\widetilde{\mathcal{O}}(\kappa^{2.5}\epsilon^{-5})$ number of Hessian samples.

| Reference | Algorithm | Setting | $f(\cdot, y)$ | Metric[1] | #Gradients[2] |
|---|---|---|---|---|---|
| Nouiehed et al. (2019) | Multi-step GDA | Deterministic | PL | $\epsilon$-Nash | $\mathcal{O}\big(\epsilon^{-2}\ln(\epsilon^{-1})\big)$ |
| Nouiehed et al. (2019) | Multi-step GDA | Deterministic | Concave | $\epsilon$-Nash | $\mathcal{O}\big(\epsilon^{-3.5}\ln(\epsilon^{-1})\big)$ |
| Lin et al. (2020) | GDA | Deterministic | Strongly-concave | $\epsilon$-stationary | $\mathcal{O}(\kappa^2\epsilon^{-2})$ |
| Xu et al. (2023) | AGP | Deterministic | Strongly-concave | $\epsilon$-stationary | $\mathcal{O}(\kappa^5\epsilon^{-2})$ |
| Lin et al. (2020) | GDA | Deterministic | Concave | $\epsilon$-stationary | $\mathcal{O}(\epsilon^{-6})$ |
| Xu et al. (2023) | AGP | Deterministic | Concave | $\epsilon$-stationary | $\mathcal{O}(\epsilon^{-4})$ |
| Lin et al. (2020) | GDA | Stochastic | Strongly-concave | $\epsilon$-stationary | $\mathcal{O}(\kappa^3\epsilon^{-4})$ |
| Lin et al. (2020) | GDA | Stochastic | Concave | $\epsilon$-stationary | $\mathcal{O}(\epsilon^{-8})$ |
| Xu et al. (2020b) | SREDA | Stochastic | Strongly-concave | $\epsilon$-stationary | $\mathcal{O}(\kappa^3\epsilon^{-3})$ |
| Qiu et al. (2020) | VR-STS | Stochastic | Strongly-concave | $\epsilon$-stationary | $\mathcal{O}(\epsilon^{-3})$ |

Table 1: Comparison of first-order algorithms for nonconvex minimax optimization.

[1] "Metric" denotes the point to which the algorithm will converge, with the following choices:
· $\epsilon$-Nash: $\|\nabla_x f(x,y)\| \le \epsilon$, $\|\nabla_y f(x,y)\| \le \epsilon$.
· $\epsilon$-stationary: $\|\nabla\Phi(x)\| \le \epsilon$ where $\Phi(x) := \max_{y\in\mathbb{R}^n} f(x,y)$.
[2] "# Gradients" denotes the required number of gradient evaluations.

| Reference | Algorithm | Setting | #Gradients | #Hv products[1] |
|---|---|---|---|---|
| Luo et al. (2022) | Inexact MCN | Deterministic | $\widetilde{\mathcal{O}}\big(\kappa^2\epsilon^{-1.5}\big)$ | $\mathcal{O}\big(\kappa^{1.5}\epsilon^{-2}\big)$ |
| Our Theorem 2 | Algorithm 4 | Deterministic | $\widetilde{\mathcal{O}}\big(\kappa^{2.5}\epsilon^{-1.5}\big)$ | $\widetilde{\mathcal{O}}\big(\kappa^2\epsilon^{-2}\big)$ |
|  | Accelerated Algorithm 4 | Deterministic | $\widetilde{\mathcal{O}}\big(\kappa^2\epsilon^{-1.5}\big)$ | $\widetilde{\mathcal{O}}\big(\kappa^{1.5}\epsilon^{-2}\big)$ |
| Our Theorem 4 | Algorithm 5 | Stochastic | $\widetilde{\mathcal{O}}\big(\kappa^{3.5}\epsilon^{-3.5}\big)$ | $\widetilde{\mathcal{O}}\big(\kappa^{2.5}\epsilon^{-2.5}\big)$ |

Table 2: Comparison of second-order algorithms for finding local minimax points in nonconvex-strongly-concave optimization.

[1] The required number of Hessian-vector product evaluations.

## 1.2 Other Related Works

**Deterministic first-order algorithms for minimax optimization :** Under strongly-convex-strongly-concave setting, Zhang and Wang (2021) obtained iteration complexity $\mathcal{O}\big(\kappa^{1.5}\ln(\epsilon^{-1})\big)$ for optimistic gradient descent-ascent (OGDA) algorithm, and Mokhtari et al. (2020) obtained iteration complexity $\mathcal{O}\big(\kappa\ln(\epsilon^{-1})\big)$ for both OGDA and extra-gradient (EG) algorithm which applies two-step GDA in each iteration. Yang et al. (2020) studied an alternating gradient descent-ascent (AGDA) algorithm in which the gradient ascent step uses the current variable $x_{t+1}$ instead of $x_t$ and achieved iteration complexity $\mathcal{O}\big(\kappa^3\ln(\epsilon^{-1})\big)$ for two-sided Polyak

Łojasiewicz (PL) minimax optimization. These works require require strong geometric assumptions such as strong convexity and two-sided PL, while our work focuses on general nonconvex minimax optimization. Nonconvex minimax optimization is also widely studied in existing literature. For example, Xu et al. (2023) studied an alternating gradient projection (AGP) algorithm which applies $\ell_2$ regularizer to the local objective function of GDA followed by projection onto the constraint sets and achieves iteration complexities $\mathcal{O}(\epsilon^{-4})$ and $\mathcal{O}(\kappa^5\epsilon^{-2})$ under nonconvex-concave and nonconvex-strongly-concave settings, respectively. Nouiehed et al. (2019) studied multi-step GDA with multiple gradient ascent steps per iteration and iteration complexity $\mathcal{O}(\epsilon^{-2}\ln(\epsilon^{-1}))$ under nonconvex-PL setting , and they also studied the momentum-accelerated version which achieves iteration complexity $\mathcal{O}(\epsilon^{-3.5}\ln(\epsilon^{-1}))$ under nonconvex-concave setting. However, these works under nonconvex setting only have convergence guarantee to stationary points, whereas all the variants of our Cubic-LocalMinimax algorithm converge to local minimax points. More detailed comparison of the convergence results of the above deterministic first-order algorithms are shown in Section 1.1.

**Deterministic second-order algorithms for minimax optimization :** Adolphs et al. (2019) analyzed the stability of a second-order variant of the GDA algorithm. Huang et al. (2022b) also proposed a cubic regularized Newton algorithm which solves convex-concave minimax optimization problem with $\mathcal{O}(\ln(\epsilon^{-1}))$ and $\mathcal{O}(\epsilon^{(1-\theta)/(\theta^2)})$ iterations respectively under Lipschitz-type and Hölder-type (with parameter $\theta$) error bound conditions, but the computation complexity for solving the cubic subproblem is lacking. In (Luo et al., 2022), the authors proposed a Minimax Cubic-Newton (MCN) algorithm that is different from our Cubic-LocalMinimax in various aspects, as we elaborate with more details in Sections 4 and 5. We list three major differences as follows. First, we develop a GDA-based solver for solving the cubic subproblem which does not require cubic objective evaluation, whereas they use a gradient-based solver with Chebyshev polynomials which requires additional computation. Second, our algorithm requires much fewer gradient ascent steps in practice since we adopt more adaptive approximation error bounds (see eqs. (3) & (4)). Third, we develop a stochastic version of Cubic-LocalMinimax and analyze its sample complexity, which to our knowledge has not been studied in the existing literature. More detailed comparison of the convergence results between (Luo et al., 2022) and our work are shown in Section 1.1.

**Stochastic algorithms for minimax optimization :** Lin et al. (2020); Yang et al. (2020) analyzed stochastic GDA and stochastic AGDA, respectively, which are direct extensions of GDA and AGDA to the stochastic setting. Specifically, stochastic GDA algorithm (Lin et al., 2020) achieves sample complexities $\mathcal{O}(\kappa^3\epsilon^{-4})$ and $\mathcal{O}(\epsilon^{-8})$ under nonconvex-strongly concave and nonconvex-concave settings respectively. (Yang et al., 2020) proposed stochastic AGDA algorithm and its accelerated version via SVRG variance reduction technique, which achieve sample complexities $\mathcal{O}(\kappa^5\epsilon^{-1})$ and $\mathcal{O}(N + \kappa^9)\ln(\epsilon^{-1})$ respectively for finite-sum two-sided PL minimax optimization with $N$ samples. First-order variance reduction reduction techniques have also been applied to nonconvex strongly-concave stochastic minimax optimization. For example, SREDA (Xu et al., 2020b) and STORM (Qiu et al., 2020) achieve sample complexities $\mathcal{O}(\kappa^3\epsilon^{-3})$ and $\mathcal{O}(\epsilon^{-3})$ respectively. More detailed comparison of the convergence results of the above stochastic first-order algorithms are shown in Section 1.1. Gradient-free versions of these variance reduction techniques have also been applied to minimax optimization, including SPIDER (Xu et al., 2020c) and STORM (Huang et al., 2022a). Compared with our work which has convergence guarantee to local minimax points of nonconvex minimax optimization, the above stochastic algorithms either require strong geometric assumptions such as strong convexity and two-sided PL, or only converge to stationary points of nonconvex minimax optimization.

**Cubic regularization (CR):** The CR algorithm dates back to (Griewank, 1981), where global convergence is established. In (Nesterov and Polyak, 2006), the author analyzed the convergence rate of CR to second-order stationary points in nonconvex optimization. In (Nesterov, 2008), the authors established the sub-linear convergence of CR for solving convex smooth problems, and they further proposed an accelerated version of CR with improved sub-linear convergence. Yue et al. (2019) studied the asymptotic convergence properties of CR under the error bound condition, and established the quadratic convergence of the iterates. Recently, Hallak and Teboulle (2020) proposed a framework of two-directional method for finding second-order stationary points in general smooth nonconvex optimization. The main idea is to search for a feasible direction toward the solution and is not based on cubic regularization. Several other works proposed different methods to solve the cubic subproblem of CR, e.g., (Agarwal et al., 2017; Carmon and Duchi, 2019; Cartis et al., 2011b). Another line of work aimed at improving the computation efficiency of CR by solving the cubic subproblem

with inexact gradient and Hessian information. In particular, Ghadimi et al. (2017) proposed an inexact CR for solving convex problem. Also, Cartis et al. (2011a) proposed an adaptive inexact CR for nonconvex optimization, whereas Jiang et al. (2017) further studied the accelerated version for convex optimization. Several studies explored subsampling schemes to implement inexact CR algorithms, e.g., (Kohler and Lucchi, 2017; Xu et al., 2020a; Zhou and Liang, 2018; Wang et al., 2018b).

## 2 Problem Formulation and Preliminaries

We consider the following standard minimax optimization problem (P), where $f$ is a nonconvex-strongly-concave bivariate function and is twice-differentiable. Throughout the paper, we define the envelope function $\Phi(x) := \max_{y \in \mathbb{R}^n} f(x, y)$.

$$\min_{x \in \mathbb{R}^m} \max_{y \in \mathbb{R}^n} \ f(x, y). \tag{P}$$

Our goal is to develop algorithms that converge to a local minimax point of (P), which is defined as follows.

**Definition 2.1** (Local minimax point Jin et al. (2020))**.** *A point $(x^*, y^*)$ is a local minimax point of (P) if $y^*$ is a local maximum of function $f(x^*, \cdot)$, and there exists a constant $\delta_0 > 0$ such that $x^*$ is a local minimum of function $g_\delta(x) := \max_{y: \|y - y^*\| \le \delta} f(x, y)$ for any $\delta \in (0, \delta_0]$.*

Local minimax points are different from global minimax points, which require $x^*$ and $y^*$ to be the global minimizer and global maximizer of the functions $\Phi(\cdot)$ and $f(x, \cdot)$ simultaneously. In general minimax optimization, it has been shown that global minimax points can be neither local minimax points nor even stationary points (Jin et al., 2020). However, global minimax point necessarily implies local minimax point in nonconvex-strongly-concave optimization. Moreover, under mild conditions, many machine learning problems have been shown to possess local minimax points, e.g., generative adversarial networks (GANs) (Nagarajan and Kolter, 2017; Zhang et al., 2021), distributional robust machine learning (Sinha et al., 2018), etc.

Local minimax point is also an extension (necessary condition) of local Nash equilibrium (Jin et al., 2020). A point $(x^*, y^*)$ is defined as local Nash equilibrium if $f(x^*, y) \le f(x^*, y^*) \le f(x, y^*)$ for any $x, y$ that satisfies $\|x - x^*\| \le \delta$ and $\|y - y^*\| \le \delta$. In other words, local zero-duality gap $\min_{x: \|x - x^*\| \le \delta} \max_{y: \|y - y^*\| \le \delta} f(x, y) = \max_{y: \|y - y^*\| \le \delta} \min_{x: \|x - x^*\| \le \delta} f(x, y)$ is achieved at $(x^*, y^*)$. Such zero-duality gap condition is required for simultaneous games where the min-player $x$ and the max-player $y$ act simultaneously (Jin et al., 2020). In fact, Jin et al. (2020) pointed out that most machine learning applications are sequential games where min-player and max-player act sequentially and the sequence (i.e., who plays first) is crucial and pre-specified. For example, in adversarial training, classifier acts first and then the adversary generates an adversarial sample. In GAN training, generator acts first followed by discriminator. Moreover, in sequential games, zero duality gap does not necessarily hold, i.e., local Nash equilibrium may not exist, so local minimax solution is more commonly used (Jin et al., 2020).

In (Jin et al., 2020), the following set of second-order conditions have been proved to be sufficient conditions for local minimax points. Moreover, as we show later, our algorithm design is inspired by these conditions.

**Definition 2.2** (Sufficient conditions for local minimax)**.** *A point $(x, y)$ is a local minimax point of (P) if the following conditions hold.*

*1. Stationary: $\nabla_1 f(x, y) = \mathbf{0}$, $\nabla_2 f(x, y) = \mathbf{0}$;*

*2. Non-degeneracy: $\nabla_{22} f(x, y) \prec \mathbf{0}$, and $\left[\nabla_{11} f - \nabla_{12} f (\nabla_{22} f)^{-1} \nabla_{21} f\right](x, y) \succ \mathbf{0}$.*

Throughout the paper, we adopt the following standard assumptions on the minimax optimization problem (P). These conditions have been widely adopted in the related works (Lin et al., 2020; Jin et al., 2020; Zhang et al., 2021).

**Assumption 1.** *The minimax problem (P) satisfies:*

*1. Function $f(\cdot, \cdot)$ is $L_1$-smooth and function $f(x, \cdot)$ is $\mu$-strongly concave for any fixed $x$;*

*2. The Hessian mappings $\nabla_{11} f$, $\nabla_{12} f$, $\nabla_{21} f$, $\nabla_{22} f$ are $L_2$-Lipschitz continuous;*

3. *Function $\Phi(x) := \max_{y \in \mathbb{R}^n} f(x, y)$ is bounded below and has bounded sub-level sets.*

To elaborate, item 1 considers the class of nonconvex-strongly-concave functions $f$ that has been widely studied in the minimax optimization literature (Lin et al., 2020; Jin et al., 2020; Xu et al., 2023; Lu et al., 2020), and it is also satisfied by many machine learning applications. Item 2 assumes that the block Hessian matrices of $f$ are Lipschitz continuous, which is a standard assumption for analyzing second-order optimization algorithms (Nesterov and Polyak, 2006; Agarwal et al., 2017). Moreover, item 3 guarantees that the minimax problem has at least one solution.

## 3  A Cubic Regularization Approach for Finding Local Minimax Points

In this section, we propose a cubic regularization type algorithm that leverages the cubic regularization technique to find local minimax points of the nonconvex minimax problem (P). We first relate local minimax points to certain second-order stationary points in Section 3.1, based on which we further develop the algorithm in Section 3.2.

### 3.1  On Local Minimax and Second-Order Stationary Condition

Regarding the conditions of local minimax points listed in Definition 2.2, note that the stationary conditions in item 1 are easy to achieve, e.g., by performing standard gradient updates. For the non-degeneracy conditions listed in item 2, the first condition is guaranteed as $f(x, \cdot)$ is strongly concave. Therefore, the major challenge is to achieve the other non-degeneracy condition $\left[\nabla_{11}f - \nabla_{12}f(\nabla_{22}f)^{-1}\nabla_{21}f\right](x, y) \succ \mathbf{0}$. Interestingly, in nonconvex-strongly-concave minimax optimization, such a non-degeneracy condition has close connections to a certain second-order stationary condition on the envelope function $\Phi(x)$, as formally stated in the following proposition. Throughout, we denote $\kappa = L_1/\mu$ as the condition number.

**Proposition 3.1.** *Let Assumption 1 hold. Then, the following statements hold.*

1. *The mapping $y^*(x) := \arg\max_{y \in \mathbb{R}^n} f(x, y)$ is unique and $\kappa$-Lipschitz continuous for every fixed $x$ (Lin et al., 2020);*

2. *$\Phi(x)$ is $L_1(1 + \kappa)$-smooth and $\nabla\Phi(x) = \nabla_1 f(x, y^*(x))$ (Lin et al., 2020);*

3. *Define mapping $G(x, y) = \left[\nabla_{11}f - \nabla_{12}f(\nabla_{22}f)^{-1}\nabla_{21}f\right](x, y)$. Then, $G$ is a Lipschitz continuous mapping with Lipschitz constant $L_G = L_2(1 + \kappa)^2$;*

4. *The Hessian of $\Phi$ satisfies $\nabla^2\Phi(x) = G(x, y^*(x))$, and it is Lipschitz continuous with Lipschitz constant $L_\Phi = L_G(1 + \kappa) = L_2(1 + \kappa)^3$.*

The above proposition points out that the non-degeneracy condition $G(x, y) \succ \mathbf{0}$ actually corresponds to a second order stationary condition of the envelop function $\Phi(x)$. To explain more specifically, consider a pair of points $(x, y^*(x))$, in which $y^*(x) := \arg\max_y f(x, y)$. Since $f(x, \cdot)$ is strongly concave and $y^*(x)$ is the maximizer, we know that $y^*(x)$ must satisfy the stationary condition $\nabla_2 f(x, y^*(x)) = \mathbf{0}$ and the non-degeneracy condition $\nabla_{22}f(x, y^*(x)) \prec \mathbf{0}$. Therefore, in order to be a local minimax point, $x$ must satisfy the stationary condition $\nabla_1 f(x, y^*(x)) = \mathbf{0}$ and the non-degeneracy condition $G(x, y^*(x)) \succ \mathbf{0}$, which, by items 2 and 4 of Proposition 3.1, are equivalent to the set of second-order stationary conditions stated in the following fact.

**Fact 1.** *(Evtushenko, 1974) Let Assumption 1 hold. Then, $(x, y^*(x))$ is a local minimax point of (P) if $x$ satisfies the following set of second-order stationary conditions.*

$$\text{(Second-order stationary):} \quad \nabla\Phi(x) = \mathbf{0}, \quad \nabla^2\Phi(x) \succ \mathbf{0}.$$

To summarize, to find a local minimax point in nonconvex-strongly-concave minimax optimization, it suffices to find a second-order stationary point of the smooth nonconvex envelope function $\Phi(x)$. Such a key observation is the basis for developing our proposed algorithm in the next subsection. We also note that the proof of Proposition 3.1 is not trivial. Specifically, we need to first develop bounds for the spectrum

norm of the block Hessian matrices in Lemma A.1 (see the first page of the appendix), which helps prove the Lipschitz continuity of the $G$ mapping in item 1. Moreover, we leverage the optimality condition of $f(x, \cdot)$ to derive an expression for the maximizer mapping $y^*(x)$ (see eq. (46) in the appendix), which is used to further prove item 2.

### 3.2 The Cubic-LocalMinimax Algorithm

In the existing literature, many second-order optimization algorithms have been developed for finding second-order stationary points of nonconvex minimization problems (Nesterov and Polyak, 2006; Agarwal et al., 2017; Yue et al., 2019; Zhou et al., 2018). Hence, one may want to apply these algorithms to minimize the nonconvex function $\Phi(x)$ and find local minimax points of the minimax problem (P). However, these algorithms are not directly applicable, as the function $\Phi(x)$ involves a special *maximization* structure and hence its specific function form $\Phi$ as well as the gradient $\nabla\Phi$ and Hessian $\nabla^2\Phi$ are implicit. Instead, our algorithm design can only leverage information of the bi-variate function $f$.

Our algorithm design is inspired by the classic cubic regularization algorithm (Nesterov and Polyak, 2006). Specifically, to find a second-order stationary point of the envelope function $\Phi(x)$, the conventional cubic regularization algorithm would perform the following iterative update.

$$s_{t+1} \in \arg\min_s \nabla\Phi(x_t)^\top s + \frac{1}{2}s^\top \nabla^2\Phi(x_t)s + \frac{1}{6\eta_x}\|s\|^3,$$
$$x_{t+1} = x_t + s_{t+1}, \tag{1}$$

where $\eta_x > 0$ is a proper learning rate. However, due to the special maximization structure of $\Phi$, its gradient and Hessian have complex formulas (see Proposition 3.1 ) that involve the mapping $y^*(x)$, which cannot be computed exactly in practice. Hence, we aim to develop an algorithm that efficiently computes approximations of $\nabla\Phi(x), \nabla^2\Phi(x)$, and use them to perform the cubic regularization update.

To perform the cubic regularization update in eq. (1), we need to compute $\nabla\Phi(x_t) = \nabla_1 f(x_t, y^*(x_t))$ and $\nabla^2\Phi(x_t) = G(x_t, y^*(x_t))$ (by Proposition 3.1), both of which depend on the maximizer $y^*(x_t)$ of the function $f(x_t, \cdot)$. Since $f(x_t, \cdot)$ is strongly-concave, we can run $N_t$ iterations of gradient ascent to obtain an approximated maximizer $\widetilde{y}_{N_t} \approx y^*(x_t)$, and then approximate $\nabla\Phi(x_t), \nabla^2\Phi(x_t)$ using $\nabla_1 f(x_t, \widetilde{y}_{N_t})$ and $G(x_t, \widetilde{y}_{N_t})$, respectively. Intuitively, these are good approximations due to two reasons: (i) $\widetilde{y}_{N_t}$ converges to $y^*(x_t)$ at a fast linear convergence rate; and (ii) both $\nabla_1 f$ and $G$ are shown to be Lipschitz continuous in their second argument. We refer to this algorithm as **Cubic Regularization for Local Minimax (Cubic-LocalMinimax)**, and summarize its update rule in Algorithm 1 below, which terminates whenever the maximum of the previous two increments $\|s_{t-1}\| \vee \|s_t\|$ is below a certain threshold $\epsilon'$. Such an output rule helps characterize the computation complexity of the algorithm. In Section 5, we provide a comprehensive discussion on how to solve the cubic subproblem with the special Hessian matrix $G(x_t, y_{t+1})$ using first-order GDA-type algorithms.

## 4 Convergence and Iteration Complexity of Cubic-LocalMinimax

In this section, we study the global convergence properties and the iteration complexity of Cubic-LocalMinimax. The key to our convergence analysis is characterizing an intrinsic potential function of Cubic-LocalMinimax in nonconvex minimax optimization. We formally present this result in the following proposition.

**Proposition 4.1** (Potential function). *Let Assumption 1 hold. For any $\alpha, \beta > 0$, choose $\epsilon' \leq \frac{\alpha L_1}{\beta L_G}$, $\eta_x \leq (9L_\Phi + 18\alpha + 28\beta)^{-1}$ and $\eta_y = \frac{2}{L_1 + \mu}$. Define the potential function $H_t := \Phi(x_t) + (L_\Phi + 2\alpha + 3\beta)\|s_t\|^3$. Then, when $N_t \geq \mathcal{O}\left(\kappa \ln \frac{L_1\alpha\|s_{t-1}\| + L_1(\alpha + L_2\kappa)\|s_t\|}{L_G\beta\epsilon'^2}\right)$, the output of Cubic-LocalMinimax satisfies the following potential function decrease property for all $t \in \mathbb{N}$.*

$$H_{t+1} - H_t \leq -(L_\Phi + \alpha + \beta)\big(\|s_{t+1}\|^3 + \|s_t\|^3\big). \tag{2}$$

Proposition 4.1 reveals that Cubic-LocalMinimax admits an intrinsic potential function $H_t$, which takes the form of the envelope function $\Phi(x)$ plus the cubic increment term $\|s_t\|^3$. Moreover, the potential function $H_t$

---

**Algorithm 1** Cubic-LocalMinimax

---

**Input:** Initialize $x_0, y_0$, learning rates $\eta_x, \eta_y$, threshold $\epsilon'$, numbers of iterations $T$, $N_t$

Define $\|s_0\| = \epsilon'$

**for** $t = 0, 1, 2, \ldots, T - 1$ **do**

    Initialize $\widetilde{y}_0 = y_t$

    **for** $k = 0, 1, 2, \ldots, N_t - 1$ **do**

$$\widetilde{y}_{k+1} = \widetilde{y}_k + \eta_y \nabla_2 f(x_t, \widetilde{y}_k)$$

    **end**

    Set $y_{t+1} = \widetilde{y}_{N_t}$. Solve the cubic problem for $s_{t+1}$:

$$\operatorname*{argmin}_s \nabla_1 f(x_t, y_{t+1})^\top s + \frac{1}{2} s^\top G(x_t, y_{t+1}) s + \frac{1}{6\eta_x} \|s\|^3$$

    Update $x_{t+1} = x_t + s_{t+1}$

**end**

**Output:** $x_{T'}, y_{T'}$, $T' = \min\{t : \|s_{t-1}\| \vee \|s_t\| \le \epsilon'\}$

---

is monotonically decreasing along the optimization path of Cubic-LocalMinimax, implying that the algorithm continuously makes optimization progress.

The key for establishing such a potential function is that, by running a sufficient number of inner gradient ascent iterations, we can obtain a sufficiently accurate approximated maximizer $y_{t+1} \approx y^*(x_t)$. Consequently, the $\nabla_1 f(x_t, y_{t+1})$ and $G(x_t, y_{t+1})$ involved in the cubic subproblem are good approximations of $\nabla \Phi(x_t)$ and $\nabla^2 \Phi(x_t)$, respectively. In fact, the approximation errors are proven to satisfy the following bounds.

$$\|\nabla \Phi(x_t) - \nabla_1 f(x_t, y_{t+1})\| \le \beta(\|s_t\|^2 + \epsilon'^2), \tag{3}$$

$$\|\nabla^2 \Phi(x_t) - G(x_t, y_{t+1})\| \le \alpha(\|s_t\| + \epsilon'). \tag{4}$$

On one hand, the above bounds are tight enough to establish the decreasing potential function. On the other hand, they are flexible and are adapted to the increment $\|s_t\| = \|x_t - x_{t-1}\|$ produced by the previous cubic update. Therefore, when the increment is large (i.e., $\|s_t\| = \mathcal{O}(1) \gg \mathcal{O}(\epsilon')$), which usually occurs in the early iterations, our algorithm requires only $\mathcal{O}(1)$ gradient ascent steps. As a comparison, the Minimax Cubic-Newton (MCN) algorithm proposed in (Luo et al., 2022) adopts constant-level approximation errors, i.e., $\|\nabla \Phi(x_t) - \nabla_1 f(x_t, y_{t+1})\| \le \mathcal{O}(\epsilon'^2)$ and $\|\nabla^2 \Phi(x_t) - G(x_t, y_{t+1})\| \le \mathcal{O}(\epsilon')^1$, which requires much more gradient ascent steps ($\mathcal{O}(\ln(1/\epsilon'))$) in every iteration. In the converging phase where $\|s_t\| \le \mathcal{O}(\epsilon')$, our algorithm requires $\mathcal{O}(\ln(1/\epsilon'))$ gradient ascent steps, which is of the same order as that of MCN. Combining the two cases, our algorithm is more practical. Such an idea of adapting the inexactness to the previous increment in eqs. (3) and (4) is further leveraged to develop a scalable stochastic variant of Cubic-LocalMinimax in Section 6.

Based on Proposition 4.1, we obtain the following global convergence rate of Cubic-LocalMinimax to a second-order stationary point of $\Phi$. Throughout, we adopt the following standard measure of second-order stationary introduced in (Nesterov and Polyak, 2006).

$$\mu(x) = \sqrt{\|\nabla \Phi(x)\|} \vee \frac{-\lambda_{\min}(\nabla^2 \Phi(x))}{\sqrt{33 L_\Phi}}.$$

Intuitively, a smaller $\mu(x)$ means that the point $x$ is closer to being second-order stationary.

**Theorem 1** (Convergence and complexity of Cubic-LocalMinimax)**.** *Let the conditions of Proposition 4.1 hold with $\alpha = \beta = L_\Phi$. For any $0 < \epsilon \le \frac{L_1 \sqrt{33 L_\Phi}}{L_G}$, choose $\epsilon' = \frac{\epsilon}{\sqrt{33 L_\Phi}}$ and $T \ge \frac{\Phi(x_0) - \Phi^* + 8 L_\Phi \epsilon'^2}{3 L_\Phi \epsilon'^3}$. Then, the output of Cubic-LocalMinimax satisfies*

$$\mu(x_{T'}) \le \epsilon. \tag{5}$$

---

[1] Our $\epsilon'$ corresponds to $\mathcal{O}(\sqrt{\epsilon})$ in Luo et al. (2022).

*Consequently, the total number of required cubic iterations satisfies $T' \leq \mathcal{O}\big(\sqrt{L_2}\kappa^{1.5}\epsilon^{-3}\big)$, and the total number of required gradient ascent iterations satisfies $\sum_{t=0}^{T'-1} N_t \leq \widetilde{\mathcal{O}}\big(\sqrt{L_2}\kappa^{2.5}\epsilon^{-3}\big)$.*

**Remark 1:** The convergence rate result in Theorem 1 is about the terminated iteration $T'$, which is specified by a stopping criterion and is upper bounded by $T$. This is different from most of the existing last-iterate convergence results where the last iterate $T$ is prespecified.

**Remark 2:** We note that the gradient ascent steps for updating $\widetilde{y}_{k+1}$ in Algorithm 1 can be accelerated by using the standard Nesterov's momentum. In this way, the total number of required gradient ascent iterations will reduce to the order $\widetilde{\mathcal{O}}\big(\sqrt{L_2}\kappa^2\epsilon^{-3}\big)$, which matches that of the Minimax Cubic-Newton algorithm proposed in (Luo et al., 2022).

The above theorem shows that the gradient norm $\|\nabla\Phi(x_t)\|$ vanishes at a sublinear rate $\mathcal{O}(T^{-\frac{2}{3}})$, and the second-order stationary measure $-\lambda_{\min}\big(\nabla^2\Phi(x)\big)$ converges at a sublinear rate $\mathcal{O}(T^{-\frac{1}{3}})$. Both results match the convergence rates of the cubic regularization algorithm for nonconvex minimization (Nesterov and Polyak, 2006). As a comparison, the standard GDA does not guarantee the convergence of $-\lambda_{\min}\big(\nabla^2\Phi(x)\big)$, and its convergence rate of $\|\nabla\Phi(x_t)\|$ is of the order $\mathcal{O}(T^{-\frac{1}{2}})$ (Lin et al., 2020), which is orderwise slower than that of Cubic-LocalMinimax. Therefore, by leveraging the curvature of the approximated Hessian matrix $G(x_t, y_{t+1})$, Cubic-LocalMinimax is able to find second-order stationary points of $\Phi$ at a fast rate.

We note that the proof of the global convergence results in Theorem 1 is critically based on the intrinsic potential function $H_t$ that we characterized in Proposition 4.1. Specifically, note that the cubic subproblem in Cubic-LocalMinimax involves an approximated gradient $\nabla_1 f(x_t, y_{t+1})$ and Hessian matrix $G(x_t, y_{t+1})$. Such inexactness of the gradient and Hessian introduces non-negligible noise to the cubic regularization update of Cubic-LocalMinimax. Consequently, Cubic-LocalMinimax cannot make monotonic progress on decreasing the function value $\Phi$, as opposed to the standard cubic regularization algorithm in nonconvex minimization (which uses exact gradient and Hessian). Instead, we take a different approach and show that as long as the gradient and Hessian approximations are sufficiently accurate, one can construct a monotonically decreasing potential function $H_t$ that leads to the desired global convergence guarantee.

## 5 How to Solve the Cubic Subproblem of Cubic-LocalMinimax?

The Cubic-LocalMinimax presented in Algorithm 1 involves a cubic subproblem that takes the following form.

$$s_{t+1} = \arg\min_s \phi(s) := g^\top s + \frac{1}{2}s^\top As + \frac{1}{6\eta_x}\|s\|^3, \tag{6}$$

$$\text{where } g = \nabla_1 f(x_t, y_{t+1}), \ A = H_{11} - H_{12}H_{22}^{-1}H_{21} \text{ with } H_{k\ell} = \nabla_{k\ell} f(x_t, y_{t+1}).$$

To solve the above cubic subproblem, one standard approach is to apply the existing gradient-based solvers (Carmon and Duchi, 2019; Tripuraneni et al., 2018), which requires computing the Hessian-vector product $A \cdot s$. However, in Cubic-LocalMinimax, the Hessian matrix takes the complex form $A = H_{11} - H_{12}H_{22}^{-1}H_{21}$ that involves product of block Hessian matrices as well as matrix inverse. Hence, directly computing the product of such a Hessian matrix with any vector can be highly inefficient. On the other hand, in (Luo et al., 2022), the authors proposed two-timescale update rules for computing such Hessian-vector product, where they approximate the matrix inverse $H_{22}^{-1}$ via Chebyshev polynomials[2]. To further simplify these update rules and reduce computation, we next propose an efficient GDA-type algorithm to solve this cubic subproblem with the special Hessian matrix $A$.

Our main idea is to reformulate the cubic subproblem in order to avoid the matrix inverse $H_{22}^{-1}$ involved in the Hessian matrix $A$. Specifically, we observe that the above cubic subproblem can be rewritten as the following minimax optimization problem.

$$\min_s \max_v \widetilde{\phi}(s, v) := g^\top s + \frac{1}{2}s^\top H_{11}s + s^\top H_{12}v + \frac{1}{2}v^\top H_{22}v + \frac{1}{6\eta_x}\|s\|^3. \tag{7}$$

---

[2]See eqs. (32)-(33) of (Luo et al., 2022).

To explain, note that the above bi-variate function $\widetilde{\phi}$ is strongly concave in $v$ with the unique maximizer given by $v^*(s) := \arg\max_v \widetilde{\phi}(s, v) = -H_{22}^{-1}H_{21}s$. Substituting this maximizer into the function $\widetilde{\phi}(s, \cdot)$ yields the original cubic subproblem, i.e., $\widetilde{\phi}(s, v^*(s)) = \phi(s)$. Moreover, since $\widetilde{\phi}(s, v)$ is a nonconvex-strongly-concave function (because $H_{22} \preceq -\mu I$), we are motivated to develop a GDA-type solver to solve it. Specifically, our solver, named as **GDA-Cubic Solver**, is partially inspired by the existing gradient-based cubic solvers (Tripuraneni et al., 2018) and is summarized in Algorithms 2 and 3. To elaborate, the solver performs updates based on the following two cases.

- Large gradient $\|g\| \geq 4L_1^2\kappa^2\eta_x$: In this case, the first-order gradient $g$ is far from being stationary, and it is more preferable to constrain the solution of the cubic subproblem in eq. (6) to $s = -\frac{\gamma}{\|g\|}g$ for some $\gamma > 0$ (Tripuraneni et al., 2018). In particular, the optimal choice $\gamma^*$, named as Cauchy radius, has been shown in (Conn et al., 2000) to take the following form.

$$\gamma^* := \arg\min_{\gamma \geq 0} \phi\left(-\gamma\frac{g}{\|g\|}\right) = \sqrt{\left(\frac{\eta_x g^\top A g}{\|g\|^2}\right)^2 + 2\eta_x\|g\|} - \frac{\eta_x g^\top A g}{\|g\|^2}. \tag{8}$$

Here, to compute the quantity $\frac{g^\top A g}{\|g\|^2}$ with $A = H_{11} - H_{12}H_{22}^{-1}H_{21}$, we propose to rewrite it as

$$\frac{g^\top A g}{\|g\|^2} = \frac{g^\top H_{11} g}{\|g\|^2} - \frac{(H_{21}g)^\top w^*}{\|g\|}, \text{ where } w^* := H_{22}^{-1}\frac{H_{21}g}{\|g\|}. \tag{9}$$

Note that both $H_{11}g$ and $H_{21}g$ are Hessian-vector products that can be efficiently computed by the popular machine learning platforms such as TensorFlow (Abadi, 2015) and PyTorch (Paszke, 2019). To compute $w^*$, note that it can be viewed as the unique maximizer of the $\mu$-strongly concave problem $\max_w \frac{1}{2}w^\top H_{22}w - \frac{(H_{21}g)^\top}{\|g\|}w$. We can solve this problem by performing $K$ gradient descent steps (see eq. (10)) and obtain an approximated minimizer $w_K \approx w^*$ with high accuracy.

- Small gradient $\|g\| < 4L_1^2\kappa^2\eta_x$: In this case, we propose to solve the equivalent cubic subproblem in eq. (7) via a nested-loop GDA-type algorithm, as it is nonconvex-strongly-concave. Specifically, we first fix $s$ and maximize $\widetilde{\phi}(s, \cdot)$ via gradient ascent for multiple iterations to estimate the maximizer $v^*(s)$ (see eq. (11)). Then, we fix $v$ and minimize $\widetilde{\phi}(\cdot, v)$ via one step of perturbed gradient descent (see eq. (12)).

We note that all the steps of GDA-Cubic Solver are based on computing Hessian-vector products, which can be efficiently computed and does not require storing the Hessian matrix. Equipped with this GDA-Cubic Solver, we propose the following Inexact Cubic-LocalMinimax algorithm summarized in Algorithm 4.

A similar algorithm named as Inexact Minimax Cubic-Newton (IMCN) with inexact cubic solver was first proposed by (Luo et al., 2022). Our Algorithm 4 differs from IMCN in the following aspects.

- First, as mentioned in Section 4, we adopt the more relaxed adaptive gradient and Hessian approximations in eqs. (3) & (4) for the gradient ascent steps, whereas they adopt constant approximation errors.

- Second , both our Inexact Cubic-LocalMinimax (Algorithm 4) and our GDA-based cubic solver (Algorithm 2) adopt simple termination rules that are purely based on tracking the norm of the increments $\|s_{t-1}\|, \|s_t\|$, which are directly accessible in each iteration. As a comparison, the termination rules of their Inexact Minimax Cubic-Newton Algorithm and its cubic solver need to additionally track the approximate objective function value of the cubic subproblem, which requires additional computation.

We obtain the following overall computation complexity result of Algorithm 4.

**Theorem 2** (Computation complexity of Inexact Cubic-LocalMinimax). *Let Assumption 1 hold. For any $0 < \epsilon \leq \min\left(\frac{53L_1\kappa}{228\sqrt{L_\Phi}}, L_1^2L_2^{-1/2}\kappa^{1/2}, \frac{L_2\kappa^2}{L_1}\right)$ and $\delta \in (0, 1)$, choose $\epsilon' = \frac{\epsilon}{106\sqrt{L_\Phi}}$, $T = \Theta\left(\sqrt{L_\Phi}[\Phi(x_0) - \Phi^* + \epsilon^2]\epsilon^{-3}\right)$, $\eta_x = \Theta(L_\Phi^{-1})$, $\eta_y = \frac{2}{L_1+\mu}$ and $N_t = \Theta\left(\kappa \ln \frac{L_1\alpha\|\widetilde{s}_{t-1}\| + L_1(\alpha+L_2\kappa)\|\widetilde{s}_t\|}{L_G\epsilon^2}\right)$ (see eq. (49)) in Algorithm 4. When implementing Algorithm 2 at the t-th iteration, use hyperparameters in Lemma G.1 with $\delta' = \delta/T$ if $\|\nabla_1 f(x_t, y_{t+1})\| \leq 4L_1^2\kappa^2\eta_x$, and use those in Lemma G.2 otherwise. When implementing Algorithm 3, use the hyperparameter choices in Lemma G.3. Then, with probability at least $1 - \delta$, the output of Inexact*

---

**Algorithm 2** GDA-Cubic Solver

---

**Input:** Gradient $g$, Hessians $H_{11}$, $H_{12}$, $H_{22}$, perturbation magnitude $\sigma$, learning rates $\eta_x, \eta_v, \eta_s$, numbers of iterations $K$, $\{N_k'\}_{k=0}^{K-1}$

**if** $\|g\| \geq 4L_1^2\kappa^2\eta_x$ **then**

    $w_0 = 0$

    **for** $k = 0, \ldots, K-1$ **do**

$$w_{k+1} = w_k + \eta_v\left(H_{22}w_k - \frac{H_{21}g}{\|g\|}\right) \tag{10}$$

    **end**

$$\begin{aligned}\beta_K &= \frac{g^\top}{\|g\|}H_{11}\frac{g}{\|g\|} - \frac{(H_{21}g)^\top w_K}{\|g\|}\\ \gamma_K &= \sqrt{(\eta_x\beta_K)^2 + 2\eta_x\|g\|} - \eta_x\beta_K\\ s_K' &= -\gamma_K g\end{aligned}$$

**else**

    Obtain $\xi \sim \text{Uniform}(\{x \in \mathbb{R}^m : \|x\| = 1\})$.

    **for** $k = 0, \ldots, K-1$ **do**

        $v_{k,0} = 0$

        **for** $\ell = 0, \ldots, N_k'-1$ **do**

$$v_{k,\ell+1} = v_{k,\ell} + \eta_v(H_{12}^\top s_k' + H_{22}v_{k,\ell}) \tag{11}$$

        **end**

        $v_k = v_{k,N_k'}$.

$$s_{k+1}' = s_k' - \eta_s\left(g + \sigma\xi + H_{11}s_k' + H_{12}v_k + \frac{\|s_k'\|}{2\eta_x}s_k'\right) \tag{12}$$

    **end**

**end**

**Output:** $s_K'$.

---

**Algorithm 3** GDA-Cubic FinalSolver

---

**Input:** Gradient $g$, Hessians $H_{11}$, $H_{12}$, $H_{22}$, learning rates $\eta_x, \eta_v, \eta_s$, numbers of iterations $K$, $\{N_k'\}_{k=0}^{K-1}$

**for** $k = 0, \ldots, K-1$ **do**

    $v_{k,0} = 0$

    **for** $\ell = 0, \ldots, N_k'-1$ **do**

        Obtain $v_{k,\ell+1}$ using eq. (11) with learning rate $\eta_v$.

    **end**

    $v_k = v_{k,N_k'}$.

$$g_k = g + H_{11}s_k' + H_{12}v_k + \frac{\|s_k'\|}{2\eta_x}s_k' \tag{13}$$

$$s_{k+1}' = s_k' - \eta_s g_k \tag{14}$$

**end**

**Output:** $s_{K'}'$, $K' = \min\{k : \|g_k\| \leq L_\Phi\epsilon'^2\}$

---

**Algorithm 4** Inexact Cubic-LocalMinimax

---

**Input:** Initialize $x_0, y_0$, learning rates $\eta_x, \eta_y$, threshold $\epsilon'$, numbers of iterations $T$, $N_t$

Define $\|\widetilde{s}_0\| = \epsilon'$

**for** $t = 0, 1, 2, \ldots, T-1$ **do**

    Initialize $\widetilde{y}_0 = y_t$

    **for** $k = 0, 1, 2, \ldots, N_t - 1$ **do**

$$\widetilde{y}_{k+1} = \widetilde{y}_k + \eta_y \nabla_2 f(x_t, \widetilde{y}_k)$$

    **end**

    Set $y_{t+1} = \widetilde{y}_{N_t}$

    Approximately solve the cubic problem $\mathrm{argmin}_s \nabla_1 f(x_t, y_{t+1})^\top s + \frac{1}{2} s^\top G(x_t, y_{t+1}) s + \frac{1}{6\eta_x} \|s\|^3$ for $\widetilde{s}_{t+1}$ using

    Algorithm 2 with $g := \nabla_1 f(x_t, y_{t+1})$ and $H_{k\ell} := \nabla_{k\ell} f(x_t, y_{t+1})$ $(k, \ell \in \{1, 2\})$

    Update $x_{t+1} = x_t + \widetilde{s}_{t+1}$

    **if** $\|\widetilde{s}_{t-1}\| \vee \|\widetilde{s}_t\| \leq \epsilon'$ **then**

        Obtain $\widetilde{s}$ using Algorithm 3 with $g := \nabla_1 f(x_t, y_{t+1})$ and $H_{k\ell} := \nabla_{k\ell} f(x_t, y_{t+1})$ $(k, \ell \in \{1, 2\})$

$$T' := \min\{t : \|\widetilde{s}_{t-1}\| \vee \|\widetilde{s}_t\| \leq \epsilon'\} \leftarrow t$$
$$\widetilde{x}_{T'} = x_{T'-1} + \widetilde{s}$$

        **Output:** $\widetilde{x}_{T'}, y_{T'}$

    **end**

**end**

---

*Cubic-LocalMinimax satisfies*

$$\mu(\widetilde{x}_{T'}) \leq \epsilon. \tag{15}$$

*Consequently, the total number of required cubic iterations satisfies $T' \leq \mathcal{O}\big(\sqrt{L_2} \kappa^{1.5} \epsilon^{-3}\big)$, the total number of required gradient ascent iterations satisfies $\sum_{t=0}^{T'-1} N_t \leq \widetilde{\mathcal{O}}\big(\sqrt{L_2} \kappa^{2.5} \epsilon^{-3}\big)$, and the total number of required Hessian-vector product computations (in Algorithms 2 & 3) is of the order $\widetilde{O}(L_1 \kappa^2 \epsilon^{-4})$.*

**Remark:** We can apply the standard Nesterov's momentum to further accelerate the convergence rate of the gradient ascent steps of Algorithms 2, 3 and 4. The resulting total number of gradient ascent iterations and total number of Hessian-vector products will then be improved to $\widetilde{\mathcal{O}}\big(\sqrt{L_2} \kappa^2 \epsilon^{-3}\big)$ and $\widetilde{O}(L_1 \kappa^{1.5} \epsilon^{-4})$, respectively, which match those of the Inexact Minimax Cubic-Newton algorithm in (Luo et al., 2022).

Compared with Theorem 1, it can be seen from the above Theorem 2 that the total number of cubic iterations and that of gradient ascent iterations remain the same, demonstrating the effectiveness of our proposed GDA-based cubic solver. Moreover, since our algorithm design and cubic solver design are different from those of (Luo et al., 2022), our convergence proof of Theorem 2 is therefore substantially different from that of (Luo et al., 2022) in the following aspects.

- First, our Algorithm 4 adopts the more relaxed adaptive approximation criteria (3) & (4) to save computation in practice, and thus cannot guarantee monotonic decrease of $\Phi(x_t)$. Instead, we established $\Phi(x_{t+1}) - \Phi(x_t) \leq -(11L_\Phi + 8\alpha + 11\beta)\|\widetilde{s}_{t+1}\|^3 + (9L_\Phi + 6\alpha + 9\beta)\|\widetilde{s}_t\|^3$ (see eq. (98)), which implies that our constructed potential function $H_t := \Phi(x_t) + (10L_\Phi + 7\alpha + 10\beta)\|\widetilde{s}_t\|^3$ is monotonically decreasing as shown in Proposition G.4.

- Second, as our Inexact Cubic-LocalMinimax (Algorithm 4) uses the termination rule $\|\widetilde{s}_{t-1}\| \vee \|\widetilde{s}_t\| \leq \epsilon'$ that only relies on the norm of the increments, we need to prove $\|s_{T'}\|, \|\widetilde{s}\| \leq \mathcal{O}(\epsilon')$ in order to ensure the second-order stationary condition $\mu(x) \leq \epsilon$. In particular, we have proved eq. (76) in Lemma G.2, which implies that $\|\widetilde{s}_{T'}\| \leq \epsilon'$ cannot hold under large gradient, so we conclude that $\|\nabla_1 f(x_{T'}, y_{T'+1})\| \leq 4L_1^2 \kappa^2 \eta_x$. In this small gradient case, eq. (64) we proved in Lemma G.1 implies that the exact CR solution $s_{T'}$ satisfies $\|s_{T'}\| \leq 3\epsilon'$, which combined with the final cubic solver (Algorithm 3) yields that the final CR solution $\widetilde{s}$ must satisfy eq. (82) (i.e., $\|\widetilde{s}\| \leq 7\epsilon'$) in Lemma G.3. As a comparison, the Inexact Minimax

Cubic-Newton Algorithm in (Luo et al., 2022) terminates based on tracking the objective function value of the cubic subproblem, which requires additional Hessian-vector product computation in practice and does not involve these technical developments.

## 6 Stochastic Cubic-LocalMinimax

In this section, we apply stochastic sub-sampling to further improve the performance and complexity of Cubic-LocalMinimax in large-scale nonconvex minimax optimization with big data. Specifically, we consider the following stochastic finite-sum minimax optimization problem (Q).

$$\min_{x \in \mathbb{R}^m} \max_{y \in \mathbb{R}^n} \ f(x,y) := \frac{1}{N} \sum_{i=1}^{N} f_i(x,y), \tag{Q}$$

where $N$ denotes the total number of training samples and $f_i$ corresponds to the loss function on the $i$-th sample. We adopt the following assumptions.

**Assumption 2.** *The stochastic minimax optimization problem* (Q) *satisfies:*

1. *For any sample $i$, function $f_i(\cdot,\cdot)$ is $L_1$-smooth, function $f_i(\cdot,y)$ is $L_0$-Lipschitz for any fixed $y$, and function $f_i(x,\cdot)$ is $\mu$-strongly concave for any fixed $x$;*

2. *For any sample $i$, the Hessian mappings $\nabla_{11} f_i$, $\nabla_{12} f_i$, $\nabla_{21} f_i$, $\nabla_{22} f_i$ are $L_2$-Lipschitz continuous;*

3. *Function $\Phi(x) := \max_{y \in \mathbb{R}^n} f(x,y)$ is bounded below and has bounded sub-level sets.*

Applying Cubic-LocalMinimax to solve the above problem (Q) would require querying full partial gradients and full Hessian matrices that involve all the training samples. Instead, it is more efficient to approximate these quantities via stochastic sub-sampling. Therefore, we propose a **stochastic Cubic-LocalMinimax** algorithm, which replaces the exact quantities $\nabla_1 f$ and $G$ involved in the cubic update by their corresponding stochastic approximations.

Specifically, to approximate the partial gradient $\nabla_1 f$, we sub-sample a mini-batch of samples $B_1$ with replacement from the training set and construct the following sample average approximation.

$$\widehat{\nabla}_1 f(x,y) = \frac{1}{|B_1|} \sum_{i \in B_1} \nabla_1 f_i(x,y). \tag{16}$$

On the other hand, to approximate the matrix $G$, we sub-sample mini-batches of samples $B_{11}, B_{12}, B_{21}, B_{22}$ with replacement and construct approximated Hessian matrices $\widehat{\nabla}_{11} f, \widehat{\nabla}_{12} f, \widehat{\nabla}_{21} f, \widehat{\nabla}_{22} f$ in the same way as above. Then, we construct the following approximation of $G$.

$$\widehat{G}(x,y) = \left[ \widehat{\nabla}_{11} f - \widehat{\nabla}_{12} f (\widehat{\nabla}_{22} f)^{-1} \widehat{\nabla}_{21} f \right](x,y). \tag{17}$$

We summarize the update rule of stochastic Cubic-LocalMinimax in Algorithm 5 below. In particular, we run stochastic gradient ascent (SGA) in the inner iterations to obtain the approximated maximizer $y_{t+1}$, and its high-probability convergence rate has been established in the existing stochastic optimization literature (Harvey et al., 2019) , as shown in Theorem 3 below.

**Theorem 3.** *Let Assumption 2 hold. For all $t, k$, assume that $\|\nabla_2 f(x_t, \widetilde{y}_k)\| \leq L_0$ and $\|\widehat{\nabla}_2 f(x_t, \widetilde{y}_k) - \nabla_2 f(x_t, \widetilde{y}_k)\| \leq 1$ almost surely. The inner stochastic gradient ascent steps in Algorithm 5 converge at the following rate with probability at least $1 - \delta$.*

$$\|y_{t+1} - y^*(x_t)\| \leq \mathcal{O}\left( \sqrt{\frac{L_0 \ln(1/\delta) + L_0^2}{\mu^2 N_t}} \right).$$

The following lemma characterizes the sample complexities of all the stochastic approximators for achieving a certain approximation accuracy.

---

**Algorithm 5** Stochastic Cubic-LocalMinimax

---

**Input:** Initialize $x_0, y_0$, learning rates $\eta_x$, threshold $\epsilon'$, numbers of iterations $T$, $N_t$

Define $\|s_0\| = \epsilon'$

**for** $t = 0, 1, 2, \ldots, T - 1$ **do**

    Initialize $\widetilde{y}_0 = y_t$

    **for** $k = 0, 1, 2, \ldots, N_t - 1$ **do**

        Query a sample $\xi$ to compute $\nabla_2 f_\xi(x_t, \widetilde{y}_k)$

$$\widetilde{y}_{k+1} = \widetilde{y}_k + \frac{2}{\mu(k+1)} \nabla_2 f_\xi(x_t, \widetilde{y}_k)$$

    **end**

    Set $y_{t+1} = \sum_{k=0}^{N_t - 1} \frac{2k}{N_t(N_t - 1)} \widetilde{y}_k$

    Sample minibatch $B_1(t), B_{11}(t), B_{12}(t), B_{21}(t), B_{22}(t)$ to compute eq. (16) and eq. (17). Then, solve the following cubic problem for $s_{t+1}$:

$$\operatorname*{argmin}_s \widehat{\nabla}_1 f(x_t, y_{t+1})^\top s + \frac{1}{2} s^\top \widehat{G}(x_t, y_{t+1}) s + \frac{1}{6\eta_x} \|s\|^3$$

    Update $x_{t+1} = x_t + s_{t+1}$

**end**

**Output:** $x_{T'}, y_{T'}$ with $T' = \min\{t : \|s_{t-1}\| \vee \|s_t\| \leq \epsilon'\}$

---

**Lemma 6.1.** *Fix any $0 < \epsilon_1 \leq 2L_0$, $0 < \epsilon_2 \leq 4L_1$ and choose the following batch sizes*

$$|B_1| \geq \mathcal{O}\Big(\frac{L_0^2}{\epsilon_1^2} \ln \frac{m}{\delta}\Big), \tag{18}$$

$$|B_{11}|, |B_{12}|, |B_{21}|, |B_{22}| \geq \mathcal{O}\Big(\frac{L_1^2}{\epsilon_2^2} \ln \frac{m+n}{\delta}\Big). \tag{19}$$

*Then, the stochastic approximators satisfy the following error bounds with probability at least $1 - \delta$.*

$$\|\widehat{\nabla}_1 f(x, y) - \nabla_1 f(x, y)\| \leq \epsilon_1, \tag{20}$$

$$\|\widehat{\nabla}_{k\ell}^2 f(x, y) - \nabla_{k\ell}^2 f(x, y)\| \leq \epsilon_2, \ \forall k, \ell \in \{1, 2\}, \tag{21}$$

$$\|\widehat{G}(x, y) - G(x, y)\| \leq (\kappa + 1)^2 \epsilon_2. \tag{22}$$

Therefore, by choosing proper batch sizes, the inexactness of the stochastic gradient, Hessian and Hessian estimators can be controlled within a desired range. From this perspective, stochastic Cubic-LocalMinimax can be viewed as an inexact version of the Cubic-LocalMinimax algorithm.

To characterize the convergence and sample complexity of stochastic Cubic-LocalMinimax, we adopt an adaptive inexactness criterion for the sub-sampling scheme. Specifically, we choose time-varying batch sizes in a way such that the gradient and Hessian inexactness in iteration $t$ are proportional to the previous increment, i.e., $\epsilon_1(t) \propto \|s_t\|^2, \epsilon_2(t) \propto \|s_t\|$. Such an adaptive inexact criterion has been justified in the cubic regularization literature (Wang et al., 2018a; 2019) with the following advantages: 1) it is adapted to the optimization increment and hence leads to reduced batch sizes when the increment is large in the early iterations; 2) it makes the batch size scheduling scheme in Lemma 6.1 practical, as the batch sizes in iteration $t$ now depend on the increment $\|s_t\|$ obtained in the previous iteration $t - 1$. We also note that since the sub-sampling scheme is adapted to the previous increment, the output rule of Algorithm 5 is designed to control the value of both the current and the previous increments. This termination rule is critical to bound the adapted gradient and Hessian inexactness in the analysis.

We obtain the following global convergence and sample complexity result of stochastic Cubic-LocalMinimax.

**Theorem 4** (Convergence and sample complexity). *Let Assumption 2 and Theorem 3 hold. For any $0 < \epsilon \leq \frac{L_1 \sqrt{33 L_\Phi}}{L_G}$, choose $\epsilon' = \frac{\epsilon}{\sqrt{33 L_\Phi}}$, $\eta_x \leq \frac{1}{55 L_\Phi}$, $T \geq \frac{\Phi(x_0) - \Phi^* + 8 L_\Phi \epsilon'^2}{3 L_\Phi \epsilon'^3}$ and $N_t \geq \mathcal{O}\Big(\frac{L_0 \ln(1/\delta) + L_0^2}{\kappa^{-2}(L_\Phi^2 \|s_t\|^4 + \epsilon^4) \wedge L_1^2(\|s_t\|^2 + \epsilon^2/L_\Phi)}\Big)$.*

*Moreover, in iteration t, choose the batch sizes according to eqs.* (18) *and* (19) *with the inexactness given by*

$$\epsilon_1(t) = \frac{L_\Phi}{2}\left(\|s_t\|^2 + \frac{\epsilon^2}{33 L_\Phi}\right) \wedge 2L_0, \quad \epsilon_2(t) = \frac{L_\Phi}{2(\kappa+1)^2}\left(\|s_t\| + \frac{\epsilon}{\sqrt{33 L_\Phi}}\right) \wedge 4L_1.$$

*Then, the output of Stochastic Cubic-LocalMinimax satisfies*

$$\mu(x_{T'}) \leq \epsilon. \tag{23}$$

*Consequently, the total number of cubic iterations satisfies* $T' \leq \mathcal{O}(\sqrt{L_2}\kappa^{1.5}\epsilon^{-3})$, *the total number of queried gradient samples satisfies* $\sum_{t'=0}^{T'}\left(N_t + |B_1(t)|\right) \leq \mathcal{O}\left(\frac{L_0^2\kappa^{3.5}\sqrt{L_2}}{\epsilon^7}\ln\frac{m}{\delta}\right)$, *and the total number of queried Hessian samples satisfies* $\sum_{t=0}^{T'-1}\sum_{k=1}^{2}\sum_{\ell=1}^{2}|B_{k,\ell}(t)| \leq \mathcal{O}\left(\frac{L_1^2\kappa^{2.5}}{\sqrt{L_2}\epsilon^5}\ln\frac{m+n}{\delta}\right)$.

Therefore, under adaptive sub-sampling, the induced gradient and Hessian inexactness $\epsilon_1(t), \epsilon_2(t)$ are properly controlled so that the iteration complexity $T'$ of stochastic Cubic-LocalMinimax remains in the same level as that of Cubic-LocalMinimax. Moreover, as opposed to the sample complexity of Cubic-LocalMinimax that scales linearly with regard to the data size $N$, the sample complexity of stochastic Cubic-LocalMinimax is independent of $N$. For example, by comparing the more expensive Hessian sample complexity between Cubic-LocalMinimax and stochastic Cubic-LocalMinimax, we conclude that stochastic sub-sampling helps reduce the sample complexity so long as the data size is large, i.e., $N \geq \widetilde{\mathcal{O}}\left(\frac{L_1^2\kappa}{L_2\epsilon^2}\right)$.

## 7  Experiments

In this section, we test the numerical performance of Cubic-LocalMinimax and compare it with existing algorithms in solving a synthetic minimax optimization problem and an adversarial deep learning problem. All the experiments are implemented on Google Colab with Intel(R) Xeon(R) CPU (12 cores, 2.20GHz), A100 GPU of cuda 11.2, and 83.48 GB memory. The code can be downloaded from `https://github.com/datou30/Cubic-Localminimax-experiment` .

### 7.1  Synthetic Minimax Optimization Problem

We consider the following finite-sum minimax optimization problem with parameters $x = [x_1, x_2, x_3] \in \mathbb{R}^3$ and $y = [y_1, y_2] \in \mathbb{R}^2$.

$$\min_{x\in\mathbb{R}^3}\max_{y\in\mathbb{R}^2} f(x,y) := \frac{1}{N}\sum_{i=1}^{N} f_i(x,y), \text{ with } f_i(x,y) = w(x_3) - \frac{y_1^2}{40} + A_i x_1 y_1 - \frac{5y_2^2}{2} + B_i x_2 y_2, \tag{24}$$

where $A_i, B_i > 0$ are independently drawn from a uniform distribution over the interval $[0.5, 1.5]$, $N = 1000$ is the total number of samples, and the function $w(\cdot)$ is a W-shaped nonconvex function whose exact form is presented in Appendix H.1. In this setting, each function $f_i$ is nonconvex-strongly-concave.

We apply our stochastic Cubic-LocalMinimax algorithm to solve the above synthetic problem. We choose the initialization point to be $x = [0.1; 0.1; 1], y = [1; 1]$ and set the batch size $|B_1(t)| = |B_{11}(t)| = |B_{12}(t)| = |B_{21}(t)| = |B_{22}(t)|$ to be $20, 100, 1000$, respectively. We choose the number of gradient ascent steps $N_t = 10$ for each outer loop, the strong concavity constant $\mu = 1$, and choose the learning rate $\eta_x = 0.01$. To solve the cubic subproblem, we use the standard gradient descent with learning rate 0.01, as the gradient and Hessian of $\Phi(x)$ can be analytically computed for this synthetic problem.

Figure 1 shows the performance at each epoch (each update of $x$ is considered as an epoch) (left figure) and time complexity (right figure) of stochastic Cubic-LocalMinimax under different batch sizes. Here, the y-axis denotes the function value of $\Phi(x) = \max_y f(x, y)$, which we aim to minimize. It can be seen that stochastic Cubic-LocalMinimax takes more epochs but much less time to converge under a smaller batch size. This is because the synthetic minimax problem involves relatively small noise so that the stochastic gradients computed over a small batch of samples are sufficiently accurate.

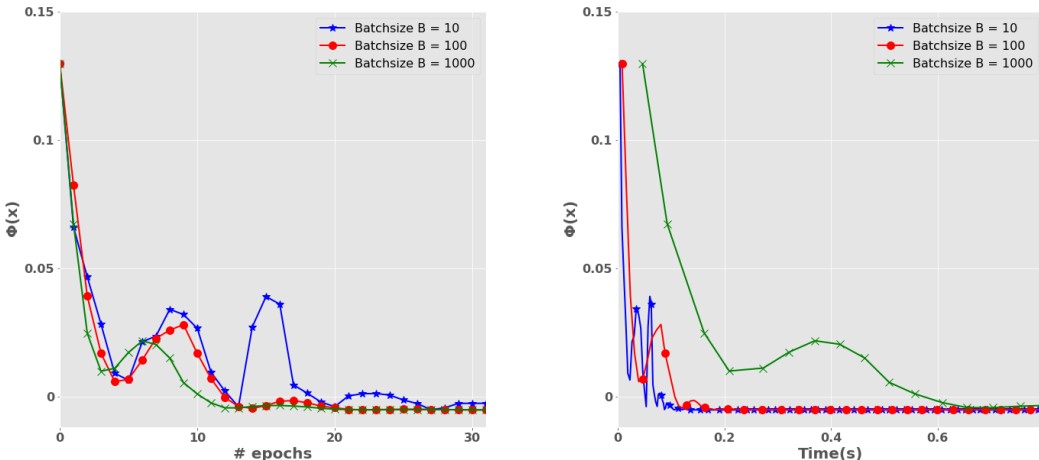

Figure 1: Performance at each epoch (left) and time complexity (right) of stochastic Cubic-LocalMinimax under different batch sizes. The $y$-axis denotes the function value of $\Phi(x) = \max_y f(x, y)$ that we aim to minimize.

We further compare the performance of stochastic Cubic-LocalMinimax with that of the standard stochastic GDA and second order algorithms including GDN/FR/TGDA/CN (Zhang et al., 2021) and Minimax Cubic-Newton (MCN) algorithm (Luo et al., 2022). All the algorithms use batch size 100 and the fixed initialization point $x = [0; 0; 1], y = [1; 1]$. For both stochastic Cubic-LocalMinimax and stochastic GDA , we choose the gradient ascent steps $N_t = 10$ and $\mu = 1$. We do one step of cubic descent and one step of gradient descent in each algorithm, respectively. The difference between the two algorithms is that the cubic solver in Algorithm 5 is replaced with one gradient descent step, and both cubic solver and gradient descent step have the same learning rate 0.01. The implementation details of GDN/TGDA/FR/CN algorithms are a bit involved and we refer them to Appendix H.3. For the MCN algorithm (Algorithm 2 of (Luo et al., 2022)), we apply 10 Nesterov's accelerated gradient ascent steps on $y$ with learning rate $\eta = 0.5$ and momentum coefficient $\theta = 0.5$.

Figure 2 shows the comparison of the performance at each epoch and time complexity for both algorithms. It can be seen that our stochastic Cubic-LocalMinimax has comparable performance to CN and MCN, but converges faster than all the other algorithms under both number of epochs and time complexity measures, demonstrating the effectiveness of both the cubic regularization approach and our proposed GDA-Cubic Solver.

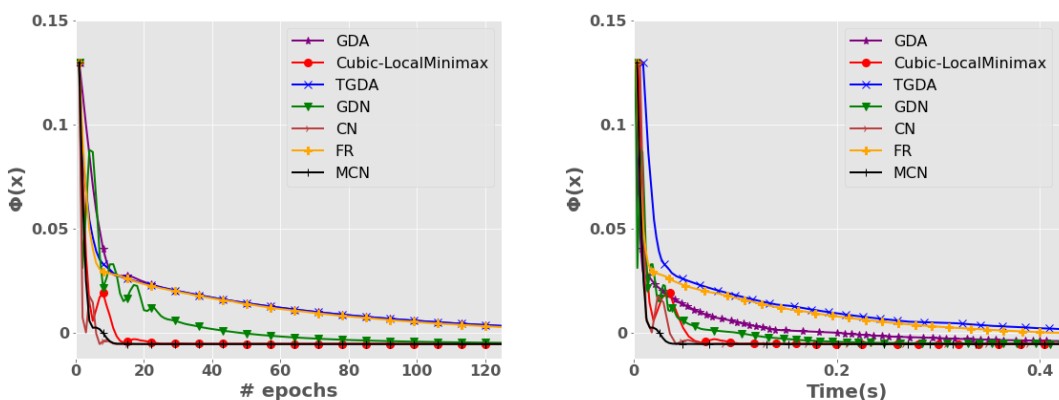

Figure 2: Comparison of all the algorithms for synthetic experiment.

## 7.2 Adversarial Deep Learning

We further compare the stochastic Cubic-LocalMinimax with the classical GDA and second order algorithms including GDN/FR/TGDA/CN (Zhang et al., 2021) and Inexact Minimax Cubic-Newton (IMCN) algorithm (Luo et al., 2022) in the application of adversarial deep learning, which aims to train an adversarially-robust convolutional neural network model (see Section H.2 for the detail of the neural network structure) by solving the following minimax optimization problem using the MNIST dataset with 50k training samples and 10k test samples.

$$\min_{\theta} \max_{\{\xi_i\}_{i=1}^{n}} \frac{1}{n} \sum_{i=1}^{n} \left[ \ell(h_\theta(\xi_i, y_i)) - \lambda \|\xi_i - x_i\|^2 \right], \tag{25}$$

where $n = 50k$ is the number of training samples, $\theta$ is the parameter of the neural network $h_\theta$, $(x_i, y_i)$ corresponds to the $i$-th image and label respectively, $\xi_i$ refers to the adversarial sample corresponding to $x_i$, and we choose cross-entropy loss function as $\ell$ and penalty coefficient $\lambda = 2.0$.

When implementing stochastic Cubic-LocalMinimax (Algorithm 5), we choose batchsize $|B_1(t)| = |B_{11}(t)| = |B_{12}(t)| = |B_{21}(t)| = |B_{22}(t)| = 512$ and implement $N_t = 20$ gradient ascent steps with learning rate 0.1. For the cubic solver, we implement Algorithm 2 with $\eta_v = 1$ when $\|g\| > 10$ until $K = 30$ iterations is reached or $\|H_{22}w_k - \frac{H_{21}g}{\|g\|}\| < 10^{-3}$ in the update rule (10); Otherwise, we choose $\sigma = 0$ (i.e. no random perturbation), $K = 30$, $\eta_x = 0.01$, $\eta_s = 0.002$, $\eta_v = 0.1$ and $N'_k = 5$ until either $K = 30$ iterations is reached or the gradient terms of both $s'_k$ and $v_k$ are sufficiently small (i.e., $\max \left( \|H_{12}^\top s'_k + H_{22}v_{k,\ell}\|, \|g + H_{11}s'_k + H_{12}v_k + \frac{\|s'_k\|}{2\eta_x}s'_k\| \right) < 10^{-3}$ in the update rules (11) & (12)). When implementing the classical GDA algorithm, we replace the cubic solver of stochastic Cubic-LocalMinimax with one stochastic gradient descent step to update $\theta$ using also batchsize 512 and learning rate 0.01, while the rest hyperparameters are not changed. The implementation details of GDN/TGDA/FR/CN algorithms are a bit involved and we refer them to Appendix H.3. For IMCN algorithm (Algorithm 3 of (Luo et al., 2022)), we use $\ell = 1$, $\mu = 0.4$ and the small gradient case of cubic-solver with learning rate 0.002 and early termination rule $\|\nabla\phi(s)\| < 0.001$ where $\phi$ is the cubic-regularization objective function. The Chebyshev polynomial used to compute Hessian-vector product is truncated to $K' = 5$ terms.

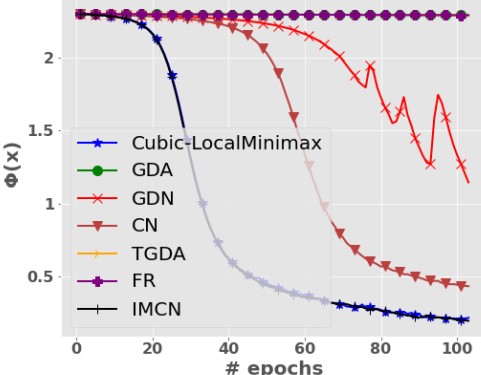
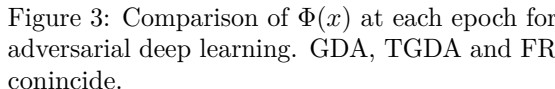

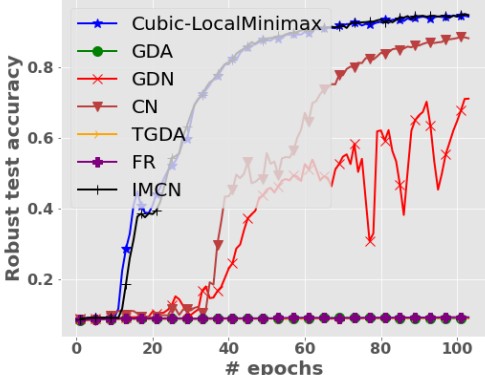

Figure 3: Comparison of $\Phi(x)$ at each epoch for adversarial deep learning. GDA, TGDA and FR conincide.

Figure 4: Comparison of robust test accuracy at each epoch for adversarial deep learning. GDA, TGDA and FR conincide.

Figure 3 compares the objective function value $\Phi(x)$ at each epoch of both algorithms, which is estimated via 40 gradient ascent steps with learning rate 0.1 to obtain the approximate maximizer $\{\xi_i\}_{i=1}^{n}$. It can be seen that our stochastic Cubic-LocalMinimax algorithm and IMCN are comparable and both algorithms have significantly faster convergence in terms of epoch than the other algorithms . Figure 4 further demonstrates

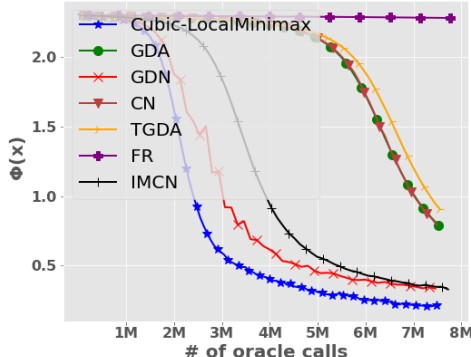

Figure 5: Comparison of $\Phi(x)$ per oracle call for adversarial deep learning.

Figure 6: Comparison of robust test accuracy per oracle call for adversarial deep learning .

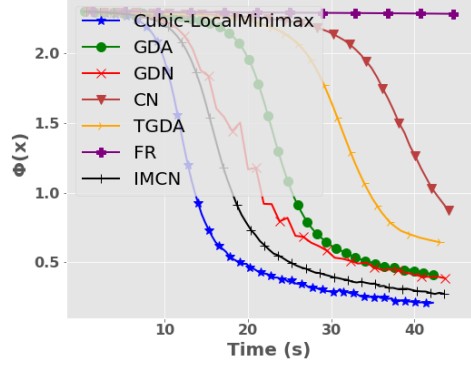

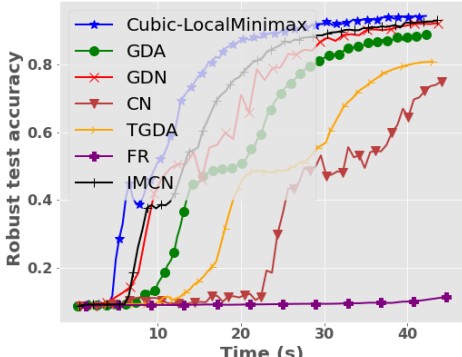

Figure 7: Comparison of $\Phi(x)$ on time complexity for adversarial deep learning.

Figure 8: Comparison of robust test accuracy on time complexity for adversarial deep learning.

the advantage of stochastic Cubic-LocalMinimax algorithm and IMCN in robust test accuracy which is estimated on test samples. It can be seen that the robust models trained by stochastic Cubic-LocalMinimax and IMCN are also more robust in generalization.

Figure 5 and 6 compare the objective function value $\Phi(x)$ and robust test accuracy respectively per oracle call (i.e., an evaluation of gradient or Hessian-vector product) of all the algorithms. Similarly, Figures 7 and 8 compare $\Phi(x)$ and robust test accuracy respectively on time complexity of all the algorithms. It can be seen from these figures that our stochastic Cubic-LocalMinimax algorithm takes significantly fewer oracle calls and less time than all the other algorithms to converge in both objective function and robust test accuracy.

## 8 Conclusion

We developed cubic regularization type algorithms that globally converge to local minimax points in nonconvex-strongly-concave minimax optimization. These algorithms include the basic Cubic-LocalMinimax, Inexact Cubic-LocalMinimax with our proposed GDA-based cubic solver and stochastic Cubic-LocalMinimax for large-scale minimax optimization. By designing and leveraging an intrinsic potential function that monotonically decreases over the iterations, we have obtained the computation or sample complexities required by each algorithm to achieve an $\epsilon$-approximate local-minimax point. Experimental results demonstrate faster convergence of our stochastic Cubic-LocalMinimax than the standard stochastic GDA algorithm. A future direction is to extend the proposed algorithms to bilevel optimization, which is a generalization of minimax optimization.

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

## A  Supporting Lemmas

We first prove the following auxiliary lemma that bounds the spectral norm of the Hessian matrices.

**Lemma A.1.** *Let Assumption 1 hold. Then, for any $x \in \mathbb{R}^m$ and $y \in \mathbb{R}^n$, the Hessian matrices of $f(x,y)$ and $G(x,y) = \left[ \nabla_{11}f - \nabla_{12}f(\nabla_{22}f)^{-1}\nabla_{21}f \right](x,y)$ satisfy the following bounds.*

$$\| [\nabla_{22}f(x,y)]^{-1} \| \leq \mu^{-1}, \tag{26}$$

$$\| \nabla_{12}f(x,y) \| = \| \nabla_{21}f(x,y) \| \leq L_1, \tag{27}$$

$$\| \nabla_{11}f(x,y) \| \leq L_1, \tag{28}$$

$$\| G(x,y) \| \leq L_1(1+\kappa). \tag{29}$$

*The same bounds also hold for $\widehat{\nabla}_{11}f$, $\widehat{\nabla}_{12}f$, $\widehat{\nabla}_{21}f$, $\widehat{\nabla}_{22}f$ and $\widehat{G}$ defined in Section 6 under Assumption 2.*

*Proof.* We first prove eq. (26). Consider any $x \in \mathbb{R}^m$ and $y \in \mathbb{R}^n$. By Assumption 1 we know that $f(x,\cdot)$ is $\mu$-strongy concave, which implies that $-\nabla_{22}f(x,y) \succeq \mu I$. Thus, we further conclude that

$$\| [\nabla_{22}f(x,y)]^{-1} \| = \lambda_{\max}\left([-\nabla_{22}f(x,y)]^{-1}\right) = \left(\lambda_{\min}\left(-\nabla_{22}f(x,y)\right)\right)^{-1} \leq \mu^{-1}.$$

Next, we prove eq. (27). Consider any $x, u \in \mathbb{R}^m$ and $y \in \mathbb{R}^n$, we have

$$
\begin{aligned}
\| \nabla_{21}f(x,y)u \| &= \left\| \frac{\partial}{\partial t}\nabla_2 f(x+tu,y)\Big|_{t=0} \right\| \\
&= \left\| \lim_{t \to 0}\frac{1}{t}\left[\nabla_2 f(x+tu,y) - \nabla_2 f(x,y)\right] \right\| \\
&= \lim_{t \to 0}\frac{1}{|t|}\left\| \nabla_2 f(x+tu,y) - \nabla_2 f(x,y) \right\| \\
&\leq \lim_{t \to 0}\frac{L_1}{|t|}\left\| tu \right\| = L_1\|u\|,
\end{aligned}
\tag{30}
$$

which implies that $\| \nabla_{21}f(x,y) \| \leq L_1$. Since $f$ is twice differentiable and has continuous second-order derivative, we have $\nabla_{12}f(x,y)^\top = \nabla_{21}f(x,y)$, and hence eq. (27) follows. The proof of eq. (28) is similar.

Finally, eq. (29) can be proved as follows using eqs. (26)&(27).

$$\| G(x,y) \| \leq \| \nabla_{11}f(x,y) \| + \| \nabla_{12}f(x,y) \| \| \nabla_{22}f^{-1}(x,y) \| \| \nabla_{21}f(x,y) \| \leq L_1 + L_1\mu^{-1}L_1 = L_1(1+\kappa).$$

The proof is similar for the stochastic minimax optimization problem (Q) in Section 6 under Assumption 2. □

The following lemma restates Lemma 3 of (Wang et al., 2018a), which states the necessary conditions of exact CR solution .

**Lemma A.2.** *The solution $s_{k+1}$ of the cubic regularization problem in Algorithm 1 satisfies the following conditions,*

$$\nabla_1 f(x_t, y_{t+1}) + G(x_t, y_{t+1})s_{t+1} + \frac{1}{2\eta_x}\|s_{t+1}\|s_{t+1} = \mathbf{0}, \tag{31}$$

$$G(x_t, y_{t+1}) + \frac{1}{2\eta_x}\|s_{t+1}\|I \succeq \mathbf{O}, \tag{32}$$

$$\nabla_1 f(x_t, y_{t+1})^\top s_{t+1} + \frac{1}{2}s_{t+1}^\top G(x_t, y_{t+1})s_{t+1} \leq -\frac{1}{4\eta_x}\|s_{t+1}\|^3. \tag{33}$$

**Lemma A.3.** *Suppose the gradient $\nabla_1 f(x_t, y_{t+1})$ and Hessian $G(x_t, y_{t+1})$ involved in the cubic-regularization step in Algorithm 1 satisfy the following bounds for all $t \leq T' - 1$ with $T' = \min\{t \geq 0 : \|s_{t-1}\| \vee \|s_t\| \leq \epsilon'\}$:*

$$\|\nabla\Phi(x_t) - \nabla_1 f(x_t, y_{t+1})\| \leq \beta(\|s_t\|^2 + \epsilon'^2), \tag{34}$$

$$\|\nabla^2\Phi(x_t) - G(x_t, y_{t+1})\| \leq \alpha(\|s_t\| + \epsilon'). \tag{35}$$

*Then, choosing $\eta_x \leq (9L_\Phi + 18\alpha + 28\beta)^{-1}$, the sequence $\{x_t\}_t$ generated by Algorithm 1 satisfies that for all $t \leq T' - 2$,*

$$H_{t+1} - H_t \leq -(L_\Phi + \alpha + \beta)(\|s_{t+1}\|^3 + \|s_t\|^3), \tag{36}$$

*where $H_t = \Phi(x_t) + (L_\Phi + 2\alpha + 3\beta)\|x_t - x_{t-1}\|^3$. The same conclusion holds for Algorithm 5 by replacing $\nabla_1 f(x_t, y_{t+1})$, $G(x_t, y_{t+1})$ with their stochastic estimators $\widehat{\nabla}_1 f(x_t, y_{t+1})$, $\widehat{G}(x_t, y_{t+1})$ respectively.*

This is a key lemma which provides per-iteration decrease of the potential function given that we can access sufficiently accurate gradient $\nabla_1 f(x_t, y_{t+1}) \approx \nabla\Phi(x_t)$ and Hessian matrix $G(x_t, y_{t+1}) \approx \nabla^2\Phi(x_t)$. In this case, the CR objective formed by $\nabla_1 f(x_t, y_{t+1})$ and $G(x_t, y_{t+1})$ is sufficiently close to the target CR objective formed by $\nabla\Phi(x_t)$ and $\nabla^2\Phi(x_t)$, so we can basically follow the standard CR convergence analysis with these additional approximation errors.

*Proof.* By the Lipschitz continuity of the Hessian of $\Phi$, we obtain that

$$\Phi(x_{t+1}) - \Phi(x_t)$$

$$\leq \nabla\Phi(x_t)^\top (x_{t+1} - x_t) + \frac{1}{2}(x_{t+1} - x_t)^\top \nabla^2\Phi(x_t)(x_{t+1} - x_t) + \frac{L_\Phi}{6}\|x_{t+1} - x_t\|^3$$

$$= \nabla_1 f(x_t, y_{t+1})^\top s_{t+1} + \frac{1}{2}s_{t+1}^\top G(x_t, y_{t+1})s_{t+1} + \frac{L_\Phi}{6}\|s_{t+1}\|^3$$

$$\quad + \left(\nabla\Phi(x_t) - \nabla_1 f(x_t, y_{t+1})\right)^\top s_{t+1} + \frac{1}{2}s_{t+1}^\top \left(\nabla^2\Phi(x_t) - G(x_t, y_{t+1})\right)s_{t+1}$$

$$\overset{(i)}{\leq} \left(\frac{L_\Phi}{6} - \frac{1}{4\eta_x}\right)\|s_{t+1}\|^3 + \beta\|s_{t+1}\|(\|s_t\|^2 + \epsilon'^2) + \frac{\alpha}{2}\|s_{t+1}\|^2(\|s_t\| + \epsilon')$$

$$\overset{(ii)}{\leq} \left(\frac{L_\Phi}{6} - \frac{1}{4\eta_x}\right)\|s_{t+1}\|^3 + \left(\frac{\alpha}{2} + \beta\right)(2\|s_{t+1}\|^3 + \|s_t\|^3 + \epsilon'^3)$$

$$\overset{(iii)}{\leq} \left(\frac{L_\Phi}{6} - \frac{1}{4\eta_x}\right)\|s_{t+1}\|^3 + \left(\frac{\alpha}{2} + \beta\right)(3\|s_{t+1}\|^3 + 2\|s_t\|^3)$$

$$\leq -\left(\frac{1}{4\eta_x} - \frac{L_\Phi}{6} - \frac{3\alpha}{2} - 3\beta\right)\|s_{t+1}\|^3 + (\alpha + 2\beta)\|s_t\|^3,$$

$$\overset{(iv)}{\leq} -(2L_\Phi + 3\alpha + 4\beta)\|s_{t+1}\|^3 + (\alpha + 2\beta)\|s_t\|^3$$

where (i) uses eqs. (33), (34) & (35), (ii) uses the inequality that $ab^2 \leq a^3 \vee b^3 \leq a^3 + b^3, \forall a, b \geq 0$, (iii) uses $\epsilon'^3 \leq \|s_t\|^3 \vee \|s_{t+1}\|^3 \leq \|s_t\|^3 + \|s_{t+1}\|^3, \forall 0 \leq t \leq T' - 2$ based on the termination criterion of $T'$, and (iv) uses $\eta_x \leq (9L_\Phi + 18\alpha + 28\beta)^{-1}$. Eq. (36) follows from the above inequality by defining $H_t = \Phi(x_t) + (L_\Phi + 2\alpha + 3\beta)\|x_t - x_{t-1}\|^2$.

Note that the cubic-regularization step in Algorithm 5 simply replaces $\nabla_1 f(x_t, y_{t+1})$, $G(x_t, y_{t+1})$ in Algorithm 1 with their stochastic estimators $\widehat{\nabla}_1 f(x_t, y_{t+1})$, $\widehat{G}(x_t, y_{t+1})$ respectively. Hence, eq. (36) holds for Algorithm 5 after such replacement in eqs. (34) & (35). $\qquad\square$

**Lemma A.4.** *Suppose all the conditions of Lemma A.3 hold. If $T \geq \frac{\Phi(x_0) - \Phi^* + (L_\Phi + 3\alpha + 4\beta)\epsilon'^2}{(L_\Phi + \alpha + \beta)\epsilon'^3}$ in Algorithm 1, then $T' = \min\{t \geq 1 : \|s_{t-1}\| \vee \|s_t\| \leq \epsilon'\} \leq \frac{\Phi(x_0) - \Phi^* + (L_\Phi + 3\alpha + 4\beta)\epsilon'^2}{(L_\Phi + \alpha + \beta)\epsilon'^3} \leq T$. Consequently, the output of Algorithm 1 has the following convergence rate*

$$\|\nabla\Phi(x_{T'})\| \leq \left(\frac{1}{2\eta_x} + L_\Phi + 2\alpha + 2\beta\right)\epsilon'^2, \tag{37}$$

$$\nabla^2 \Phi(x_{T'}) \succeq -\Big(\frac{1}{2\eta_x} + L_\Phi + 2\alpha\Big)\epsilon' I. \tag{38}$$

*The same conclusion holds for Algorithm 5 by replacing* $\nabla_1 f(x_t, y_{t+1})$, $G(x_t, y_{t+1})$ *in the conditions (34) &*
*(35) with their stochastic estimators* $\widehat{\nabla}_1 f(x_t, y_{t+1})$, $\widehat{G}(x_t, y_{t+1})$ *respectively.*

This Lemma provides two nice properties of the last-iterate $T'$, i.e., $T'$ has a finite upper bound $T$, and $x_{T'}$ is
close to the second-order stationary condition $\nabla\Phi(x) = \mathbf{0}$, $\nabla^2\Phi(x) \succ \mathbf{0}$ given by Fact 1. These properties
can be proved respectively by two facts implied by the termination rule $\|s_{t-1}\| \vee \|s_t\| \le \epsilon'$. First, before
termination (i.e., $t \le T'-1$), we have $\|s_{t-1}\| \vee \|s_t\| > \epsilon'$, which results in sufficient potential function decrease
$H_{t+1} - H_t \le -(L_\Phi + \alpha + \beta)\epsilon'^3$ given by eq. (36). The number $T'$ of such sufficient decreases has to be
finitely bounded since $H_t \ge \min_x \Phi(x) > -\infty$. Second, at the last iteration $t = T'$, $\nabla_1 f(x_t, y_{t+1})$ is very
close to $\mathbf{0}$ and $G(x_t, y_{t+1})$ is very close to positive semi-definite based on eqs. (31) & (32). Also, the gradient
$\nabla_1 f(x_t, y_{t+1}) \approx \nabla\Phi(x_t)$ and Hessian $G(x_t, y_{t+1}) \approx \nabla^2\Phi(x_t)$ have $\epsilon'$-level small approximation error based
on eqs. (34) & (35) respectively. Hence, $x_t$ is approximately a second-order stationary point of $\Phi$, i.e., eqs.
(37) & (38) hold.

*Proof.* Suppose $T' \le T$ does not hold, i.e., $\|s_{t-1}\| \vee \|s_t\| > \epsilon', \forall 1 \le t \le T$, which implies that eq. (36) holds
for all $0 \le t \le T-1$ based on Lemma A.3.

On one hand, telescoping eq. (36) over $t = 0, 1, \ldots, T-1$ yield that

$$
\begin{aligned}
H_0 - H_T &\ge (L_\Phi + \alpha + \beta) \sum_{t=0}^{T-1} (\|s_{t+1}\|^3 + \|s_t\|^3) \\
&\ge (L_\Phi + \alpha + \beta) \sum_{t=0}^{T-1} (\|s_t\| \vee \|s_{t+1}\|)^3 \\
&> T(L_\Phi + \alpha + \beta)\epsilon'^3
\end{aligned}
\tag{39}
$$

On the other hand, recalling the definition of $H_t$ in Lemma A.3, we have

$$
\begin{aligned}
H_0 - H_T &= \Phi(x_0) - \Phi(x_T) + (L_\Phi + 2\alpha + 3\beta)(\|s_0\|^2 - \|s_T\|^2) \\
&\overset{(i)}{\le} \Phi(x_0) - \Phi^* + (L_\Phi + 3\alpha + 4\beta)\epsilon'^2,
\end{aligned}
\tag{40}
$$

where (i) uses $\|s_0\| = \epsilon'$ and $\Phi(x_T) \ge \Phi^* = \min_{x \in \mathbb{R}^m} \Phi(x)$. Note that eqs. (39) & (40) contradict.
Therefore, we must have $1 \le T' \le T$ for any $T \ge \frac{\Phi(x_0) - \Phi^* + (L_\Phi + 3\alpha + 4\beta)\epsilon'^2}{(L_\Phi + \alpha + \beta)\epsilon'^3}$, which implies that $T' \le$
$\frac{\Phi(x_0) - \Phi^* + (L_\Phi + 3\alpha + 4\beta)\epsilon'^2}{(L_\Phi + \alpha + \beta)\epsilon'^3} \le T$.

Finally, we conclude that

$$
\begin{aligned}
&\|\nabla\Phi(x_{T'})\| \\
&\overset{(i)}{=} \left\|\nabla\Phi(x_{T'}) - \nabla_1 f(x_{T'-1}, y_{T'}) - G(x_{T'-1}, y_{T'})s_{T'} - \frac{1}{2\eta_x}\|s_{T'}\|s_{T'}\right\| \\
&\le \|\nabla\Phi(x_{T'}) - \nabla\Phi(x_{T'-1}) - \nabla^2\Phi(x_{T'-1})s_{T'}\| + \|\nabla\Phi(x_{T'-1}) - \nabla_1 f(x_{T'-1}, y_{T'})\| \\
&\quad + \|\nabla^2\Phi(x_{T'-1})s_{T'} - G(x_{T'-1}, y_{T'})s_{T'}\| + \frac{1}{2\eta_x}\|s_{T'}\|^2 \\
&\overset{(ii)}{\le} \Big(L_\Phi + \frac{1}{2\eta_x}\Big)\|s_{T'}\|^2 + \beta(\|s_{T'-1}\|^2 + \epsilon'^2) + \alpha(\|s_{T'-1}\| + \epsilon')\|s_{T'}\| \\
&\overset{(iii)}{\le} \Big(\frac{1}{2\eta_x} + L_\Phi + 2\alpha + 2\beta\Big)\epsilon'^2,
\end{aligned}
\tag{41}
$$

where (i) uses eq. (31), (ii) uses eqs. (34) & (35) and the item 4 of Proposition 3.1 that $\nabla^2\Phi$ is $L_\Phi$-Lipschitz,
and (iii) uses $\|s_{T'-1}\| \vee \|s_{T'}\| \le \epsilon'$. Also,

$$\nabla^2\Phi(x_{T'}) \overset{(i)}{\succeq} G(x_{T'-1}, y_{T'}) - \|G(x_{T'-1}, y_{T'}) - \nabla^2\Phi(x_{T'})\|I$$

$$\overset{(ii)}{\succeq} -\frac{1}{2\eta_x}\|s_{T'}\|I - \|G(x_{T'-1}, y_{T'}) - \nabla^2\Phi(x_{T'-1})\|I - \|\nabla^2\Phi(x_{T'}) - \nabla^2\Phi(x_{T'-1})\|I$$

$$\overset{(iii)}{\succeq} -\Big(\frac{1}{2\eta_x} + L_\Phi\Big)\|s_{T'}\|I - \alpha(\|s_{T'-1}\| + \epsilon')I$$

$$\overset{(iv)}{\succeq} -\Big(\frac{1}{2\eta_x} + L_\Phi + 2\alpha\Big)\epsilon'I, \tag{42}$$

where (i) uses Weyl's inequality, (ii) uses eq. (32), (iii) uses eq. (35) and the item 4 of Proposition 3.1 that $\nabla^2\Phi$ is $L_\Phi$-Lipschitz, and (iv) uses $\|s_{T'-1}\| \vee \|s_{T'}\| \leq \epsilon'$. □

For the stochastic minimax optimization problem (Q) in Section 6, we prove the following supporting lemma on the error of the stochastic estimators.

**Lemma A.5.** *Fix any $0 < \epsilon_1 \leq 2L_0$, $0 < \epsilon_2 \leq 4L_1$ and choose the following batch sizes*

$$|B_1| \geq \mathcal{O}\Big(\frac{L_0^2}{\epsilon_1^2}\ln\frac{m}{\delta}\Big), \tag{18}$$

$$|B_{11}|, |B_{12}|, |B_{21}|, |B_{22}| \geq \mathcal{O}\Big(\frac{L_1^2}{\epsilon_2^2}\ln\frac{m+n}{\delta}\Big). \tag{19}$$

*Then, the stochastic approximators satisfy the following error bounds with probability at least $1 - \delta$.*

$$\|\widehat{\nabla}_1 f(x, y) - \nabla_1 f(x, y)\| \leq \epsilon_1, \tag{20}$$

$$\|\widehat{\nabla}_{k\ell}^2 f(x, y) - \nabla_{k\ell}^2 f(x, y)\| \leq \epsilon_2, \ \forall k, \ell \in \{1, 2\}, \tag{21}$$

$$\|\widehat{G}(x, y) - G(x, y)\| \leq (\kappa + 1)^2 \epsilon_2. \tag{22}$$

*Proof.* Based on Lemmas 6 & 8 of (Kohler and Lucchi, 2017), we obtain that with probability at least $1 - \delta$, the following bounds hold. (We replaced $\delta$ in (Kohler and Lucchi, 2017) with $\delta/5$ by applying union bound to the following 5 events.)

$$\|\widehat{\nabla}_1 f(x, y) - \nabla_1 f(x, y)\| \leq 4\sqrt{2}L_0\sqrt{\frac{\ln(10m/\delta) + 1/4}{|B_1|}} \leq \epsilon_1, \tag{43}$$

$$\|\widehat{\nabla}_{k,\ell}^2 f(x, y) - \nabla_{k,\ell}^2 f(x, y)\| \leq 4L_1\sqrt{\frac{\ln\big(10(m \vee n)/\delta\big)}{|B_{k,\ell}|}} \leq \epsilon_2; k, \ell \in \{1, 2\}. \tag{44}$$

Note that here we only consider the cases that $|B_1|, |B_{k,\ell}| < N$ for all $k, \ell \in \{1, 2\}$. Otherwise, $|B_1| = N$ yields $\|\widehat{\nabla}_1 f(x, y) - \nabla_1 f(x, y)\| = 0 < \epsilon_1$ and $|B_{k,\ell}| = N$ yields $\|\widehat{\nabla}_{k,\ell}^2 f(x, y) - \nabla_{k,\ell}^2 f(x, y)\| = 0 < \epsilon_2$. Hence, in both cases, the above high probability bounds hold, which further implies eq. (22) following the argument below.

$$\|\widehat{G}(x, y) - G(x, y)\|$$
$$\leq \|\widehat{\nabla}_{11} f(x, y) - \nabla_{11} f(x, y)\| + \|(\widehat{\nabla}_{12} f(\widehat{\nabla}_{22} f)^{-1}\widehat{\nabla}_{21} f)(x, y) - (\nabla_{12} f(\nabla_{22} f)^{-1}\nabla_{21} f)(x, y)\|$$
$$\leq \epsilon_2 + \|(\widehat{\nabla}_{12} f - \nabla_{12} f)\big((\widehat{\nabla}_{22} f)^{-1}\widehat{\nabla}_{21} f\big)(x, y)\|$$
$$\quad + \|\nabla_{12} f\big((\widehat{\nabla}_{22} f)^{-1} - (\nabla_{22} f)^{-1}\big)\widehat{\nabla}_{21} f(x, y)\| + \|\nabla_{12} f(\nabla_{22} f)^{-1}(\widehat{\nabla}_{21} f - \nabla_{21} f)(x, y)\|$$
$$\overset{(i)}{\leq} \epsilon_2 + \epsilon_2\mu^{-1}L_1 + L_1^2\|\nabla_{22} f(x, y)^{-1}\|\|(\nabla_{22} f - \widehat{\nabla}_{22} f)(x, y)\|\|\widehat{\nabla}_{22} f(x, y)^{-1}\| + L_1\mu^{-1}\epsilon_2$$
$$\overset{(ii)}{\leq} \epsilon_2 + 2\kappa\epsilon_2 + L_1^2\mu^{-2}\epsilon_2 \leq (\kappa + 1)^2\epsilon_2. \tag{45}$$

where (i) and (ii) use Lemma A.1. □

Regarding the high-probability convergence rate of the inner stochastic gradient ascent (SGA) in Algorithm 5, the following result is a direct application of Theorem 3.1 in (Harvey et al., 2019).

# B  Proof of Proposition 3.1

**Proposition B.1.** *Let Assumption 1 hold. Then, the following statements hold.*

1. *The mapping $y^*(x) := \arg\max_{y \in \mathbb{R}^n} f(x, y)$ is unique and $\kappa$-Lipschitz continuous for every fixed $x$ (Lin et al., 2020);*

2. *$\Phi(x)$ is $L_1(1 + \kappa)$-smooth and $\nabla\Phi(x) = \nabla_1 f(x, y^*(x))$ (Lin et al., 2020);*

3. *Define mapping $G(x, y) = \left[\nabla_{11} f - \nabla_{12} f (\nabla_{22} f)^{-1} \nabla_{21} f\right](x, y)$. Then, $G$ is a Lipschitz continuous mapping with Lipschitz constant $L_G = L_2(1 + \kappa)^2$;*

4. *The Hessian of $\Phi$ satisfies $\nabla^2 \Phi(x) = G(x, y^*(x))$, and it is Lipschitz continuous with Lipschitz constant $L_\Phi = L_G(1 + \kappa) = L_2(1 + \kappa)^3$.*

*Proof.* The items 1 & 2 have been proved in (Lin et al., 2020).

We first prove the item 3. Consider any $x, x' \in \mathbb{R}^m$ and $y, y' \in \mathbb{R}^n$. For convenience we denote $z = (x, y)$ and $z' = (x', y')$. Then, by Assumption 1 and using the bounds of Lemma A.1, we have that

$$
\begin{aligned}
&\|G(x', y') - G(x, y)\| \\
&\leq \|\nabla_{11} f(x', y') - \nabla_{11} f(x, y)\| + \|\nabla_{12} f(x', y') - \nabla_{12} f(x, y)\| \|[\nabla_{22} f(x', y')]^{-1}\| \|\nabla_{21} f(x', y')\| \\
&\quad + \|\nabla_{12} f(x, y)\| \|[\nabla_{22} f(x', y')]^{-1} - [\nabla_{22} f(x, y)]^{-1}\| \|\nabla_{21} f(x', y')\| \\
&\quad + \|\nabla_{12} f(x, y)\| \|[\nabla_{22} f(x, y)^{-1}]\| \|\nabla_{21} f(x', y') - \nabla_{21} f(x, y)\| \\
&\leq L_2 \|z' - z\| + (L_2 \|z' - z\|)\mu^{-1} L_1 \\
&\quad + L_1^2 \|[\nabla_{22} f(x', y')]^{-1}\| \|\nabla_{22} f(x, y) - \nabla_{22} f(x', y')\| \|[\nabla_{22} f(x, y)]^{-1}\| + L_1 \mu^{-1}(L_2 \|z' - z\|) \\
&\leq L_2(1 + 2\kappa)\|z' - z\| + L_1^2 \mu^{-1}(L_2 \|z' - z\|)\mu^{-1} \\
&\leq L_2(1 + \kappa)^2 \|z' - z\|.
\end{aligned}
$$

Next, we prove the item 4. Consider any fixed $x \in \mathbb{R}^m$, we know that $f(x, \cdot)$ achieves its maximum at $y^*(x)$, where the gradient vanishes, i.e., $\nabla_2 f(x, y^*(x)) = \mathbf{0}$. Thus, we further obtain that

$$
\mathbf{0} = \nabla_x \nabla_2 f(x, y^*(x)) = \nabla_{21} f(x, y^*(x)) + \nabla_{22} f(x, y^*(x)) \nabla y^*(x),
$$

which implies that

$$
\nabla y^*(x) = -[\nabla_{22} f(x, y^*(x))]^{-1} \nabla_{21} f(x, y^*(x)). \tag{46}
$$

With the above equation, we take derivative of $\nabla\Phi(x) = \nabla_1 f(x, y^*(x))$ and obtain that

$$
\begin{aligned}
\nabla^2 \Phi(x) &= \nabla_{11} f(x, y^*(x)) + \nabla_{12} f(x, y^*(x)) \nabla y^*(x) \\
&= \nabla_{11} f(x, y^*(x)) - \nabla_{12} f(x, y^*(x))[\nabla_{22} f(x, y^*(x))]^{-1} \nabla_{21} f(x, y^*(x)) \\
&= G(x, y^*(x)).
\end{aligned} \tag{47}
$$

Moreover, we have that

$$
\begin{aligned}
\|\nabla^2 \Phi(x') - \nabla^2 \Phi(x)\| &= \|G(x', y^*(x')) - G(x, y^*(x))\| \\
&\leq L_G\big[\|x' - x\| + \|y^*(x') - y^*(x)\|\big] \\
&\leq L_G(1 + \kappa)\|x' - x\|,
\end{aligned} \tag{48}
$$

where the last step uses the item 1 of Proposition 3.1. This proves the item 4.

$\square$

## C   Proof of Proposition 4.1

**Proposition C.1** (Potential function)**.** *Let Assumption 1 hold.  For any* $\alpha, \beta > 0$*, choose* $\epsilon' \leq \frac{\alpha L_1}{\beta L_G}$*,* $\eta_x \leq (9L_\Phi + 18\alpha + 28\beta)^{-1}$ *and* $\eta_y = \frac{2}{L_1 + \mu}$*.  Define the potential function* $H_t := \Phi(x_t) + (L_\Phi + 2\alpha + 3\beta)\|s_t\|^3$*. Then, when* $N_t \geq \mathcal{O}\big(\kappa \ln \frac{L_1 \alpha \|s_{t-1}\| + L_1(\alpha + L_2\kappa)\|s_t\|}{L_G \beta \epsilon'^2}\big)$*, the output of Cubic-LocalMinimax satisfies the following potential function decrease property for all* $t \in \mathbb{N}$*.*

$$H_{t+1} - H_t \leq -(L_\Phi + \alpha + \beta)\big(\|s_{t+1}\|^3 + \|s_t\|^3\big). \tag{2}$$

Note that Lemma A.3 already proves eq.  (2) under the condition that the approximate gradient $\nabla_1 f(x_t, y_{t+1}) \approx \nabla\Phi(x_t)$ and Hessian $G(x_t, y_{t+1}) \approx \nabla^2\Phi(x_t)$ are sufficiently accurate, as given by eqs. (34) & (35) respectively.  Hence, it remains to prove eqs. (34) & (35) by showing that the gradient ascent steps on the $\mu$-strongly convex function $f(x_t, y)$ yields sufficiently small error $\|y_{t+1} - y^*(x_t)\|$ as shown in eq. (50) below.

*Proof.*  The required number of inner gradient ascent steps is shown below

$$N_0 \geq \kappa \ln \big(L_1\|y_0 - y^*(x_0)\|/(2\beta\epsilon'^2)\big)$$
$$N_t \geq \kappa \ln \Big(\frac{2L_1 \alpha \|s_{t-1}\| + L_1(\alpha + L_G\kappa)\|s_t\|}{L_G \beta \epsilon'^2}\Big)$$
$$= \mathcal{O}\Big(\kappa \ln \frac{L_1 \alpha \|s_{t-1}\| + L_1(\alpha + L_2\kappa)\|s_t\|}{L_G \beta \epsilon'^2}\Big); 1 \leq t \leq T'. \tag{49}$$

To prove eq. (2), we first prove by induction that for any $t \geq 0$.

$$\|y_{t+1} - y^*(x_t)\| \leq \frac{\alpha(\|s_t\| + \epsilon')}{L_G} \wedge \frac{\beta(\|s_t\|^2 + \epsilon'^2)}{L_1}; 0 \leq t \leq T' - 1. \tag{50}$$

Note that $y_{t+1}$ is obtained by applying $N_t$ gradient ascent steps starting from $y_t$.  Hence, by the convergence rate of gradient ascent algorithm under strong concavity, we conclude that with learning rate $\eta_y = \frac{2}{L+\mu}$,

$$\|y_{t+1} - y^*(x_t)\| \leq (1 - \kappa^{-1})^{N_t}\|y_t - y^*(x_t)\|. \tag{51}$$

When $t = 0$, eq. (51) implies that

$$\|y_1 - y^*(x_0)\| \leq \|y_0 - y^*(x_0)\| \exp \big(N_0 \ln(1 - \kappa^{-1})\big)$$
$$\overset{(i)}{\leq} \|y_0 - y^*(x_0)\| \exp \Big(-\ln \Big(\frac{L_1\|y_0 - y^*(x_0)\|}{2\beta\epsilon'^2}\Big)\Big)$$
$$= \frac{2\beta\epsilon'^2}{L_1} \overset{(ii)}{\leq} \frac{\alpha(\|s_0\| + \epsilon')}{L_G} \wedge \frac{\beta(\|s_0\|^2 + \epsilon'^2)}{L_1} \tag{52}$$

where (i) uses eq. (49) and $\ln(1 - x) \leq -x < 0$ for $x = \kappa^{-1} \in (0, 1)$ and (ii) uses $\|s_0\| = \epsilon' \leq \frac{\alpha L_1}{\beta L_G}$.  Hence, eq. (50) holds when $t = 0$.

If eq. (50) holds for $t = k - 1 \in [0, T' - 2]$, then

$$\|y_{k+1} - y^*(x_k)\|$$
$$\leq (1 - \kappa^{-1})^{N_k}\|y_k - y^*(x_{k-1})\| + (1 - \kappa^{-1})^{N_k}\|y^*(x_{k-1}) - y^*(x_k)\|$$
$$\overset{(i)}{\leq} \exp \big(N_k \ln(1 - \kappa^{-1})\big)\Big(\Big(\frac{\alpha(\|s_{k-1}\| + \epsilon')}{L_G} \wedge \frac{\beta(\|s_{k-1}\|^2 + \epsilon'^2)}{L_1}\Big) + \kappa\|s_k\|\Big)$$
$$\overset{(ii)}{\leq} \exp \Big(-\ln \Big(\frac{2L_1\alpha\|s_{k-1}\| + L_1(\alpha + L_G\kappa)\|s_k\|}{L_G \beta \epsilon'^2}\Big)\Big)\Big(\frac{\alpha(\|s_{k-1}\| + \epsilon')}{L_G} + \kappa\|s_k\|\Big)$$
$$\overset{(iii)}{\leq} \frac{L_G \beta \epsilon'^2}{2L_1\alpha\|s_{k-1}\| + L_1(\alpha + L_G\kappa)\|s_k\|} \frac{\alpha(2\|s_{k-1}\| + \|s_k\|) + L_G\kappa\|s_k\|}{L_G}$$

$$= \frac{\beta\epsilon'^2}{L_1} \overset{(iv)}{=} \frac{\alpha\epsilon'}{L_G} \wedge \frac{\beta\epsilon'^2}{L_1} \leq \frac{\alpha(\|s_k\| + \epsilon')}{L_G} \wedge \frac{\beta(\|s_k\|^2 + \epsilon'^2)}{L_1}$$

where (i) uses eq. (50) for $t = k-1$ and the fact that $y^*$ is $\kappa$-Lipschitz mapping (see (Lin et al., 2020; Chen et al., 2021) for the proof), (ii) uses $\ln(1 - \kappa^{-1}) \leq -\kappa^{-1}$ and eq. (49), (iii) uses $\epsilon' \leq \|s_{k-1}\| \vee \|s_k\| \leq \|s_{k-1}\| + \|s_k\|$ for $k \leq T' - 1$, and (iv) uses the condition that $\epsilon' \leq \frac{\alpha L_1}{\beta L_G}$. This proves eq. (50) holds for $t = k$ and thus for all $t \in [0, T' - 1]$, which further implies eqs. (34) & (35). Hence, by Lemma A.3, we prove that eq. (2) holds for all $0 \leq t \leq T' - 1$. $\qquad\square$

## D  Proof of Theorem 1

**Theorem 1** (Convergence and complexity of Cubic-LocalMinimax). *Let the conditions of Proposition 4.1 hold with $\alpha = \beta = L_\Phi$. For any $0 < \epsilon \leq \frac{L_1\sqrt{33L_\Phi}}{L_G}$, choose $\epsilon' = \frac{\epsilon}{\sqrt{33L_\Phi}}$ and $T \geq \frac{\Phi(x_0) - \Phi^* + 8L_\Phi\epsilon'^2}{3L_\Phi\epsilon'^3}$. Then, the output of Cubic-LocalMinimax satisfies*

$$\mu(x_{T'}) \leq \epsilon. \tag{5}$$

*Consequently, the total number of required cubic iterations satisfies $T' \leq \mathcal{O}(\sqrt{L_2}\kappa^{1.5}\epsilon^{-3})$, and the total number of required gradient ascent iterations satisfies $\sum_{t=0}^{T'-1} N_t \leq \widetilde{\mathcal{O}}(\sqrt{L_2}\kappa^{2.5}\epsilon^{-3})$.*

Eq. (5) is directly implied by the second-order stationary conditions (37) & (38). Hence, it remains to compute the iteration numbers $T'$ and $\sum_{t=0}^{T'-1} N_t$ by substituting the hyperparameters.

*Proof.* Substituting $\alpha = \beta = L_\Phi = L_2(1 + \kappa)^3$ and $\epsilon' = \frac{\epsilon}{\sqrt{33L_\Phi}}$ into Lemma A.4 yields that when $T \geq \sqrt{33L_\Phi}\frac{33(\Phi(x_0) - \Phi^*) + 8\epsilon^2}{3\epsilon^3}$ and $\eta_x = (55L_\Phi)^{-1}$, we have $T' \leq \sqrt{33L_\Phi}\frac{33(\Phi(x_0) - \Phi^*) + 8\epsilon^2}{3\epsilon^3} = \mathcal{O}(\sqrt{L_2}\kappa^{1.5}\epsilon^{-3}) \leq T$, and moreover,

$$\|\nabla\Phi(x_{T'})\| \leq \left(\frac{1}{2\eta_x} + L_\Phi + 2\alpha + 2\beta\right)\epsilon'^2 \leq \epsilon^2,$$

$$\lambda_{\min}\left(\nabla^2\Phi(x_{T'})\right) \geq -\left(\frac{1}{2\eta_x} + L_\Phi + 2\alpha\right)\epsilon' \geq -\sqrt{33L_\Phi}\epsilon.$$

This proves eq. (5).

Note that the number of gradient ascent iterations $N_t$ should satisfy eq. (49). Substituting $\alpha = \beta = L_\Phi = L_2(1 + \kappa)^3$ and $\epsilon' = \frac{\epsilon}{\sqrt{33L_\Phi}}$ into eq. (49) yields that

$$N_0 \geq \kappa\ln\left(L_1\|y_0 - y^*(x_0)\|/(2\beta\epsilon'^2)\right) = \kappa\ln\left(33L_1\|y_0 - y^*(x_0)\|/(2\epsilon^2)\right)$$

$$N_t \geq \kappa\ln\left(\frac{2L_1\alpha\|s_{t-1}\| + L_1(\alpha + L_G\kappa)\|s_t\|}{L_G\beta\epsilon'^2}\right)$$

$$\overset{(i)}{=} \kappa\ln\left(\frac{66L_1(1 + \kappa)\|s_{t-1}\| + 33L_1(2 + \kappa)\|s_t\|}{\epsilon^2}\right); 1 \leq t \leq T'$$

where (i) uses $L_G = L_2(1 + \kappa)^2$ and $L_\Phi = L_2(1 + \kappa)^3$ in Proposition 3.1.

When we select $N_t$ such that the above holds with equality, then the total number of gradient ascent iterations is upper bounded as follows

$$\sum_{t=0}^{T'-1} N_t = \kappa\ln\left(33L_1\|y_0 - y^*(x_0)\|/(2\epsilon^2)\right) + \kappa\sum_{t=0}^{T'-1}\ln\left(\frac{66L_1(1 + \kappa)\|s_{t-1}\| + 33L_1(2 + \kappa)\|s_t\|}{\epsilon^2}\right)$$

$$\leq \kappa\ln\left(33L_1\|y_0 - y^*(x_0)\|/(2\epsilon^2)\right) + \kappa\sum_{t=0}^{T'-1}\ln\left(\|s_{t-1}\| + \|s_t\|\right) + \kappa T'\ln\left(66L_1\epsilon^{-2}(1 + \kappa)\right)$$

$$\overset{(i)}{\leq} \kappa \ln\left(33L_1\|y_0 - y^*(x_0)\|/(2\epsilon^2)\right) + \frac{\kappa T'}{3}\frac{1}{T'}\sum_{t=0}^{T'-1}\ln\left(4\|s_{t-1}\|^3 + 4\|s_t\|^3\right)$$
$$\quad + \kappa T' \ln\left(66L_1\epsilon^{-2}(1+\kappa)\right)$$

$$\overset{(ii)}{\leq} \kappa \ln\left(33L_1\|y_0 - y^*(x_0)\|/(2\epsilon^2)\right) + \frac{\kappa T'}{3}\ln\left(\frac{4}{T'}\sum_{t=0}^{T'-1}\left(\|s_{t-1}\|^3 + \|s_t\|^3\right)\right)$$
$$\quad + \kappa T' \ln\left(66L_1\epsilon^{-2}(1+\kappa)\right)$$

$$\overset{(iii)}{\leq} \kappa \ln\left(33L_1\|y_0 - y^*(x_0)\|/(2\epsilon^2)\right) + \frac{\kappa T'}{3}\ln\left(\frac{4(H_0 - H^*)}{3L_\Phi T'}\right)$$
$$\quad + \kappa T' \ln\left(66L_1\epsilon^{-2}(1+\kappa)\right)$$
$$= \widetilde{\mathcal{O}}(\kappa T') \leq \widetilde{\mathcal{O}}\left(\sqrt{L_2}\kappa^{2.5}\epsilon^{-3}\right)$$

where (i) uses $(a+b)^3 \leq 4(a^3 + b^3)$ for any $a, b \geq 0$, (ii) applies Jensen's inequality to the concave function $\ln(\cdot)$, and (iii) telescopes eq. (2) over $t = 0, 1, \ldots, T' - 1$. $\qquad\square$

## E  Proof of Theorem 3

**Theorem 3.** *Let Assumption 2 hold. For all $t, k$, assume that $\|\nabla_2 f(x_t, \widetilde{y}_k)\| \leq L_0$ and $\|\widehat{\nabla}_2 f(x_t, \widetilde{y}_k) - \nabla_2 f(x_t, \widetilde{y}_k)\| \leq 1$ almost surely. The inner stochastic gradient ascent steps in Algorithm 5 converge at the following rate with probability at least $1 - \delta$.*

$$\|y_{t+1} - y^*(x_t)\| \leq \mathcal{O}\left(\sqrt{\frac{L_0 \ln(1/\delta) + L_0^2}{\mu^2 N_t}}\right).$$

*Proof.* Based on Theorem 3.1 of (Harvey et al., 2019), the inner SGA steps in Algorithm 5 has the following convergence rate with probability at least $1 - \delta$.

$$f(x_t, y^*(x_t)) - f(x_t, y_{t+1}) \leq \mathcal{O}\left(\frac{L_0 \ln(1/\delta) + L_0^2}{\mu N_t}\right), \tag{53}$$

which by $\mu$-strong concavity of $f(x_t, \cdot)$ proves the following convergence rate.

$$\|y_{t+1} - y^*(x_t)\| \leq \sqrt{2\left(f(x_t, y^*(x_t)) - f(x_t, y_{t+1})\right)/\mu} \leq \mathcal{O}\left(\sqrt{\frac{L_0 \ln(1/\delta) + L_0^2}{\mu^2 N_t}}\right).$$

$\qquad\square$

## F  Proof of Theorem 4

**Theorem 4** (Convergence and sample complexity). *Let Assumption 2 and Theorem 3 hold. For any $0 < \epsilon \leq \frac{L_1\sqrt{33L_\Phi}}{L_G}$, choose $\epsilon' = \frac{\epsilon}{\sqrt{33L_\Phi}}$, $\eta_x \leq \frac{1}{55L_\Phi}$, $T \geq \frac{\Phi(x_0) - \Phi^* + 8L_\Phi\epsilon'^2}{3L_\Phi\epsilon'^3}$ and $N_t \geq \mathcal{O}\left(\frac{L_0 \ln(1/\delta) + L_0^2}{\kappa^{-2}(L_\Phi^2\|s_t\|^4 + \epsilon^4) \wedge L_1^2(\|s_t\|^2 + \epsilon^2/L_\Phi)}\right)$. Moreover, in iteration $t$, choose the batch sizes according to eqs. (18) and (19) with the inexactness given by*

$$\epsilon_1(t) = \frac{L_\Phi}{2}\left(\|s_t\|^2 + \frac{\epsilon^2}{33L_\Phi}\right) \wedge 2L_0, \quad \epsilon_2(t) = \frac{L_\Phi}{2(\kappa+1)^2}\left(\|s_t\| + \frac{\epsilon}{\sqrt{33L_\Phi}}\right) \wedge 4L_1.$$

*Then, the output of Stochastic Cubic-LocalMinimax satisfies*

$$\mu(x_{T'}) \leq \epsilon. \tag{23}$$

*Consequently, the total number of cubic iterations satisfies $T' \leq \mathcal{O}(\sqrt{L_2}\kappa^{1.5}\epsilon^{-3})$, the total number of queried gradient samples satisfies $\sum_{t'=0}^{T'}\left(N_t + |B_1(t)|\right) \leq \mathcal{O}\left(\frac{L_0^2\kappa^{3.5}\sqrt{L_2}}{\epsilon^7}\ln\frac{m}{\delta}\right)$, and the total number of queried Hessian samples satisfies $\sum_{t=0}^{T'-1}\sum_{k=1}^{2}\sum_{\ell=1}^{2}|B_{k,\ell}(t)| \leq \mathcal{O}\left(\frac{L_1^2\kappa^{2.5}}{\sqrt{L_2}\epsilon^5}\ln\frac{m+n}{\delta}\right)$.*

Compared with deterministic optimization, the stochastic gradient error $\|\widehat{\nabla}_1 f(x_t, y_{t+1}) - \nabla\Phi(x_t)\|$ results from not only the gradient ascent induced error $\|\nabla_1 f(x_t, y_{t+1}) - \nabla\Phi(x_t)\|$, but also stochastic approximation error $\|\widehat{\nabla}_1 f(x_t, y_t) - \nabla_1 f(x_t, y_t)\|$. The stochastic Hessian error $\|\widehat{G}(x_t, y_{t+1}) - \nabla^2\Phi(x_t)\|$ is similar. Hence, the main idea is to show that after incorporating the stochastic approximation errors from Lemma 6.1, the stochastic gradient and Hessian errors still satisfy the conditions (34) & (35), which by Lemma A.4 shows the desired convergence result.

*Proof.* In Lemma 6.1, replace $x, y$ with $x_t, y_t$, $B_1$ with $B_1(t)$, and $B_{k,\ell}$ with $B_{k,\ell}(t)$ for any $k, \ell \in \{1, 2\}$, and substitute the following hyperparameters.

$$\epsilon_1(t) = \frac{L_\Phi}{2}\left(\|s_t\|^2 + \frac{\epsilon^2}{33L_\Phi}\right) \wedge 2L_0 = \mathcal{O}(L_\Phi\|s_t\|^2 + \epsilon^2),$$

$$\epsilon_2(t) = \frac{L_\Phi}{2(\kappa+1)^2}\left(\|s_t\| + \frac{\epsilon}{\sqrt{33L_\Phi}}\right) \wedge 4L_1 = \mathcal{O}\left(\kappa^{-2}(L_\Phi\|s_t\| + \sqrt{L_\Phi}\epsilon)\right).$$

Then, we obtain that using the following batchsizes

$$
\begin{cases}
|B_1(t)| \geq \mathcal{O}\left(\dfrac{L_0^2}{(L_\Phi^2\|s_t\|^4 + \epsilon^4)}\ln\dfrac{m}{\delta}\right) \\[3mm]
|B_{k,\ell}(t)| \geq \mathcal{O}\left(\dfrac{L_1^2\kappa^4}{L_\Phi(L_\Phi\|s_t\|^2 + \epsilon^2)}\ln\dfrac{m+n}{\delta}\right)
\end{cases},
\tag{54}
$$

the stochastic approximators satisfy the following error bounds with probability at least $1 - \delta$.

$$\|\widehat{\nabla}_1 f(x_t, y_t) - \nabla_1 f(x_t, y_t)\| \leq \epsilon_1(t) \leq \frac{L_\Phi}{2}\left(\|s_t\|^2 + \frac{\epsilon^2}{33L_\Phi}\right), \tag{55}$$

$$\|\widehat{G}(x_t, y_t) - G(x_t, y_t)\| \leq \epsilon_2(t) \leq \frac{L_\Phi}{2}\left(\|s_t\| + \frac{\epsilon}{\sqrt{33L_\Phi}}\right). \tag{56}$$

Based on Theorem 3, using the following number of stochastic gradient ascent steps

$$
\begin{aligned}
N_t &\geq \mathcal{O}\left(\frac{L_0\ln(1/\delta) + L_0^2}{\mu^2 L_\Phi^2\left(([\|s_t\|^2 + \epsilon^2/(33L_\Phi)]/L_1) \wedge ([\|s_t\| + \epsilon/\sqrt{33L_\Phi}]/L_G)\right)^2}\right) \\
&\overset{(i)}{=} \mathcal{O}\left(\frac{L_0\ln(1/\delta) + L_0^2}{\left(([L_\Phi\|s_t\|^2 + \epsilon^2]/\kappa) \wedge L_1(\|s_t\| + \epsilon/\sqrt{L_\Phi})\right)^2}\right) \\
&= \mathcal{O}\left(\frac{L_0\ln(1/\delta) + L_0^2}{\kappa^{-2}(L_\Phi^2\|s_t\|^4 + \epsilon^4) \wedge L_1^2(\|s_t\|^2 + \epsilon^2/L_\Phi)}\right),
\end{aligned}
\tag{57}
$$

where (i) uses $\kappa = L_1/\mu$ and $L_\Phi = L_G(1 + \kappa)$ (Proposition 3.1). we have

$$
\begin{aligned}
\|y_{t+1} - y^*(x_t)\| &\leq \mathcal{O}\left(\sqrt{\frac{L_0\ln(1/\delta) + L_0^2}{\mu^2 N_t}}\right) \\
&\leq \frac{L_\Phi}{2}\min\left(\frac{1}{L_1}\left(\|s_t\|^2 + \frac{\epsilon^2}{33L_\Phi}\right), \frac{1}{L_G}\left(\|s_t\| + \frac{\epsilon}{\sqrt{33L_\Phi}}\right)\right).
\end{aligned}
\tag{58}
$$

Therefore,

$$
\begin{aligned}
&\|\nabla\Phi(x_t) - \widehat{\nabla}_1 f(x_t, y_{t+1})\| \\
&\overset{(i)}{\leq} \|\nabla_1 f(x_t, y^*(x_t)) - \nabla_1 f(x_t, y_{t+1})\| + \|\widehat{\nabla}_1 f(x_t, y_{t+1}) - \nabla_1 f(x_t, y_{t+1})\| \\
&\overset{(ii)}{\leq} L_1\|y_{t+1} - y^*(x_t)\| + \frac{L_\Phi}{2}\left(\|s_t\|^2 + \frac{\epsilon^2}{33L_\Phi}\right) \\
&\overset{(iii)}{\leq} L_\Phi\left(\|s_t\|^2 + \frac{\epsilon^2}{33L_\Phi}\right),
\end{aligned}
\tag{59}
$$

where (i) uses $\nabla\Phi(x) = \nabla_1 f(x, y^*(x))$ in Proposition 3.1, (ii) uses eq. (55) and Assumption 1, and (iii) uses eq. (58).

$$
\begin{aligned}
&\|\nabla^2\Phi(x_t) - \widehat{G}(x_t, y_{t+1})\| \\
&\overset{(i)}{\leq} \|G(x_t, y^*(x_t)) - G(x_t, y_{t+1})\| + \|\widehat{G}(x_t, y_{t+1}) - G(x_t, y_{t+1})\| \\
&\overset{(ii)}{\leq} L_G\|y_{t+1} - y^*(x_t)\| + \frac{L_\Phi}{2}\Big(\|s_t\| + \frac{\epsilon}{\sqrt{33L_\Phi}}\Big) \\
&\overset{(iii)}{\leq} L_\Phi\Big(\|s_t\| + \frac{\epsilon}{\sqrt{33L_\Phi}}\Big),
\end{aligned} \tag{60}
$$

where (i) uses $\nabla^2\Phi(x) = G(x, y^*(x))$ in Proposition 3.1, (ii) uses eq. (56) and the item 3 of Proposition 3.1, and (iii) uses eq. (58). Eqs. (59) & (60) imply that the conditions (34) & (35) hold with $\alpha = \beta = L_\Phi = L_2(1+\kappa)^3$ and $\epsilon' = \frac{\epsilon}{\sqrt{33L_\Phi}}$. In Lemma A.4, by substituting these values of $\alpha$, $\beta$, $\epsilon'$ and $\eta_x = (55L_\Phi)^{-1}$, we obtain that when $T \geq \frac{\Phi(x_0) - \Phi^* + 8L_\Phi \epsilon'^2}{3L_\Phi \epsilon'^3}$, we have $T' \leq \sqrt{33L_\Phi}\frac{33(\Phi(x_0) - \Phi^*) + 8\epsilon^2}{3\epsilon^3} = \mathcal{O}(\sqrt{L_2}\kappa^{1.5}\epsilon^{-3}) \leq T$, and moreover,

$$
\|\nabla\Phi(x_{T'})\| \leq \Big(\frac{1}{2\eta_x} + L_\Phi + 2\alpha + 2\beta\Big)\epsilon'^2 \leq \epsilon^2,
$$

$$
\lambda_{\min}\big(\nabla^2\Phi(x_{T'})\big) \geq -\Big(\frac{1}{2\eta_x} + L_\Phi + 2\alpha\Big)\epsilon' \overset{(i)}{\geq} -\sqrt{33L_\Phi}\epsilon,
$$

This proves that eq. (23) holds with probability at least $1 - \delta$.

Choosing both $N_t$ in eq. (57) and the batchsizes in eq. (54) with equality, the number of gradient computations has the following upper bound.

$$
\begin{aligned}
&\sum_{t=0}^{T'-1}\big(N_t + |B_1(t)|\big) \\
&= \sum_{t=0}^{T'-1}\mathcal{O}\Big(\frac{L_0\ln(1/\delta) + L_0^2}{\kappa^{-2}(L_\Phi^2\|s_t\|^4 + \epsilon^4) \wedge L_1^2(\|s_t\|^2 + \epsilon^2/L_\Phi)} + \frac{L_0^2}{L_\Phi^2\|s_t\|^4 + \epsilon^4}\ln\frac{m}{\delta}\Big) \\
&\leq \sum_{t=0}^{T'-1}\mathcal{O}\Big(\frac{L_0\ln(1/\delta) + L_0^2}{\epsilon^4\kappa^{-2} \wedge L_1^2\epsilon^2/L_\Phi} + \frac{L_0^2}{\epsilon^4}\ln\frac{m}{\delta}\Big) \\
&\overset{(i)}{\leq} T'\mathcal{O}\Big(\frac{L_0\kappa^2}{\epsilon^4}\big(\ln(1/\delta) + L_0\big) + \frac{L_0^2}{\epsilon^4}\ln\frac{m}{\delta}\Big) \\
&\overset{(ii)}{\leq} \mathcal{O}\big(\sqrt{L_2}\kappa^{1.5}\epsilon^{-3}\big)\mathcal{O}\Big(\frac{L_0^2\kappa^2}{\epsilon^4}\ln\frac{m}{\delta}\Big) \\
&\leq \mathcal{O}\Big(\frac{L_0^2\kappa^{3.5}\sqrt{L_2}}{\epsilon^7}\ln\frac{m}{\delta}\Big),
\end{aligned} \tag{61}
$$

where (i) uses $\epsilon' = \frac{\epsilon}{\sqrt{33L_\Phi}} \leq \frac{L_1}{L_G} = \frac{L_1(1+\kappa)}{L_\Phi}$ which implies that $\epsilon^4\kappa^{-2} \leq \mathcal{O}(L_1^2\epsilon^2/L_\Phi)$, (ii) uses $T' \leq \mathcal{O}(\sqrt{L_2}\kappa^{1.5}\epsilon^{-3})$ we proved above. The number of Hessian computations has the following upper bound.

$$
\begin{aligned}
&\sum_{t=0}^{T'-1}\sum_{k=1}^{2}\sum_{\ell=1}^{2}|B_{k,\ell}(t)| \\
&= \sum_{t=0}^{T'-1}\mathcal{O}\Big(\frac{L_1^2\kappa^4}{L_\Phi(L_\Phi\|s_t\|^2 + \epsilon^2)}\ln\frac{m+n}{\delta}\Big) \\
&\leq \sum_{t=0}^{T'-1}\mathcal{O}\Big(\frac{L_1^2\kappa^4}{L_\Phi\epsilon^2}\ln\frac{m+n}{\delta}\Big)
\end{aligned}
$$

$$\overset{(i)}{\leq} T'\mathcal{O}\Big(\frac{L_1^2\kappa}{L_2\epsilon^2}\ln\frac{m+n}{\delta}\Big)$$

$$\overset{(ii)}{\leq} \mathcal{O}\Big(\frac{L_1^2\kappa^{2.5}}{\sqrt{L_2}\epsilon^5}\ln\frac{m+n}{\delta}\Big) \tag{62}$$

where (i) uses $L_\Phi = L_2(1+\kappa)^3 = \mathcal{O}(L_2\kappa^3)$, (ii) uses $T' \leq \sqrt{33L_\Phi}\frac{33(\Phi(x_0)-\Phi^*)+8\epsilon^2}{3\epsilon^3} = \mathcal{O}\big(\sqrt{L_2}\kappa^{1.5}\epsilon^{-3}\big)$ we proved above. $\qquad\square$

## G    Solving Cubic-Regularization problem

In this section, we obtain the properties of the GDA-Cubic Solver (Algorithm 2) and GDA-Cubic FinalSolver (Algorithm 3) in Lemmas G.1-G.3, and show in Proposition G.4 that Algorithm 4 using these cubic solvers admits an intrinsic potential function $H_t$ (see Proposition 4.1) that monotonically decreases over the iterations. Finally, using these lemmas and Proposition G.4, we will prove Theorem 2 on the computation complexity of Algorithm 4.

**Lemma G.1.** *For any $0 < \delta' < 1$, when $\|g\| \leq 4L_1^2\kappa^2\eta_x$, implement Algorithm 2 with initialization $s_0' = 0$ and hyperparameters $K = \Theta\Big(L_1\kappa\eta_x\epsilon'^{-1}\Big[\ln\Big(1 + \frac{\sqrt{m}(L_1^2\kappa^2\eta_x + L_\Phi\epsilon'^2)}{L_\Phi\epsilon'^2\delta'}\Big) + \ln(L_1\kappa\eta_x\epsilon'^{-1})\Big]\Big)]$, $N_k' = N' = \Theta\Big(\frac{\ln[(L_1^3\kappa^3\eta_x^2)/(L_\Phi\epsilon'^3)]}{\ln[(1-\kappa^{-1})^{-1}]}\Big)$, $\eta_v = \frac{1}{L_1}$, $\eta_s = \frac{1}{22L_1\kappa}$, $\sigma = \frac{L_\Phi\epsilon'^3}{108L_1\kappa\eta_x}$. Then, the output $s_K'$ satisfies the following inequalities with probability at least $1 - \delta'$ where $s^* := \arg\min_s \phi(s)$.*

$$\phi(s_K') - \phi(s^*) \leq \frac{\epsilon'\|s^*\|^2}{240\eta_x} + 2L_\Phi\epsilon'^3 \tag{63}$$

$$\|s^*\| \leq 2\|s_K'\| + \frac{\epsilon'}{20} + \frac{24\eta_x L_\Phi\epsilon'^3}{\|s^*\|^2} \tag{64}$$

Lemma G.1 provides convergence properties of GDA-Cubic Solver (Algorithm 2) in the small gradient case $\|g\| \leq 4L_1^2\kappa^2\eta_x$. The key to prove the convergence rate (63) is to equivalently write the gradient descent step (12) as gradient descent on the CR objective function $\phi_{N'}$ defined by eq. (67) below, which is $\epsilon'$-close to the target CR objective function $\phi$ with sufficiently large number $N'$ of gradient ascent steps. Therefore, we can leverage the existing convergence result of gradient descent on $\phi_{N'}$ from (Carmon and Duchi, 2019) and extend the result to $\phi$. The CR function $\phi$ is also close to convex, so small $\phi(s_K') - \phi(s^*)$ means $s^*$ is not far from $s_K'$, which implies eq. (64).

*Proof.* The gradient ascent step (11) can be rewritten as $v_{k,\ell+1} = (I + \eta_v H_{22})v_{k,\ell} + \eta_v H_{12}^\top s_k'$. By iterating it over $\ell = 0, 1, \ldots, N'$ and using $v_{k,0} = 0$, we obtain that

$$v_k = v_{k,N'} = \eta_v \sum_{\ell=0}^{N'-1}(I + \eta_v H_{22})^\ell H_{12}^\top s_k' = -H_{22}^{-1}\big(I - (I + \eta_v H_{22})^{N'}\big)H_{12}^\top s_k'. \tag{65}$$

By substituting the above equality, the gradient descent step (12) can be rewritten as

$$s_{k+1}' = s_k' - \eta_s\Big(g + \sigma\xi + A_{N'}s_k' + \frac{\|s_k'\|}{2\eta_x}s_k'\Big), \text{ where } A_{N'} := H_{11} - H_{12}H_{22}^{-1}\big(I - (I + \eta_v H_{22})^{N'}\big)H_{12}^\top. \tag{66}$$

It can be easily verified that the above update rule (66) can be seen as gradient descent steps with random perturbation $\sigma\xi$ on the following cubic-regularization problem

$$s_{N'}^* = \arg\min_s \phi_{N'}(s) := g^\top s + \frac{1}{2}s^\top A_{N'}s + \frac{1}{6\eta_x}\|s\|^3 \tag{67}$$

Note that

$$\|A_{N'}\| \leq \|H_{11}\| + \|H_{12}\|\|H_{22}^{-1}\|\big\|I - (I + \eta_v H_{22})^{N'}\big\|\|H_{12}\|$$

$$
\overset{(i)}{\leq} \|H_{11}\| + \|H_{12}\| \|H_{22}^{-1}\| \|H_{12}\|
$$

$$
\overset{(ii)}{\leq} L_1 + L_1 \mu^{-1} L_1 \leq A_{\max} := 2L_1 \kappa \tag{68}
$$

where (i) uses $-L_1 I \preceq H_{22} \preceq -\mu I$ (Assumption 1) and $\eta_v = 1/L_1$ which imply that $O \preceq I + \eta_v H_{22} \preceq (1 - \kappa^{-1})I$ and thus $\|I - (I + \eta_v H_{22})^{N'}\| \leq 1$, and (ii) uses $\|H_{11}\| \leq L_1$, $\|H_{12}\| \leq L_1$, $\|H_{22}^{-1}\| \leq \mu^{-1}$ (Assumption 1). Similarly, we obtain that

$$
\|A\| \leq \|H_{11}\| + \|H_{12}\| \|H_{22}^{-1}\| \|H_{12} \leq A_{\max} := 2L_1 \kappa, \tag{69}
$$

and that

$$
\|A_{N'} - A\| = \|(I + \eta_v H_{22})^{N'}\overset{(i)}{\leq} (1 - \kappa^{-1})^{N'} \|A_0 - A\| \overset{(ii)}{\leq} \frac{L_\Phi \epsilon'^3}{(7L_1 \kappa \eta_x)^2}, \tag{70}
$$

where (i) uses $-L_1 I \preceq H_{22} \preceq -\mu I$ (Assumption 1) and $\eta_v = 1/L_1$ which imply that $O \preceq I + \eta_v H_{22} \preceq (1 - \kappa^{-1})I$, and (ii) uses $N' = \frac{\ln[(98L_1^3 \kappa^3 \eta_x^2)/(L_\Phi \epsilon'^3)]}{\ln[(1-\kappa^{-1})^{-1}]}$, $A_0 = O$ and $\|A\| \leq 2L_1 \kappa$.

Hence, the optimal solutions $s^* := \arg\min_s \phi(s)$ satisfies

$$
\|s^*\| \overset{(i)}{\leq} \|A\|\eta_x + \sqrt{(\|A\|\eta_x)^2 + 2\eta_x \|g\|}
$$

$$
\overset{(ii)}{\leq} 2\|A\|\eta_x + \sqrt{2\eta_x \|g\|}
$$

$$
\overset{(iii)}{\leq} 7L_1 \kappa \eta_x := s_{\max}, \tag{71}
$$

where the proof of (i) is in eq. (7a) of (Carmon and Duchi, 2019), (ii) uses the inequality that $\sqrt{a+b} \leq \sqrt{a} + \sqrt{b}$ for any $a, b \geq 0$, and (iii) uses $\|A\| \leq 2L_1 \kappa$ and $\|g\| \leq 4L_1^2 \kappa^2 \eta_x$. Similarly, $s_{N'}^* := \arg\min_s \phi_{N'}(s)$ satisfies

$$
\|s_{N'}^*\| \leq s_{\max}. \tag{72}
$$

Then, based on Lemma 2.6 and Theorem 3.2 of (Carmon and Duchi, 2019), the gradient descent step (66) of the cubic-regularization problem (67) yields that $\|s_k'\| \leq \|s_{N'}^*\| \leq s_{\max}$ for all $k$ and that $\phi_{N'}(s_k') \leq \phi_{N'}(s_{N'}^*) + \frac{\epsilon' \|s_{N'}^*\|^2}{240\eta_x} + L_\Phi \epsilon'^3$ with probability at least $1 - \delta'$, using the hyperparameter choices below which can be easily verified to be satisfied by those used in this Lemma.

$$
s_0' = 0
$$

$$
\eta_s = \frac{1}{4(A_{\max} + s_{\max}/(2\eta_x))} = \frac{1}{22L_1 \kappa}
$$

$$
\sigma = \frac{[\epsilon' \|s_{N'}^*\|^2/(240\eta_x) + L_\Phi \epsilon'^3]/(2\eta_x)}{A_{\max} + \|s_{N'}^*\|/\eta_x} \frac{\overline{\sigma}}{12} = \frac{(2L_\Phi \epsilon'^3)/(2\eta_x)}{12(2L_1 \kappa + s_{\max}/\eta_x)} = \frac{L_\Phi \epsilon'^3}{108L_1 \kappa \eta_x} \tag{73}
$$

$$
K \geq \frac{6 \ln\left(1 + \frac{3\sqrt{m}}{\delta' \sigma_{\min}}\right) + 14 \ln \frac{\|s_{N'}^*\|^2 (A_{\max} + s_{\max}/\eta_x)}{\epsilon' \|s_{N'}^*\|^2/(240\eta_x) + L_\Phi \epsilon'^3}}{(1/2)\eta_s} \frac{10\|s_{N'}^*\|^2}{\epsilon' \|s_{N'}^*\|^2/(240\eta_x) + L_\Phi \epsilon'^3} \tag{74}
$$

$$
= \frac{211200 L_1 \kappa \eta_x \left[3 \ln\left(1 + \frac{3\sqrt{m}(5L_1^2 \kappa^2 \eta_x + 24L_\Phi \epsilon'^2)}{10L_\Phi \epsilon'^2 \delta'}\right) + 7 \ln \frac{2160 L_1 \kappa \eta_x}{\epsilon' + 240\eta_x L_\Phi \epsilon'^3 \|s_{N'}^*\|^{-2}}\right]}{\epsilon' + 240\eta_x L_\Phi \epsilon'^3 \|s_{N'}^*\|^{-2}},
$$

where eq. (73) corresponds to $\overline{\sigma} = \frac{A_{\max} + \|s_{N'}^*\|/\eta_x}{A_{\max} + s_{\max}/\eta_x} \frac{2L_\Phi \epsilon'^3}{\epsilon' \|s_{N'}^*\|^2/(240\eta_x) + L_\Phi \epsilon'^3} \in [\sigma_{\min}, 1]$ ($\sigma_{\min} := \frac{10L_\Phi \epsilon'^2}{5L_1^2 \kappa^2 \eta_x + 24L_\Phi \epsilon'^2}$, since $0 \leq \|s_{N'}^*\| \leq s_{\max} = 7L_1 \kappa \eta_x$) defined in Theorem 3.2 of (Carmon and Duchi, 2019), which yields $\sigma_{\min}$ and $(1/2)\eta$ in eq. (74).

Then, eq. (63) can be proved as follows

$$
\phi(s_K') - \phi(s^*) = \left(\phi(s_K') - \phi_{N'}(s_K')\right) + \left(\phi_{N'}(s_K') - \phi_{N'}(s_{N'}^*)\right) + \left(\phi_{N'}(s_{N'}^*) - \phi(s^*)\right)
$$

$$
\begin{aligned}
&\overset{(i)}{\leq} \left( \phi(s'_K) - \phi_{N'}(s'_K) \right) + \frac{\epsilon' \|s^*_{N'}\|^2}{240\eta_x} + L_\Phi \epsilon'^3 + \left( \phi_{N'}(s^*) - \phi(s^*) \right) \\
&\overset{(ii)}{\leq} \frac{1}{2} s'^\top_K (A - A_{N'}) s'_K + \frac{\epsilon' \|s^*_{N'}\|^2}{240\eta_x} + L_\Phi \epsilon'^3 + \frac{1}{2} s^{*\top}(A - A_{N'}) s^*, \\
&\overset{(iii)}{\leq} (7L_1 \kappa \eta_x)^2 \cdot \frac{L_\Phi \epsilon'^3}{(7L_1\kappa\eta_x)^2} + \frac{\epsilon'\|s^*_{N'}\|^2}{240\eta_x} + L_\Phi \epsilon'^3 \\
&= \frac{\epsilon'\|s^*_{N'}\|^2}{240\eta_x} + 2L_\Phi \epsilon'^3
\end{aligned}
\tag{75}
$$

where (i) uses $\phi_{N'}(s'_k) \leq \phi_{N'}(s^*_{N'}) + \frac{\epsilon'\|s^*_{N'}\|^2}{240\eta_x} + L_\Phi \epsilon'^3$ and $\phi_{N'}(s^*_{N'}) \leq \phi_{N'}(s^*)$, (ii) uses the definitions of $\phi(\cdot)$ and $\phi_{N'}(\cdot)$ in eqs. (6) & (67) respectively, (iii) uses eq. (70) and $\max(\|s^*\|, \|s'_k\|) \leq s_{\max} := 7L_1 \kappa \eta_x$.

Eq. (64) can be proved as follows.

$$
\begin{aligned}
\frac{\epsilon'\|s^*\|^2}{240\eta_x} + 2L_\Phi \epsilon'^3 &\overset{(i)}{\geq} \phi(s'_K) - \phi(s^*) \\
&\overset{(ii)}{=} \frac{1}{2}(s'_K - s^*)^\top \left( A + \frac{\|s^*\|}{2\eta_x} I \right)(s'_K - s^*) + \frac{1}{12\eta_x}(\|s^*\| - \|s'_K\|)^2 (\|s^*\| + 2\|s'_K\|) \\
&\overset{(iii)}{\geq} \frac{\|s^*\|}{12\eta_x}(\|s^*\|^2 + \|s'_K\|^2 - 2\|s'_K\|\|s^*\|) \\
&\geq \frac{\|s^*\|^2}{12\eta_x}(\|s^*\| - 2\|s'_K\|),
\end{aligned}
$$

where (i) uses eq. (63) proved above, (ii) uses eq. (6) of (Carmon and Duchi, 2019) and (iii) uses Proposition 2.1 of (Carmon and Duchi, 2019) which states that $A + \frac{\|s^*\|}{2\eta_x} I \succeq O$. $\qquad\square$

**Lemma G.2.** *When* $\|g\| > 4L_1^2 \kappa^2 \eta_x$*, implement Algorithm 2 with hyperparameters* $K \geq \frac{2\ln[100/(7\eta_x)]}{\ln[(1-\kappa^{-1})^{-1}]}$*,* $\eta_v = 1/L_1$ *and initialize* $s_0 = 0$*. Then, the output* $s'_K$ *satisfies* $\phi(s'_K) \leq \frac{7}{20}\sqrt{\frac{\epsilon^3}{L_2}}$ *with probability at least* $1 - \delta$*. Correspondingly, the approximate CR solution* $s_t$ *in Algorithm 1 satisfies*

$$
\|s'_K\| \geq L_1 \kappa \eta_x \tag{76}
$$

$$
\phi(s'_K) \leq -\frac{1}{4} L_1^3 \kappa^3 \eta_x^2 \tag{77}
$$

Lemma G.2 provides the properties of the solution given by GDA-Cubic Solver (Algorithm 2) in the large gradient case $\|g\| > 4L_1^2\kappa^2\eta_x$. The main idea of proof is to show that $s'_K = -\frac{\gamma_K}{\|g\|} g$ is close to $-\frac{\gamma^*}{\|g\|} g$, so the properties about $s'_K$ can be approximately obtained by studying $-\frac{\gamma^*}{\|g\|} g$. To show that $\gamma_K \approx \gamma^*$, we only need to show that $w_K$ obtained by gradient ascent step of $\mu$-strongly concave function converges to $w^*$ exponentially fast as shown in eq. (80), and that there exists a Lipschitz continuous function $\xi$ such that $\gamma_K = \xi\left( \frac{g^\top}{\|g\|} H_{11} \frac{g}{\|g\|} - \frac{(H_{21}g)^\top w_K}{\|g\|} \right)$ and $\gamma^* = \xi\left( \frac{g^\top}{\|g\|} H_{11} \frac{g}{\|g\|} - \frac{(H_{21}g)^\top w^*}{\|g\|} \right)$.

*Proof.* $\gamma^*$ has the following lower bound.

$$
\begin{aligned}
\gamma^* &= \sqrt{\left( \frac{\eta_x g^\top A g}{\|g\|^2} \right)^2 + 2\eta_x \|g\|} - \frac{\eta_x g^\top A g}{\|g\|^2} \\
&\overset{(i)}{\geq} \eta_x \left( \sqrt{\left( \frac{g^\top A g}{\|g\|^2} \right)^2 + 8L_1^2\kappa^2} - \frac{g^\top A g}{\|g\|^2} \right) \\
&\overset{(ii)}{\geq} \frac{7}{5} L_1 \kappa \eta_x
\end{aligned}
\tag{78}
$$

where (i) uses $\|g\| \geq 4L_1^2\kappa^2\eta_x$, and (ii) uses the monotonically decreasing property of the function $\sqrt{x^2 + 8L_1^2} - x$ and $\|A\| \leq \|H_{11}\| + \|H_{12}\|\|H_{22}^{-1}\|\|H_{21}\| \leq L_1(1 + \kappa) \leq 2L_1\kappa$ based on Lemma A.1.

$\gamma^* = \arg\min_{\gamma \geq 0} \phi(-\gamma g/\|g\|)$ also satisfies the stationary condition below

$$0 = \left.\frac{\partial \phi(-\gamma g)}{\partial \gamma}\right|_{\gamma = \gamma^*} = -\|g\| + \frac{\gamma^*}{\|g\|^2}g^\top A g + \frac{\gamma^{*2}}{2\eta_x}. \tag{79}$$

Note that the gradient steps (10) aim at the $L_1$-smooth, $\mu$-strongly concave maximization problem $w^* := \arg\max_w \frac{1}{2}w^\top H_{22}w - \frac{(H_{21}g)^\top}{\|g\|}w$. Therefore, these gradient steps with learning rate $\eta_v = 1/L_1$ and initialization $w_0 = 0$ have the following convergence rate

$$\|w_K - w^*\| \leq (1 - \kappa^{-1})^{K/2}\|w_0 - w^*\| \overset{(i)}{\leq} \frac{7}{100}\kappa\eta_x, \tag{80}$$

where (i) uses $w_0 = 0$, $\|w^*\| \leq \left\|H_{22}^{-1}\frac{H_{21}g}{\|g\|}\right\| \leq \|H_{22}^{-1}\|\|H_{21}\| \leq \mu^{-1}L_1 = \kappa$ and $K \geq \frac{2\ln[40/(7\eta_x)]}{\ln[(1-\kappa^{-1})^{-1}]}$.

Denote the function $\xi(u) = \sqrt{u^2 + 2\eta_x\|g\|} - u \, (u \in \mathbb{R})$. Then we have

$$
\begin{aligned}
|\gamma_K - \gamma^*| &= \left|\xi\left(\frac{g^\top}{\|g\|}H_{11}\frac{g}{\|g\|} - \frac{(H_{21}g)^\top w_K}{\|g\|}\right) - \xi\left(\frac{g^\top}{\|g\|}H_{11}\frac{g}{\|g\|} - \frac{(H_{21}g)^\top w^*}{\|g\|}\right)\right| \\
&\overset{(i)}{=} \left|\xi'\left(\frac{g^\top}{\|g\|}H_{11}\frac{g}{\|g\|} - \frac{(H_{21}g)^\top[\omega w_K + (1-\omega)w^*]}{\|g\|}\right)\frac{(H_{21}g)^\top(w_K - w^*)}{\|g\|}\right| \\
&\overset{(ii)}{\leq} \frac{7}{50}L_1\kappa\eta_x \overset{(iii)}{\leq} 0.1\gamma^*
\end{aligned}
$$

where (i) applies the Lagrange Mean Value Theorem to the function $\xi$ with $\omega \in [0,1]$, (ii) uses $|\xi'(x)| = \left|\frac{x}{\sqrt{x^2 + 2\eta_x\|g\|}} - 1\right| \leq 2$, $\|H_{21}\| \leq L_1$ and eq. (80), and (iii) uses eq. (78). The above inequality implies that

$$0.9\gamma^* \leq \gamma_K \leq 1.1\gamma^* \tag{81}$$

Therefore, eq. (76) can be proved as follows.

$$\|s_K'\| \overset{(i)}{=} \gamma_K \overset{(ii)}{\geq} 0.9\gamma^* \overset{(iii)}{\geq} \frac{7(0.9)}{5}L_1\kappa\eta_x \geq L_1\kappa\eta_x$$

where (i) uses $s_K' = -\frac{\gamma_K}{\|g\|}g$, (ii) uses eq. (81), and (iii) uses eq. (78).

Eq. (77) can be proved as follows.

$$
\begin{aligned}
\phi(s_K') &= -\gamma_K\|g\| + \frac{\gamma_K^2}{2\|g\|^2}g^\top A g + \frac{\gamma_K^3}{6\eta_x} \\
&\overset{(i)}{\leq} -\frac{9\gamma^*}{10}\|g\| + \frac{(1.1\gamma^*)^2}{2\|g\|^2}g^\top A g + \frac{(1.1\gamma^*)^3}{6\eta_x} \\
&\overset{(ii)}{\leq} -\frac{\gamma^*}{4}\|g\| - \frac{(\gamma^*)^3}{20\eta_x} \\
&\overset{(iii)}{\leq} -\frac{(1.4L_1\kappa\eta_x)(4L_1^2\kappa^2\eta_x)}{20} - \frac{(1.4L_1\kappa\eta_x)^3}{2500\eta_x} \\
&= -\frac{1}{4}L_1^3\kappa^3\eta_x^2
\end{aligned}
$$

where (i) uses eq. (81), (ii) uses eq. (79), and (iii) uses eq. (78) and $\|g\| \geq 4L_1^2\kappa^2\eta_x$. $\qquad\square$

**Lemma G.3.** *Implement Algorithm 3 with initialization $s_0' = 0$ and hyperparameters $K = \Theta\left(\frac{L_1^2 \kappa^2}{L_\Phi^2 \epsilon'^2}\right)$,*
$N_k' = N' = \frac{\ln[2L_1 \kappa \epsilon'^{-1} \max(24\eta_x, 7/L_\Phi)]}{\ln[(1-\kappa^{-1})^{-1}]}$, $\eta_v = \frac{1}{L_1}$, $\eta_s = \frac{1}{22L_1 \kappa}$. *Then, if $\|s^*\| \le 3\epsilon'$ and $\|g\| \le 4L_1^2 \kappa^2 \eta_x$, the*
*algorithm will terminate with $K' = \min\{k : \|g_k\| \le L_\Phi \epsilon'^2\} \le K$ and the output $s_{K'}'$ satisfies*

$$\|s_{K'}'\| \le 7\epsilon'. \tag{82}$$

$$\|\nabla\phi(s_{K'}')\| = \left\|g + As_{K'}' + \frac{\|s_{K'}'\|}{2\eta_x}s_{K'}'\right\| \le 2L_\Phi \epsilon'^2. \tag{83}$$

Lemma G.3 provides convergence properties of GDA-Cubic FinalSolver (Algorithm 3) in the small gradient case $\|g\| \le 4L_1^2 \kappa^2 \eta_x$. Note that Algorithm 3 is noiseless version of Algorithm 2. Hence, the proof logic is similar to Lemma G.1. To elaborate, Algorithm 3 is equivalent to apply gradient descent to the approximate CR objective function $\phi_{N'}$ defined by eq. (67), which is $\epsilon'$-close to the target CR objective function $\phi$. Then based on the nice geometry of the two close CR objective functions, their optimizers $s_{N'}^* := \arg\min_s \phi_{N'}(s)$ and $s^* := \arg\min_s \phi(s)$ also have close norms such that $\|s^*\| \le 3\epsilon'$ implies $\|s_{N'}^*\| \le 7\epsilon'$, as shown by eqs. (87) & (88). Then eq. (82) follows from $\|s_{K'}\| \le \|s_{N'}^*\|$ which has been proved by Lemma 2.6 of (Carmon and Duchi, 2019) for gradient descent steps on CR objective function $\phi$. $K' \le K$ follows from the nonconvex convergence rate of $\min_{0 \le k \le K-1} \|g_k\|^2$ where $g_K := \nabla\phi_{N'}(s_K')$. Finally, eq. (83) follows from $\|g_{K'}\| \le L_\Phi \epsilon'^2$ and $\nabla\phi(s_K') \approx \nabla\phi_{N'}(s_K') = g_K$.

*Proof.* Since $s^* = \arg\min_s \phi(s)$, $0 = \nabla\phi(s^*) = g + As^* + \frac{\|s^*\|}{2\eta_x}s^*$, which implies that

$$\|g\| = \left\|As^* + \frac{\|s^*\|}{2\eta_x}s^*\right\| \le \|A\|\|s^*\| + \frac{\|s^*\|^2}{2\eta_x} \overset{(i)}{\le} 2L_1 \kappa(3\epsilon') + \frac{(3\epsilon')(7L_1 \kappa \eta_x)}{2\eta_x} \le 13L_1 \kappa \epsilon' \tag{84}$$

where (i) uses $\|A\| \le 2L_1\kappa$, $\|s^*\| \le 3\epsilon'$ and $\|s^*\| \le s_{\max} := 7L_1 \kappa \eta_x$.

Following the proof of eq. (70), we can prove that when $N' = \frac{\ln[2L_1 \kappa \epsilon'^{-1} \max(24\eta_x, 7/L_\Phi)]}{\ln[(1-\kappa^{-1})^{-1}]}$

$$\|A_{N'} - A\| = \|(I + \eta_v H_{22})^{N'}\| \le (1-\kappa^{-1})^{N'}\|A_0 - A\| \le 2L_1 \kappa(1-\kappa^{-1})^{N'} \le \min\left(\frac{\epsilon'}{24\eta_x}, \frac{L_\Phi \epsilon'}{7}\right), \tag{85}$$

Substituting eqs. (65) & (66) into eq. (13), we obtain that

$$g_k = g + A_{N'}s_k' + \frac{\|s_k'\|}{2\eta_x}s_k' = \nabla\phi_{N'}(s_k'). \tag{86}$$

Hence, the update rule (14) can be seen as gradient descent step on solving the cubic-regularization problem (67) without random perturbation. Therefore, based on Lemmas 2.3 and eq. (11) of (Carmon and Duchi, 2019), $\|s_k'\| \le \|s_{N'}^*\| \le s_{\max}$ and that when $\eta_s = \frac{1}{22L_1\kappa} \le \frac{1}{4(\|A\| + s_{\max}/(2\eta_x))}$, we have

$$\frac{\eta_s}{2}\sum_{k=0}^{K-1}\|g_k\|^2 \le \phi_{N'}(s_0') - \phi_{N'}(s^*)$$

$$\overset{(i)}{\le} \|g\|\|s^*\| + \frac{1}{2}\|A_{N'}\|\|s^*\|^2 + \frac{\|s^*\|^3}{6\eta_x}$$

$$\overset{(ii)}{\le} 3\epsilon'\left(13L_1 \kappa \epsilon' + L_1 \kappa(3\epsilon') + (3\epsilon')(7L_1 \kappa \eta_x)/(6\eta_x)\right)$$

$$\le \frac{115}{2}L_1 \kappa \epsilon'^2,$$

where (i) uses $s_0'' = 0$ and the definition of function $\phi_{N'}$ in eq. (67), (ii) uses eq. (84), $\|s^*\| \le \max(3\epsilon', 7L_1 \kappa \eta_x)$ and $\|A_{N'}\| \le 2L_1\kappa$. Rearranging the above inequality, we obtain that

$$\min_{0 \le k \le K-1}\|g_k\|^2 \le \frac{1}{K}\sum_{k=0}^{K-1}\|g_k\|^2 \le \frac{115L_1 \kappa \epsilon'^2}{K\eta_s} \overset{(i)}{\le} L_\Phi^2 \epsilon'^4,$$

where (i) uses $K = \frac{2530L_1^2\kappa^2}{L_\Phi^2\epsilon'^2}$ and $\eta_s = \frac{1}{22L_1\kappa}$. Hence, $K' = \min\{k : \|g_k\| \leq L_\Phi\epsilon'^2\} \leq K - 1$.

Next, we will prove eq. (82). On one hand, using the same proof logic as that of eq. (64) (see the end of the proof of Lemma G.1), we obtain that

$$
\begin{aligned}
\phi_{N'}(s^*) - \phi_{N'}(s_{N'}^*) &\overset{(i)}{=} \frac{1}{2}(s^* - s_{N'}^*)^\top \left(A_{N'} + \frac{\|s_{N'}^*\|}{2\eta_x}I\right)(s^* - s_{N'}^*) + \frac{1}{12\eta_x}(\|s_{N'}^*\| - \|s^*\|)^2(\|s_{N'}^*\| + 2\|s^*\|) \\
&\overset{(ii)}{\geq} \frac{\|s_{N'}^*\|}{12\eta_x}(\|s_{N'}^*\|^2 + \|s^*\|^2 - 2\|s^*\|\|s_{N'}^*\|) \\
&\overset{(iii)}{\geq} \frac{\|s_{N'}^*\|^2}{12\eta_x}(\|s_{N'}^*\| - 6\epsilon'),
\end{aligned}
\tag{87}
$$

where (i) uses eq. (6) of (Carmon and Duchi, 2019), (ii) uses Proposition 2.1 of (Carmon and Duchi, 2019) which states that $A + \frac{\|s_{N'}^*\|}{2\eta_x}I \succeq O$ and (iii) uses $\|s^*\| \leq 3\epsilon'$. On the other hand,

$$
\begin{aligned}
\phi_{N'}(s^*) &- \phi_{N'}(s_{N'}^*) \\
&= \left(\phi_{N'}(s^*) - \phi(s^*)\right) + \left(\phi(s^*) - \phi_{N'}(s_{N'}^*)\right) \\
&\overset{(i)}{\leq} \left(\phi_{N'}(s^*) - \phi(s^*)\right) + \left(\phi(s_{N'}^*) - \phi_{N'}(s_{N'}^*)\right) \\
&\overset{(ii)}{\leq} \frac{1}{2}s^{*\top}(A_{N'} - A)s^* + \frac{1}{2}s_{N'}^{*\top}(A - A_{N'})s_{N'}^* \\
&\overset{(iii)}{\leq} \frac{\epsilon'}{24\eta_x}(9\epsilon'^2 + \|s_{N'}^*\|^2),
\end{aligned}
\tag{88}
$$

where (i) uses $\phi(s^*) = \min_s \phi(s) \leq \phi(s_{N'}^*)$, (ii) uses the definitions of $\phi(\cdot)$ and $\phi_{N'}(\cdot)$ in eqs. (6) & (67 respectively, and (iii) uses eq. (85) and $\|s^*\| \leq 3\epsilon'$. Combining eqs. (87) & (88) yields that

$$
\|s_{N'}^*\| \leq 6\epsilon' + \frac{\epsilon'}{2\|s_{N'}^*\|^2}(9\epsilon'^2 + \|s_{N'}^*\|^2) = 6.5\epsilon' + \frac{9\epsilon'^3}{2\|s_{N'}^*\|^2}.
$$

Suppose $\|s_{N'}^*\| > 7\epsilon'$ and substitute it into the right side of the above inequality. Then we obtain the contradiction that $\|s_{N'}^*\| < 6.8\epsilon'$. Therefore, $\|s_{K'}\| \leq \|s_{N'}^*\| \leq 7\epsilon'$, i.e., eq. (82) is proved. Finally, eq. (83) can be proved as follows

$$
\begin{aligned}
\|\nabla\phi(s_{K'})\| &\leq \|\nabla\phi(s_{K'}) - g_{K'}\| + \|g_{K'}\| \\
&\overset{(i)}{\leq} \|(A - A_{N'})s_{K'}\| + L_\Phi\epsilon'^2 \\
&\overset{(ii)}{\leq} \frac{L_\Phi\epsilon'}{7}(7\epsilon') + L_\Phi\epsilon'^2 = 2L_\Phi\epsilon'^2,
\end{aligned}
\tag{89}
$$

where (i) uses $\|g_{K'}\| \leq L_\Phi\epsilon'^2$ and the definitions of $\phi(\cdot)$ and $\phi_{N'}(\cdot)$ in eqs. (6) & (67) respectively, (ii) uses eq. (85) and $\|s_{K'}\| \leq 7\epsilon'$. $\qquad\square$

**Proposition G.4** (Potential decrease for Inexact Cubic-LocalMinimax)**.** *Let Assumption 1 hold. For any $\alpha, \beta > 0$, $0 < \epsilon' \leq \frac{\alpha L_1}{\beta L_G}$ and $\delta \in (0,1)$, choose $\eta_x = (168L_\Phi + 120\alpha + 168\beta)^{-1}$, $\eta_y = \frac{2}{L_1+\mu}$ and $N_t \geq \Theta\left(\kappa \ln \frac{L_1\alpha\|\widetilde{s}_{t-1}\| + L_1(\alpha+L_2\kappa)\|\widetilde{s}_t\|}{L_G\beta\epsilon'^2}\right)$ (see eq. (49)). When implementing Algorithm 2 at the t-th iteration, use hyperparameters in Lemma G.1 with $\delta' = \delta/T$ if $\|\nabla_1 f(x_t, y_{t+1})\| \leq 4L_1^2\kappa^2\eta_x$, and use those in Lemma G.2 otherwise. Define the potential function $H_t := \Phi(x_t) + (10L_\Phi + 7\alpha + 10\beta)\|\widetilde{s}_t\|^3$. Then, the output of Cubic-LocalMinimax satisfies the following potential decrease property with probability at least $1 - \delta$.*

$$
H_{t+1} - H_t \leq -(L_\Phi + \alpha + \beta)(\|\widetilde{s}_{t+1}\|^3 + \|\widetilde{s}_t\|^3).
\tag{90}
$$

Compared with Proposition 4.1 which also provides potential function decrease, Proposition G.4 only has access to the inexact CR solution obtained from GDA-Cubic Solver (Algorithm 2). In small and large gradient

cases respectively, this inexact CR solution has properties given by Lemmas G.1 & G.2, and the Taylor expansion of $\Phi(x_{t+1}) - \Phi(x_t)$ can be upper bounded by eqs. (63) & (77), as shown in eqs. (96) & (98). Both cases yield eq. (90).

*Proof.* Following the proof of Proposition 4.1, it can be seen that when $\epsilon' \leq \frac{\alpha L_1}{\beta L_G}$ and $N_t \geq \Theta\left(\kappa \ln \frac{L_1 \alpha \|\widetilde{s}_{t-1}\| + L_1(\alpha + L_2\kappa)\|\widetilde{s}_t\|}{L_G \beta \epsilon'^2}\right)$, the following bounds always hold, which are analogous to eqs. (34 & 35 with exact solutions $s_{t-1}$ and $s_t$ replaced by $\widetilde{s}_{t-1}$ and $\widetilde{s}_t$ respectively.

$$\|\nabla \Phi(x_t) - \nabla_1 f(x_t, y_{t+1})\| \leq \beta(\|\widetilde{s}_t\|^2 + \epsilon'^2), \tag{91}$$

$$\|\nabla^2 \Phi(x_t) - G(x_t, y_{t+1})\| \leq \alpha(\|\widetilde{s}_t\| + \epsilon'). \tag{92}$$

Then we consider the following two cases.

(Case 1) When $\|\nabla_1 f(x_t, y_{t+1})\| \leq 4L_1^2 \kappa^2 \eta_x$, the output $s'_K$ satisfies eqs. (63) & (94) with probability at least $1 - \delta'$. Also note that the input variables of Algorithm 2 are $g := \nabla_1 f(x_t, y_{t+1})$ and $A := G(x_t, y_{t+1}) = H_{11} - H_{12}H_{22}^{-1}H_{21}$, the output $s'_K$ of Algorithm 2 is assigned to $\widetilde{s}_{t+1}$ which is later used for the update $x_{t+1} = x_t + \widetilde{s}_{t+1}$, and the optimal CR solution $s^*$ in Algorithm 2 corresponds to $s_{t+1}$ in Algorithm 1. Therefore, eqs. (63) & (64) transform to the following inequalities at the $t$-th iteration of Algorithm 1, which by applying union bound hold simultaneously for all $0 \leq t \leq T - 1$ with probability at least $1 - T\delta' = 1 - \delta$.

$$\nabla_1 f(x_t, y_{t+1})^\top(\widetilde{s}_{t+1} - s_{t+1}) + \frac{1}{2}\widetilde{s}_{t+1}^\top G(x_t, y_{t+1})\widetilde{s}_{t+1} - \frac{1}{2}s_{t+1}^\top G(x_t, y_{t+1})s_{t+1} + \frac{\|\widetilde{s}_{t+1}\|^3 - \|s_{t+1}\|^3}{6\eta_x}$$

$$\leq \frac{\epsilon'\|s_{t+1}\|^2}{240\eta_x} + 2L_\Phi \epsilon'^3 \tag{93}$$

$$\|s_t\| \leq 2\|\widetilde{s}_t\| + \frac{\epsilon'}{20} + \frac{24\eta_x L_\Phi \epsilon'^3}{\|s_t\|^2}. \tag{94}$$

Based on eq. (94), if $\|s_t\| > \epsilon'$, then $\|s_t\| \leq 2\|\widetilde{s}_t\| + \epsilon'$ since $\eta_x \leq (168L_\Phi)^{-1}$. Otherwise, $\|s_t\| \leq \epsilon'$. Combining the two cases yields the following inequality.

$$\|s_t\| \leq 2\|\widetilde{s}_t\| + \epsilon' \Rightarrow \|s_t\|^2 \leq 8\|\widetilde{s}_t\|^2 + 2\epsilon'^2. \tag{95}$$

Therefore,

$$\Phi(x_{t+1}) - \Phi(x_t)$$

$$\overset{(i)}{\leq} \widetilde{s}_{t+1}^\top \nabla \Phi(x_t) + \frac{1}{2}\widetilde{s}_{t+1}^\top \nabla^2 \Phi(x_t)\widetilde{s}_{t+1} + \frac{L_\Phi}{6}\|\widetilde{s}_{t+1}\|^3$$

$$= \widetilde{s}_{t+1}^\top\big(\nabla \Phi(x_t) - \nabla_1 f(x_t, y_{t+1})\big) + \widetilde{s}_{t+1}^\top \nabla_1 f(x_t, y_{t+1})$$

$$\quad + \frac{1}{2}\widetilde{s}_{t+1}^\top\big(\nabla^2 \Phi(x_t) - G(x_t, y_{t+1})\big)\widetilde{s}_{t+1} + \frac{1}{2}\widetilde{s}_{t+1}^\top G(x_t, y_{t+1})\widetilde{s}_{t+1} + \frac{L_\Phi}{6}\|\widetilde{s}_{t+1}\|^3$$

$$\overset{(ii)}{\leq} \beta\|\widetilde{s}_{t+1}\|(\|\widetilde{s}_t\|^2 + \epsilon'^2) + \frac{\alpha}{2}\|\widetilde{s}_{t+1}\|^2(\|\widetilde{s}_t\| + \epsilon') + s_{t+1}^\top \nabla_1 f(x_t, y_{t+1}) + \frac{1}{2}s_{t+1}^\top G(x_t, y_{t+1})s_{t+1}$$

$$\quad + \frac{\|s_{t+1}\|^3}{6\eta_x} + \left(\frac{L_\Phi}{6} - \frac{1}{6\eta_x}\right)\|\widetilde{s}_{t+1}\|^3 + \frac{\epsilon'\|s_{t+1}\|^2}{240\eta_x} + 2L_\Phi \epsilon'^3$$

$$\overset{(iii)}{\leq} \left(\frac{\alpha}{2} + \beta\right)(2\|\widetilde{s}_{t+1}\|^3 + \|\widetilde{s}_t\|^3 + \epsilon'^3) - \frac{\|s_{t+1}\|^3}{12\eta_x} + \left(\frac{L_\Phi}{6} - \frac{1}{6\eta_x}\right)\|\widetilde{s}_{t+1}\|^3 + \frac{4\epsilon'\|\widetilde{s}_{t+1}\|^2 + \epsilon'^3}{120\eta_x} + 2L_\Phi \epsilon'^3$$

$$\overset{(iv)}{\leq} \left(\frac{\alpha}{2} + \beta\right)(3\|\widetilde{s}_{t+1}\|^3 + 2\|\widetilde{s}_t\|^3) + \left(\frac{L_\Phi}{6} - \frac{1}{6\eta_x}\right)\|\widetilde{s}_{t+1}\|^3$$

$$\quad + \frac{4\|\widetilde{s}_{t+1}\|^2(\|\widetilde{s}_{t+1}\| + \|\widetilde{s}_t\|) + \|\widetilde{s}_{t+1}\|^3 + \|\widetilde{s}_t\|^3}{120\eta_x} + 2L_\Phi(\|\widetilde{s}_{t+1}\|^3 + \|\widetilde{s}_t\|^3)$$

$$\overset{(v)}{\leq} \left(\frac{\alpha}{2} + \beta\right)(3\|\widetilde{s}_{t+1}\|^3 + 2\|\widetilde{s}_t\|^3) + \left(\frac{L_\Phi}{6} - \frac{1}{6\eta_x}\right)\|\widetilde{s}_{t+1}\|^3$$

$$+ \frac{4\|\widetilde{s}_{t+1}\|^3 + 4(\|\widetilde{s}_t\|^3 + \|\widetilde{s}_{t+1}\|^3) + \|\widetilde{s}_{t+1}\|^3 + \|\widetilde{s}_t\|^3}{120\eta_x} + 2L_\Phi(\|\widetilde{s}_{t+1}\|^3 + \|\widetilde{s}_t\|^3)$$

$$\leq \left(\frac{3\alpha}{2} + 3\beta + \frac{13L_\Phi}{6} - \frac{1}{12\eta_x}\right)\|\widetilde{s}_{t+1}\|^3 + \left(\alpha + 2\beta + 2L_\Phi + \frac{1}{24\eta_x}\right)\|\widetilde{s}_t\|^3$$

$$\overset{(vi)}{\leq} -(11L_\Phi + 8\alpha + 11\beta)\|\widetilde{s}_{t+1}\|^3 + (9L_\Phi + 6\alpha + 9\beta)\|\widetilde{s}_t\|^3 \tag{96}$$

where (i) uses $x_{t+1} = x_t + s_{t+1}$ and the fact that $\nabla^2\Phi(x)$ is $L_\Phi$-Lipschitz continuous (see the item 4 of Proposition 3.1), (ii) uses eqs. (91), (92) & (93), (iii) uses eqs. (33) & (95) and the inequality that $ab^2 \leq a^3 \vee b^3 \leq a^3 + b^3, \forall a, b \geq 0$, (iv) uses $\epsilon'^n \leq \|\widetilde{s}_t\|^n \vee \|\widetilde{s}_{t+1}\|^n \leq \|\widetilde{s}_t\|^n + \|\widetilde{s}_{t+1}\|^n, \forall 0 \leq t \leq T'-2, n \in \{1,3\}$ based on the termination criterion of $T'$ in Algorithm 4, (v) uses $ab^2 \leq a^3 \vee b^3 \leq a^3 + b^3, \forall a, b \geq 0$, and (vi) uses $\eta_x = (168L_\Phi + 120\alpha + 168\beta)^{-1}$.

(Case 2) When $\|\nabla_1 f(x_t, y_{t+1})\| > 4L_1^2\kappa^2\eta_x$, similar to case 1, eq. (77) transforms as follows in Algorithm 1.

$$\nabla_1 f(x_t, y_{t+1})^\top \widetilde{s}_{t+1} + \frac{1}{2}\widetilde{s}_{t+1}^\top G(x_t, y_{t+1})\widetilde{s}_{t+1} + \frac{1}{6\eta_x}\|\widetilde{s}_{t+1}\|^3 \leq -\frac{1}{4}L_1^3\kappa^3\eta_x^2 \tag{97}$$

Then, we obtain that

$$\Phi(x_{t+1}) - \Phi(x_t)$$

$$\leq s_{t+1}^\top \nabla\Phi(x_t) + \frac{1}{2}s_{t+1}^\top \nabla^2\Phi(x_t)s_{t+1} + \frac{L_\Phi}{6}\|s_{t+1}\|^3$$

$$= s_{t+1}^\top\big(\nabla\Phi(x_t) - \nabla_1 f(x_t, y_{t+1})\big) + s_{t+1}^\top \nabla_1 f(x_t, y_{t+1})$$

$$\quad + \frac{1}{2}s_{t+1}^\top\big(\nabla^2\Phi(x_t) - G(x_t, y_{t+1})\big)s_{t+1} + \frac{1}{2}s_{t+1}^\top G(x_t, y_{t+1})s_{t+1} + \frac{L_\Phi}{6}\|s_{t+1}\|^3$$

$$\overset{(i)}{\leq} \beta\|s_{t+1}\|(\|s_t\|^2 + \epsilon'^2) + \frac{\alpha}{2}\|s_{t+1}\|^2(\|s_t\| + \epsilon') + \left(\frac{L_\Phi}{6} - \frac{1}{6\eta_x}\right)\|s_{t+1}\|^3 - \frac{1}{4}L_1^3\kappa^3\eta_x^2$$

$$\overset{(ii)}{\leq} -(11L_\Phi + 8\alpha + 11\beta)\|\widetilde{s}_{t+1}\|^3 + (9L_\Phi + 6\alpha + 9\beta)\|\widetilde{s}_t\|^3 \tag{98}$$

where (i) uses eqs. (91), (92) & (97), and (ii) follows the same proof logic as that of eq. (96).

Eq. (96) in case 1 and eq. (98) in case 2 are the same. By rearranging them and using $H_t := \Phi(x_t) + (10L_\Phi + 7\alpha + 10\beta)\|\widetilde{s}_t\|^3$, we can prove eq. (90). $\qquad\square$

**Theorem 2** (Computation complexity of Inexact Cubic-LocalMinimax). *Let Assumption 1 hold. For any $0 < \epsilon \leq \min\left(\frac{53L_1\kappa}{228\sqrt{L_\Phi}}, L_1^2L_2^{-1/2}\kappa^{1/2}, \frac{L_2\kappa^2}{L_1}\right)$ and $\delta \in (0,1)$, choose $\epsilon' = \frac{\epsilon}{106\sqrt{L_\Phi}}$, $T = \Theta\big(\sqrt{L_\Phi}[\Phi(x_0) - \Phi^* + \epsilon^2]\epsilon^{-3}\big)$, $\eta_x = \Theta\big(L_\Phi^{-1}\big)$, $\eta_y = \frac{2}{L_1+\mu}$ and $N_t = \Theta\Big(\kappa \ln\frac{L_1\alpha\|\widetilde{s}_{t-1}\| + L_1(\alpha + L_2\kappa)\|\widetilde{s}_t\|}{L_G\epsilon^2}\Big)$ (see eq. (49)) in Algorithm 4. When implementing Algorithm 2 at the $t$-th iteration, use hyperparameters in Lemma G.1 with $\delta' = \delta/T$ if $\|\nabla_1 f(x_t, y_{t+1})\| \leq 4L_1^2\kappa^2\eta_x$, and use those in Lemma G.2 otherwise. When implementing Algorithm 3, use the hyperparameter choices in Lemma G.3. Then, with probability at least $1 - \delta$, the output of Inexact Cubic-LocalMinimax satisfies*

$$\mu(\widetilde{x}_{T'}) \leq \epsilon. \tag{15}$$

*Consequently, the total number of required cubic iterations satisfies $T' \leq \mathcal{O}\big(\sqrt{L_2}\kappa^{1.5}\epsilon^{-3}\big)$, the total number of required gradient ascent iterations satisfies $\sum_{t=0}^{T'-1} N_t \leq \widetilde{\mathcal{O}}\big(\sqrt{L_2}\kappa^{2.5}\epsilon^{-3}\big)$, and the total number of required Hessian-vector product computations (in Algorithms 2 & 3) is of the order $\widetilde{O}(L_1\kappa^2\epsilon^{-4})$.*

The proof logic is very similar to that of Theorem 1, with the major difference that we leverage the properties of inexact CR solution given by Lemmas G.1-G.3, and the potential decrease given by Proposition G.4 instead of Proposition 4.1.

*Proof.* First, it can be easily verified that the hyperparameter choices of this Theorem fits those in Proposition G.4 with $\alpha = \beta = L_\Phi$. In particular, $\epsilon' = \frac{\epsilon}{106\sqrt{L_\Phi}}$ with $0 < \epsilon \leq \frac{106 L_1 \kappa}{\sqrt{L_\Phi}}$ and $\delta \in (0, 1)$ satisfies $0 < \epsilon' \leq \frac{\alpha L_1}{\beta L_G}$ required by Proposition G.4, since $\eta_x = (456 L_\Phi)^{-1}$, $\alpha = \beta = L_\Phi$ and $L_G = L_\Phi/(1+\kappa) \leq L_\Phi/\kappa$ (see Proposition 3.1).

Suppose $T' \leq T$ does not hold, i.e., $\|s_{t-1}\| \vee \|s_t\| > \epsilon' = \frac{\epsilon}{106\sqrt{L_\Phi}}, \forall 1 \leq t \leq T$. Then, on one hand, telescoping eq. (90) over $t = 0, 1, \ldots, T-1$ yield that

$$
\begin{aligned}
H_0 - H_T &\geq (L_\Phi + \alpha + \beta) \sum_{t=0}^{T-1} (\|\widetilde{s}_{t+1}\|^3 + \|\widetilde{s}_t\|^3) \\
&\geq 3 L_\Phi \sum_{t=0}^{T-1} (\|s_t\| \vee \|s_{t+1}\|)^3 \\
&\geq 3 T L_\Phi \Big(\frac{\epsilon}{106\sqrt{L_\Phi}}\Big)^3 \\
&\overset{(i)}{\geq} \Phi(x_0) - \Phi^* + \epsilon^2.
\end{aligned}
\tag{99}
$$

where (i) uses $T = 397006\sqrt{L_\Phi}[\Phi(x_0) - \Phi^* + \epsilon^2]\epsilon^{-3}$. On the other hand, recalling the definition of $H_t$ in Proposition G.4, we have

$$
H_0 - H_T = \Phi(x_0) - \Phi(x_T) + 27 L_\Phi(\|\widetilde{s}_0\|^2 - \|\widetilde{s}_T\|^2) \overset{(i)}{\leq} \Phi(x_0) - \Phi^* + \frac{\epsilon^2}{137},
\tag{100}
$$

where (i) uses $\|s_0\| = \epsilon' = \frac{\epsilon}{106\sqrt{L_\Phi}}$ and $\Phi(x_T) \geq \Phi^* = \min_{x \in \mathbb{R}^m} \Phi(x)$. Note that eqs. (99) & (100) contradict. Therefore, we must have $1 \leq T' \leq T = 397006\sqrt{L_\Phi}[\Phi(x_0) - \Phi^* + \epsilon^2]\epsilon^{-3}$.

Since $\|\widetilde{s}_{T'}\| \leq \epsilon'$, we have $\|\nabla_1 f(x_{T'}, y_{T'+1})\| \leq 4 L_1^2 \kappa^2 \eta_x$. Otherwise, eq. (76) directly implies the ccontradiction that $\|\widetilde{s}_{T'}\| \geq L_1 \kappa \eta_x = \frac{L_1 \kappa}{456 L_\Phi} > \epsilon' = \frac{\epsilon}{106\sqrt{L_\Phi}}$ (since $\epsilon < \frac{53 L_1 \kappa}{228\sqrt{L_\Phi}}$). Hence, $\|\nabla_1 f(x_{T'}, y_{T'+1})\| \leq 4 L_1^2 \kappa^2 \eta_x$, which implies eq. (95) and thus implies $\|s_{T'}\| \leq 2\|\widetilde{s}_{T'}\| + \epsilon' \leq 3\epsilon'$. Therefore, the conditions of Lemma G.3 are met, so eqs. (82) & (83) hold. Rewriting eqs. (82) & (83) with $g \leftarrow \nabla_1 f(x_{T'-1}, y_{T'})$, $A \leftarrow G(x_{T'-1}, y_{T'})$ and $\widetilde{s} \leftarrow s'_{K'}$ yields that

$$
\|\widetilde{s}\| \leq 7\epsilon'
\tag{101}
$$

$$
\left\|\nabla_1 f(x_{T'-1}, y_{T'}) + G(x_{T'-1}, y_{T'})\widetilde{s} + \frac{\|\widetilde{s}\|}{2\eta_x}\widetilde{s}\right\| \leq 2 L_\Phi \epsilon'^2
\tag{102}
$$

Therefore,

$$
\begin{aligned}
&\|\nabla\Phi(\widetilde{x}_{T'})\| \\
&\overset{(i)}{=} \left\|\nabla\Phi(\widetilde{x}_{T'}) - \nabla_1 f(x_{T'-1}, y_{T'}) - G(x_{T'-1}, y_{T'})\widetilde{s} - \frac{\|\widetilde{s}\|}{2\eta_x}\widetilde{s}\right\| + 2 L_\Phi \epsilon'^2 \\
&\leq \|\nabla\Phi(\widetilde{x}_{T'}) - \nabla\Phi(x_{T'-1}) - \nabla^2\Phi(x_{T'-1})\widetilde{s}\| + \|\nabla\Phi(x_{T'-1}) - \nabla_1 f(x_{T'-1}, y_{T'})\| \\
&\quad + \|\nabla^2\Phi(x_{T'-1})\widetilde{s} - G(x_{T'-1}, y_{T'})\widetilde{s}\| + \frac{\|\widetilde{s}\|^2}{2\eta_x} + 2 L_\Phi \epsilon'^2 \\
&\overset{(ii)}{\leq} L_\Phi\|\widetilde{s}\|^2 + \beta(\|\widetilde{s}_{T'-1}\|^2 + \epsilon'^2) + 7\alpha\epsilon'(\|\widetilde{s}_{T'-1}\| + \epsilon') + 228 L_\Phi(7\epsilon')^2 + 2 L_\Phi \epsilon'^2 \\
&\overset{(iii)}{\leq} 11191 L_\Phi \epsilon'^2 \overset{(iv)}{=} \epsilon^2,
\end{aligned}
\tag{103}
$$

where (i) uses eq. (102), (ii) uses eqs. (91), (92) & (101), $\widetilde{x}_{T'} = x_{T'-1} + \widetilde{s}$ and the item 4 of Proposition 3.1 that $\nabla^2\Phi$ is $L_\Phi$-Lipschitz, (iii) uses $\eta_x = (168 L_\Phi + 120\alpha + 168\beta)^{-1} = (456 L_\Phi)^{-1}$, $\|\widetilde{s}_{T'-1}\| \leq \epsilon'$, $\alpha = \beta = L_\Phi$ and eq. (101), and (iv) uses $\epsilon' = \frac{\epsilon}{106\sqrt{L_\Phi}}$. Also,

$$
\nabla^2\Phi(\widetilde{x}_{T'}) \overset{(i)}{\succeq} G(x_{T'-1}, y_{T'}) - \|G(x_{T'-1}, y_{T'}) - \nabla^2\Phi(x_{T'-1})\|I - \|\nabla^2\Phi(\widetilde{x}_{T'}) - \nabla^2\Phi(x_{T'-1})\|I
$$

$$\overset{(ii)}{\succeq} -\frac{1}{2\eta_x}\|s_{T'}\|I - \alpha(\|\widetilde{s}_{T'-1}\| + \epsilon')I - L_\Phi\|\widetilde{s}\|I$$

$$\overset{(iii)}{\succeq} -237L_\Phi\epsilon'I \succeq -3\sqrt{L_\Phi}\epsilon I, \tag{104}$$

where (i) uses Weyl's inequality, (ii) uses $\widetilde{x}_{T'} = x_{T'-1} + \widetilde{s}$, eqs. (32) & (92) and the item 4 of Proposition 3.1 that $\nabla^2\Phi$ is $L_\Phi$-Lipschitz, (iii) uses eq. (101), $\eta_x = (456L_\Phi)^{-1}$, $\alpha = L_\Phi$, $\|\widetilde{s}_{T'-1}\| \leq \epsilon'$, and (iv) uses $\epsilon' = \frac{\epsilon}{106\sqrt{L_\Phi}}$. Combining eqs. (103) & (104) proves eq. (15) where $\mu(x) = \sqrt{\|\nabla\Phi(x)\|} \vee \frac{-\lambda_{\min}[\nabla^2\Phi(x)]}{\sqrt{33L_\Phi}}$.

Finally, we compute the computation complexities. We have proved that the total number of cubic iterations satisfies $T' \leq T = 397006\sqrt{L_\Phi}[\Phi(x_0) - \Phi^* + \epsilon^2]\epsilon^{-3} = \mathcal{O}(\sqrt{L_2}\kappa^{1.5}\epsilon^{-3})$ $(L_\Phi = L_2(1+\kappa)^3 = \mathcal{O}(L_2\kappa^3)$ based on Proposition 3.1). We can also prove that the total number of gradient ascent iterations have the same bound $\sum_{t=0}^{T'-1} N_t \leq \widetilde{\mathcal{O}}(\sqrt{L_2}\kappa^{2.5}\epsilon^{-3})$ as that of Theorem 1, following the proof logic at the end of Appendix D. Then we compute the total number of Hessian-vector product computations in cubic solvers (Algorithms 2 & 3). When implementing Algorithm 2 at the $t$-th iteration, if $\|\nabla_1 f(x_t, y_{t+1})\| \leq 4L_1^2\kappa^2\eta_x$, then based on Lemma G.1, the number of Hessian-vector product computations is proportional to

$$KN' = \frac{2\ln[(98L_1^3\kappa^3\eta_x^2)/(L_\Phi\epsilon'^3)]}{\ln[(1 - \kappa^{-1})^{-1}]} \cdot \widetilde{\mathcal{O}}(L_1\kappa\eta_x\epsilon'^{-1})$$

$$= \widetilde{\mathcal{O}}\Big[L_1\kappa^2 L_\Phi^{-1}\Big(\frac{\epsilon}{\sqrt{L_\Phi}}\Big)^{-1}\Big]$$

$$= \widetilde{O}(L_1 L_\Phi^{-1/2}\kappa^2\epsilon^{-1}) \overset{(i)}{=} \widetilde{O}(L_1 L_2^{-1/2}\kappa^{1/2}\epsilon^{-1})$$

where (i) uses $L_\Phi = L_2(1+\kappa)^3 = \mathcal{O}(L_2\kappa^3)$. If $\|\nabla_1 f(x_t, y_{t+1})\| > 4L_1^2\kappa^2\eta_x$, based on Lemma G.2, the number of Hessian-vector product computations is proportional to $K = \mathcal{O}(\kappa)$, whose order is not larger than the above $\widetilde{O}(L_1^2 L_2^{-1/2}\kappa^{3/2}\epsilon^{-1})$ since $\epsilon \leq L_1^2 L_2^{-1/2}\kappa^{1/2}$. Based on Lemma G.3, Algorithm 3 is implemented once with the total number of Hessian-vector product computations proportional to

$$KN' = \mathcal{O}\Big(\frac{L_1^2\kappa^2}{L_\Phi^2\epsilon'^2}\Big)\widetilde{O}(\kappa) = \widetilde{O}\Big(\frac{L_1^2\kappa^3}{L_\Phi^2(\epsilon/\sqrt{L_\Phi})^2}\Big) = \widetilde{O}\Big(\frac{L_1^2\kappa^3}{L_\Phi(\epsilon)^2}\Big) = \widetilde{O}(L_1^2 L_2^{-1}\epsilon^{-2}).$$

As a result, the total number of Hessian-vector product computations in Algorithm 4 is

$$T\widetilde{O}(L_1 L_2^{-1/2}\kappa^{1/2}\epsilon^{-1}) + \widetilde{O}(L_1^2 L_2^{-1}\epsilon^{-2})$$

$$= \mathcal{O}(\sqrt{L_\Phi}\epsilon^{-3})\widetilde{O}(L_1 L_2^{-1/2}\kappa^{1/2}\epsilon^{-1}) + \widetilde{O}(L_1^2 L_2^{-1}\epsilon^{-2})$$

$$= \widetilde{O}(\sqrt{L_2\kappa^3}L_1 L_2^{-1/2}\kappa^{1/2}\epsilon^{-4}) + \widetilde{O}(L_1^2 L_2^{-1}\epsilon^{-2})$$

$$= \widetilde{O}(L_1\kappa^2\epsilon^{-4}),$$

where (i) uses $\epsilon \leq \frac{L_2\kappa^2}{L_1}$ to absorb the term $\widetilde{O}(L_1^2 L_2^{-1}\epsilon^{-2})$. $\qquad\square$

## H   Experiment Details

In this section, we present the details of both synthetic minimax problem and the neural network simulation.

### H.1   Details of Synthetic Minimax Problem

In this section we aim to solve the problem below:

$$\min_{\mathbf{x}\in\mathcal{R}^3} \max_{\mathbf{y}\in\mathcal{R}^2} \frac{1}{N}\sum_{i=1}^{N}\Big[w(x_3) - \frac{y_1^2}{40} + A_i x_1 y_1 - \frac{5y_2^2}{2} + B_i x_2 y_2\Big] \tag{105}$$

where $A_i$ and $B_i$ are independent random variables from uniform distribution range from 0.5 to 1.5 and $w(x_3)$ has the exact form below :

$$w(x) = \begin{cases} \sqrt{\epsilon}(x + (L+1)\sqrt{\epsilon})^2 - \frac{1}{3}(x + (L+1)\sqrt{\epsilon})^3 - \frac{1}{3}(3L+1)\epsilon^{3/2}, & x \leq -L\sqrt{\epsilon}; \\ \epsilon x + \frac{\epsilon^{3/2}}{3}, & -L\sqrt{\epsilon} < x \leq -\sqrt{\epsilon}; \\ -\sqrt{\epsilon}x^2 - \frac{x^3}{3}, & -\sqrt{\epsilon} < x \leq 0 \\ -\sqrt{\epsilon}x^2 + \frac{x^3}{3}, & 0 < x \leq \sqrt{\epsilon} \\ -\epsilon x + \frac{\epsilon^{3/2}}{3}, & \sqrt{\epsilon} < x \leq L\sqrt{\epsilon}; \\ \sqrt{\epsilon}(x - (L+1)\sqrt{\epsilon})^2 + \frac{1}{3}(x - (L+1)\sqrt{\epsilon})^3 - \frac{1}{3}(3L+1)\epsilon^{3/2}, & L\sqrt{\epsilon} \leq x. \end{cases} \tag{106}$$

and we set $\epsilon = 0.01$ and $L = 5$ in our experiment. Through simple computing we can calculate that:

$$\Phi(x) = w(x_3) + 10\Big(\frac{x_1}{N}\sum_{i=1}^N A_i\Big)^2 + \frac{1}{10}\Big(\frac{x_2}{N}\sum_{i=1}^N B_i\Big)^2 \tag{107}$$

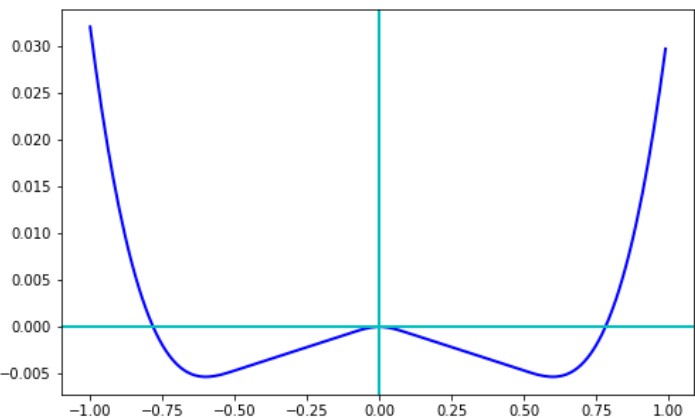

Figure 9: Figure of the w-shaped function $w(x)$.

## H.2 Details of Neural Network Simulation

The network model we use is a convolutional neural network that consists of two convolution blocks followed by two fully connected layers. Specifically, each convolution block contains a convolution layer, a max-pooling layer with stride step 2, and a ReLU activation layer. The convolution layers in the two blocks have 1, 10 input channels and 10, 20 output channels, respectively, and both of them have kernel size 5, stride step 1 and no padding. The two fully connected layers have input dimensions 320, 50 and output dimensions 50, 10, respectively.

## H.3 Details of GDN/TGDA/FR/CN

We apply a slight variant of GDN (Zhang et al., 2021) to the equivalent minimax optimization problem $\max_y \min_x -f(x,y)$ with 20 gradient ascent steps $y_{t+1} = y_t + 0.01\nabla_2 f(x_t, y_t)$ followed by Newton descent step $x_{t+1} = x_t - (\nabla_{11}f)^{-1}(\nabla_1 f)(x_t, y_{t+1})^3$. $s = (\nabla_{11}f)^{-1}(\nabla_1 f)(x_t, y_{t+1})$ for the update of $x$ has analytical solution for synthetic simulation. For adversarial deep learning, we obtain $s$ by solving the equivalent optimization problem $\max_s F_1(s) := \frac{1}{2}s^\top \nabla_{11}f(x_t, y_{t+1})s + \nabla_1 f(x_t, y_{t+1})^\top s$ via 30 gradient ascent steps with learning rate 0.002 and early termination rule $\|\nabla F_1(s)\| < 0.001$. Batchsizes 512 and 100 are used to compute all the stochastic gradients and Hessians for synthetic simulation and adversarial deep learning respectively.

For TGDA, we apply 10, 20 gradient ascent steps on $y$ with learning rates $0.01, 0.1$ for synthetic simulation and adversarial deep learning respectively, followed by 1 Newton-type update step $x_{t+1} =$

---

[3]Compared with the original GDN (Zhang et al., 2021), we switch the roles between $x$ and $y$, since $f(x,\cdot)$ is strongly concave for which gradient ascent is sufficiently fast.

$x_t - 0.01\big(\nabla_1 f - \nabla_{12} f (\nabla_{22} f)^{-1} \nabla_2 f\big)(x_t, y_{t+1})$ on $x$. $s = (\nabla_{22} f)^{-1}(\nabla_2 f)(x_t, y_{t+1})$ has analytical solution for synthetic simulation. For adversarial deep learning, we obtain $s$ by solving the equivalent optimization problem $\max_s F_2(s) := \frac{1}{2} s^\top \nabla_{22} f(x_t, y_{t+1}) s - \nabla_2 f(x_t, y_{t+1}) s$ via 30 gradient ascent steps with learning rate 0.1. Batchsizes 512 and 100 are used to compute all the stochastic gradients and Hessians for synthetic simulation and adversarial deep learning respectively.

For FR, we apply 20 gradient ascent steps on $y$ with learning rates 0.01, 0.1 for synthetic simulation and adversarial deep learning respectively, followed by 1 Newton-type update step $x_{t+1} = x_t - 0.01 \nabla_1 f(x_t, y_{t+1}) - \alpha(\nabla_{11} f)^{-1}(\nabla_{12} f)(\nabla_2 f)(x_t, y_{t+1})$ on $x$. $(\nabla_{11} f)^{-1}(\nabla_{12} f)(\nabla_2 f)(x_t, y_{t+1})$ has analytical solution for synthetic simulation. For adversarial deep learning, we obtain $s$ by solving the equivalent optimization problem $\max_s F_2(s) := \frac{1}{2} s^\top \nabla_{11} f(x_t, y_{t+1}) s - (\nabla_{12} f)(\nabla_2 f)(x_t, y_{t+1}) s$ via 30 gradient ascent steps with early termination rule $\|\nabla F(s)\| < 0.001$ and learning rate 0.002. Batchsizes 512 and 100 are used to compute all the stochastic gradients and Hessians for synthetic simulation and adversarial deep learning respectively.

CN algorithm follows the original update rule in (Zhang et al., 2021) that $x_{t+1} = x_t - G(x_t, y_t)^{-1}(\nabla_1 f)(x_t, y_t)$, $y_{t+1} = y_t - (\nabla_{22} f)^{-1}(\nabla_2 f)(x_{t+1}, y_t)$ where $G(x, y) = \big[\nabla_{11} f - \nabla_{12} f(\nabla_{22} f)^{-1} \nabla_{21} f\big](x, y)$. For synthetic experiment, $s = G(x_t, y_t)^{-1} \nabla_1 f(x_t, y_t)$ has analytical solution and we compute $s = \big(G(x_t, y_t) - 0.5 I\big)^{-1} \nabla_1 f(x_t, y_t)$ instead to avoid singularity. For adversarial deep learning, to compute $s = G(x_t, y_t)^{-1} \nabla_1 f(x_t, y_t)$, we apply GDA with 50 iterations to the minimax optimization problem $\min_s \max_v (\nabla_1 f(x_{t+1}, y_t)^\top s + \frac{1}{2} s^\top \nabla_{11} f(x_{t+1}, y_t) s + s^\top \nabla_{12} f(x_{t+1}, y_t) v + \frac{1}{2} v^\top \nabla_{22} f(x_{t+1}, y_t) v$, where each iteration has 1 gradient ascent step on $v$ with learning rate 0.1 followed by 1 gradient descent step on $s$ with learning rate 0.002. Batchsizes 512 and 100 are used to compute all the stochastic gradients and Hessians for synthetic simulation and adversarial deep learning respectively.

