# OpenReview forum: "A Cubic Regularization Approach for Finding Local Minimax Points in Nonconvex Minimax Optimization"
_TMLR — Accepted by TMLR_

### Review · Reviewer_YWBv · 2022-12-19

**Summary Of Contributions:**

This work analyzes the global convergence of cubic-regularized (CR) methods to strict local minimax points with deterministic and finite-sum stochastic settings. It also conducts experiments on synthetic data and robust classification to verify the convergence of CR methods.


--post rebuttal--

Thank you for your effort in the revision and for addressing my comments carefully. After checking with our reviews, it is clear that the current paper does not improve Luo et al 2022 in terms of the theoretical convergence rate, in the deterministic setting. For the stochastic setting, it provides a new convergence bound, which is not available in Luo et al. I like the new experiments added and the comprehensive comparison with existing methods. The proposed method seems to be comparable with (I)MCN and CN/GDN but is not always better.

I agree that the comparison table should be moved to the main paper.

**Audience:**

Yes

**Broader Impact Concerns:**

This is theoretical work, and I don't have broader impact concerns.

**Claims And Evidence:**

Yes

**Requested Changes:**

See strengths and weaknesses.

**Strengths And Weaknesses:**

Strengths:
1. It has a lot of references regarding recent work and compares in detail the differences with (Luo and Chen 2021).
2. It analyzes the exact gradient computations of solving cubic regularization.
3. Both deterministic and stochastic cases are considered with detailed analysis.
4. The authors conduct experiments to verify the practicability of CR methods in robust classification.
5. The paper is clearly written with detailed proof steps.
6. A new potential function is proposed to analyze the convergence.

Weaknesses:
1. One of my main concerns is the comparison with existing literature. My suggestion would be to list all related methods in a table, including algorithm, setting, reference, and convergence rates.
* a) with (Luo and Chen 2021): both papers analyze the CR methods for nonconvex-strongly-concave (NCSC) functions. What is the main advantage of the current paper? Does this work achieve faster convergence? Or is it the stochastic part? Or is the difference in the algorithmic design?
* b) with (HZZ 2020 [1]): HZZ 2020 also studies CR method for saddle points with strongly-convex-strongly-concave (SCSC) functions, which are special cases of NCSC. Could the authors provide a comparison wrt the convergence rate?
* c) with other methods (TGDA/FR/GDN/CN, Zhang et al 2021): although these methods study local convergence, at least some experimental results should be shown to compare the convergence.
2. The main solutions this paper aims to find are strict local minimax points. With strong concavity of y given x, finding the best response can be efficient. Therefore, the main problem reduces to finding stationary points that have positive curvatures. There have already been such work in the literature, such as [2]. In my opinion, any such work could be applied to the given problem with guaranteed convergence. The current work seems more like a simple combination of Gradient Ascent of y and CR for the envelope function.
3. Minor issues:
* a) Definition 1 is not quite intuitive. The authors are suggested to use the clearer equivalent version. See Lemma 16 of https://arxiv.org/pdf/1902.00618.pdf and Prop 3.6 from [3]. In fact, the case when $f(x, y)$ is *not* strongly concave in $y$ is more common and more interesting for practical use.
* b) Fact 1 is known, for example, in [4].
* c) It's better to add section numbers to equations.
* d) In Figure 3/4, the comparison with GDA is not fair. The $x$-axis should be the number of gradient calls rather than the number of epochs because CR takes more gradient steps.
* e) Theorem 1 only finds stationary points with at most small negative curvatures, but this condition is not Fact 1.
* f) Algorithms 4 and 5 should be brought to the main text for easy reference.



[1] Cubic Regularized Newton Method for Saddle Point Models: a Global and Local Convergence Analysis, https://arxiv.org/pdf/2008.09919.pdf.

[2] Escape saddle points by a simple gradient-descent based algorithm, NeurIPS 2021, https://openreview.net/pdf?id=lEf52hTHq0Q.

[3] Optimality and Stability in Non-Convex Smooth Games, https://jmlr.org/papers/volume23/20-918/20-918.pdf

[4] Some local properties of minimax problems, https://www.sciencedirect.com/science/article/abs/pii/0041555374901074

---

> ### Author Response · Authors · 2023-01-24
> **Authors' reply to Reviewer YWBv**
>
> Thank you very much for reviewing our manuscript and providing valuable feedback. Below is a response to the review comments. We have submitted a revised version with all revisions marked in "red". Please let us know if further clarifications are needed.
>
> **Q1:** List all related methods in a table, including algorithm, setting, reference, and convergence rates.
>
> **A:** Thanks for your great suggestion. We have added the table in Appendix I of the revision.
>
> **Q1a:** Compared with (Luo and Chen 2021), what is the main advantage of the current paper? Does this work achieve faster convergence? Or is it the stochastic part? Or is the difference in the algorithmic design?
>
> **A:** Great question. One big difference is that we develop stochastic algorithm. Also, we made the deterministic algorithm more practical by removing cubic objective evaluation in the cubic-solver and allowing more adaptive approximation error bounds (eqs. (4)-(5) in our paper) for gradient ascent steps. The practical efficiency is demonstrated by the newly added experiments in our revision. We have added this explanation in Section 1.2 of our revision.
>
> **Q1b:** Compare convergence rate with (HZZ 2020).
>
> **A:** Thanks for your great suggestion. We have added convergence rate of (HZZ 2020) to Section 1.2 in the revision. Since (HZZ 2020) studies convex-concave minimax optimization and does not provide the number of gradient or Hessian-vector product evaluations we care about, we did not make direct comparison with (HZZ 2020).
>
> **Q1c:** Compared with other methods TGDA/FR/GDN/CN (Zhang et al 2021) in experiment.
>
> **A:** Thanks for your great suggestion. We have added the TGDA/FR/GDN/CN to the experiments in the revision and our algorithm shows practical advantage.
>
> **Q2:** The current work seems more like a simple combination of Gradient Ascent of y and CR for the envelope function.
>
> **A:** We agree with the reviewer's view point. However, as we discussed in the paper, it turns out that applying CR to the envelope function leads to a very special and complex cubic subproblem. To solve this problem, we proposed a new minimax reformulation of it and developed GDA-type solvers with provable convergence guarantee.
> In addition, our algorithm design is adaptive and practical, e.g., it adopts adaptive approximation error for gradient ascent of y (see eqs. (4)-(5)) and avoids the need to  evaluate the cubic objective function in the cubic-solvers.
>
>
> **Q3a:** Use clearer version of Definition 1 by using Lemma 16 of https://arxiv.org/pdf/1902.00618.pdf and Prop 3.6 from [3].
>
> [3] Optimality and Stability in Non-Convex Smooth Games, https://jmlr.org/papers/volume23/20-918/20-918.pdf
>
> **A:** Thanks for your great suggestion. We have adopted the clearer definition that you provided in the revision.
>
> **Q3b:** Fact 1 is known, for example, in [4].
> [4] Some local properties of minimax problems, https://www.sciencedirect.com/science/article/abs/pii/0041555374901074.
>
> **A:** Thanks for pointing that out. We have cited [4] in Section 3.1 of the revision.
>
> **Q3c:** It's better to add section numbers to equations.
>
> **A:** Thanks for your great suggestion. We have added section numbers to equations in the revision.
>
> **Q3d** In Figure 3/4, the x-axis should be the number of gradient calls.
>
> **A:** Thanks for your great suggestion. We addeed figures with number of oracle calls (i.e., number of gradient and Hessian-vector product evaluations) as x-axis in the revision.
>
> **Q3e** Theorem 1 only finds stationary points with at most small negative curvatures, but this condition is not Fact 1.
>
> **A:** Theorem 1 implies that $\lambda_{\min}\big(\nabla^2\Phi(x_{T'})\big)\ge -\sqrt{33L_{\Phi}}\epsilon$, which converges to $\lambda_{\min}\big(\nabla^2\Phi(x_{T'})\big)\ge 0$ in Fact 1 as $\epsilon\to +0$. Such form of small negative curvature is commonly used in convergence results to local optimal points. (e.g., eq. (1) of [2], Definition 3 of [5], etc.)
>
> [2] Escape saddle points by a simple gradient-descent based algorithm, NeurIPS 2021. https://openreview.net/pdf?id=lEf52hTHq0Q.
>
> [5] Luo, L., Li, Y., \& Chen, C. (2022). Finding Second-Order Stationary Points in Nonconvex-Strongly-Concave Minimax Optimization. In Advances in Neural Information Processing Systems. https://openreview.net/pdf?id=Jb-d9fZX14
>
> **Q3f** Algorithms 4 and 5 should be brought to the main text for easy reference.
>
> **A:** Thanks for your great suggestion. We have moved them to the main text in our revision.

---

> > ### Author Response · Authors · 2023-01-29
> > **Authors' reply to Reviewer YWBv (2)**
> >
> > Thank Reviewer YWBv for your suggestion.
> > We have moved tables to the main text.

---

### Review · Reviewer_VJjz · 2022-12-29

**Summary Of Contributions:**

The article investigates the minimax optimization problem, for smooth non-convex-strongly-concave function.
Its goal is to find a local minimax point that satisfies second-order non-degeneracy conditions.

Authors reformulate the objective as a minimization problem of an envelope function $\Phi$ and show the properties of the reformulation mapping.

Due to the structure of the reformulated envelope function, known methods are not applicable. Consequently, authors propose an algorithm Cubic-LocalMinimax for solving the problem.
Cubic-LocalMinimax tackles the structure of the problem by using (implicit) second-order cubically regularized Newton method. Authors explain how to solve implicit subproblem, they propose to utilize (first-order) Gradient Ascent method for solving the subroutine.

Authors extend their method to allow using stochastic estimates of gradients and Hessians (for empirical risk minimization) and provide high-probability bounds for the convergence as (Harvey et al., 2019).

Authors show global convergence of the proposed method in theory and experimentally show practical advantages compared to a Gradient Descent Ascent.

**Questions:**

- Regarding GDA-Cubic Solver: When gradients are small, GDA-Cubic Solver uses first-order GDA? For small gradient case, wouldn’t it make more sense to use a second-order methods instead?

**Audience:**

Yes

**Broader Impact Concerns:**

Work is of theoretical nature. I don't see any concerns.

**Claims And Evidence:**

No

**Requested Changes:**

**Critical adjustments:**
- Add concise technical comparison of the literature (in theory).
- Experimentally compare proposed algorithms to similar algorithms (not just first-order GDA).
- Clarify assumptions in the stochastic setup (main paper) to obtain high-probability bounds.
- Revise paper to improve presentation.

**Details/suggestions:**
- Footnote 2 & citation:  (Luo et al., 2021) had revision on arXiv on November 9, 2022
    - it doesn’t have eq. (36)-(37)
    - it has 3 authors
- Name of algorithm _Inexact Cubic-LocalMinimax_ has inconsistent delimiters: [space], "-", [no space]


**Strengths And Weaknesses:**

**Strengths**:
- Authors show reformulation that local minimax points are equivalent to a second-order stationarity points of an envelope function $\Phi$.
Reformulation simplifies bivariate minimax optimization to an univariate minimization problem with specific structure. Envelope function $\Phi$ is designed to preserve Lipschitz continuity and shows insights into minimax optimization.

- Convergence rates and proofs are sound. I checked main paper and whole Appendix except Appendix G.

- Even though there are some confusing parts in the paper (see weaknesses), I liked reading the paper. I encourage authors to revise those, and submit again.

------

**Weaknesses:**
- **High probability bounds** in the stochastic setup (from Harvey et al., 2019) have an important restriction on heterogeneity of functions $f_i$; they needs variance of stochastic gradients to be bounded. With restriction on functions $f_i$ to be almost homogeneous, results in the stochastic setup are much weaker.
This restriction is correctly reported in the Appendix F, but it is not mentioned at all in the main paper.

- **Literature comparison** is lacking.
    - Literature section (1.2 Other Related Work): While literature comparison lists multiple papers and their objectives, it is unclear how different results do they present.\
 E.g., ``Yang et al. (2020) studied an alternating gradient descent-ascent (AGDA) algorithm in which the gradient ascent step uses the current variable $x_{t+1}$ instead of $x_t$.” What is their resulting convergence rate? How did usage of $x_{t+1}$ influence their analysis?
    - While theoretical comparison to other algorithms is in paper, this information is scattered over paper and is not complete. I'd appreciate table (or section) with concise technical comparison. Possible columns: method name, reference, assumptions, oracle type, convergence rate.
    - Paper proposes algorithm and compares it to GDA. While this comparison gives some insight, GDA doesn’t seem to be most relevant baseline (first-order method vs second-order method comparison). There is no experimental comparison to any other similar (second-order) methods.
    - Some relevant methods to compare to: Follow-the-Ridge (Wang et al., 2020), Minimax Cubic-Newton (Luo et al., 2021), Inexact Minimax Cubic-Newton (Luo et al., 2021), Cubic-Solver (Luo et al., 2021), Final-Cubic-Solver (Luo et al., 2021), Cubic-Solver (Tripuraneni et al., 2018).

----
 **Presentation issues:**

- Paper narrative seems discontinuous.\
E.g., Propositions 1,2 and Fact 1 include very similar results, yet they are (unnecessarily) separated by section break and paragraphs of text.\
E.g., first two paragraphs of Section 3.2 are literature comparison, and can be included with the rest of literature comparison (section 1.2).

- On page 9, there is detailed comparison to work (Luo et al., 2021), but it lacks technical details:
    - Does replacing constant precision approximations by adaptive ones, lead to improvement theory/practice?
    - `` [their solver] requires more Hessian-vector product computation per iteration!" How much more? Is it significant?
    - `` [their algorithm MCN and their solver] need to additionally track the approximate objective function value of the cubic subproblem, which requires additional computation." How much more computation is needed? Is it significant?

- Experiments section:
    - Left and right plots in Figures 1, 2 are very similar (time is almost proportional to samples). What may be more interesting is to plot performance per-iteration.
    - What is performance compared to other second-order minmax algorithms?
    - I’d appreciate if stochastic versions would have been compared to a full batch case -- to highlight how much time/samples were actually saved by introducing stochasticity.

---

> ### Author Response · Authors · 2023-01-24
> **Authors' reply to Reviewer VJjz (1)**
>
> Thank you very much for reviewing our manuscript and providing valuable feedback. Below is a response to the review comments. We have submitted a revised version with all revisions marked in "red". Please let us know if further clarifications are needed.
>
> **Q1:** Regarding GDA-Cubic Solver: When gradients are small, GDA-Cubic Solver uses first-order GDA? For small gradient case, wouldn’t it make more sense to use a second-order methods instead?
>
> **A:** Good point. We want to clarify that the objective of the GDA-Cubic Solver is the cubic problem $\min_s \phi(s):=g^{\top}s+\frac{1}{2}s^{\top}As+\frac{1}{6\eta_x}||s||^3$ (eq. (7) in our paper), which itself originates from the second-order CR method. The CR objective $\phi(s)$ has been proved to be "almost convex" such that simple gradient descent can solve it efficiently [2,3], so [2,3] uses gradient descent to solve this problem. In our work, $\nabla \phi(s)$ is not directly accessible, so we equivalently rewrite $\min_s \phi(s)$ as the minimax optimization problem $\min_s \max_v \widetilde{\phi}(s,v):=g^{\top}s+\frac{1}{2}s^{\top}H_{11}s+s^{\top}H_{12}v+\frac{1}{2}v^{\top}H_{22}v+\frac{1}{6\eta_x}||s||^3$, which is strongly concave with respect to $v$ and ``almost convex'' with respect to $s$. Therefore, as we proved in the paper, simple first-order GDA method can efficiently solve this well-shaped minimax optimization problem.
>
> [2] Carmon, Y., \& Duchi, J. (2019). Gradient descent finds the cubic-regularized nonconvex Newton step. SIAM Journal on Optimization, 29(3), 2146-2178..
>
> [3] Tripuraneni, N., Stern, M., Jin, C., Regier, J., \& Jordan, M. I. (2018). Stochastic cubic regularization for fast nonconvex optimization. Advances in neural information processing systems, 31.
>
> **Q2:** Report in the main text the restriction that variance of stochastic gradients to be bounded.
>
> **A:** Thanks for your great suggestion. We have added that restriction to the main text of the revised version.
>
> **Q3:** In related work section, it is unclear how different results do the cited papers present. E.g., "Yang et al. (2020) studied an alternating gradient descent-ascent (AGDA) algorithm in which the gradient ascent step uses the current variable $x_{t+1}$ instead of $x_t$.” What is their resulting convergence rate? How did usage of $x_{t+1}$ influence their analysis?
>
> **A:** Thanks for pointing that out. In general, the major difference in their results and ours is that most of the works mentioned in the related work section are first-order algorithms that only converges to stationary point, whereas our algorithm is second-order that converges to local minimax point. We have added that clarification to the related work section of the revision. More specifically, we have added convergence rates of these algorithms to Table 1 in Appendix I of the revised paper. As to the analysis (e.g. how does $x_{t+1}$ affect the analysis), we did not explain that in the related work section since the works there have little overlap with ours due to the above major difference.
>
> **Q4:**  I'd appreciate table (or section) with concise technical comparison. Possible columns: method name, reference, assumptions, oracle type, convergence rate.
>
> **A:** Thanks for your great suggestion. We have added the table to the introduction Appendix I of the revised version.
>
> **Q5:** There is no experimental comparison to any other similar (second-order) methods, such as Follow-the-Ridge (Wang et al., 2020), Minimax Cubic-Newton (Luo et al., 2021), Inexact Minimax Cubic-Newton (Luo et al., 2021), Cubic-Solver (Luo et al., 2021), Final-Cubic-Solver (Luo et al., 2021), Cubic-Solver (Tripuraneni et al., 2018).
>
> **A:** Thanks for your great suggestion, We have added the comparison to Follow-the-Ridge, Minimax Cubic-Newton (MCN) and Inexact Minimax Cubic-Newton (IMCN). We note that both the Cubic-Solver and the Final-Cubic-Solver are subroutines of IMCN for solving cubic-regularization subproblems, so they are not comparable.
>
> **Q6:** Propositions 1, 2 and Fact 1 include very similar results, yet they are (unnecessarily) separated by section break and paragraphs of text.
>
> **A:** Great point. In the revision, we have merged Proposition 1 into Proposition 2, so that this combined new Proposition 1 and Fact 1 are both in Section 3.1. Thanks for the suggestion.
>
> **Q7:** The first two paragraphs of Section 3.2 are literature comparison, and can be included with the rest of literature comparison (Section 1.2).
>
> **A:** Great suggestion. In the revision, we have moved most of the literature comparison in Section 3.2 to Section 1.2. We keep the works on second-order algorithms for minimization to motivate the design of our second-order algorithm for minimax optimization.

---

> > ### Author Response · Authors · 2023-01-24
> > **Authors' reply to Reviewer VJjz (2)**
> >
> > **Q8:** On page 9, compared with (Luo et al., 2021), does replacing constant precision approximations by adaptive ones, lead to improvement theory/practice?
> >
> > **A:** Good question. In the revision, right after eqs. (4) and (5), we have explained that when the increment $s_t=x_t-x_{t-1}$ satisfies $||s_t||=\mathcal{O}(1)\gg \mathcal{O}(\epsilon')$, which usually occurs in most iterations, our algorithm requires only $\mathcal{O}(1)$ gradient ascent steps, which is much fewer than $\mathcal{O}(\ln (1/\epsilon'))$ required by the MCN algorithm (Luo et al., 2021). In the other iterations where $||s_t||\le \mathcal{O}(\epsilon')$, our algorithm requires $\mathcal{O}(\ln (1/\epsilon'))$ gradient ascent steps of the same order as that of MCN. Combining the two cases, our algorithm is more practical and outperforms MCN, as also demonstrated by the experimental results in adversarial deep learning.
> >
> > **Q9:** In "[their solver] requires more Hessian-vector product computation per iteration!" How much more? Is it significant?
> >
> > **A:** Good question. By carefully checking eqs. (32-33) in (Luo et al. 2021) and our eq. (60), we found that both use 2 Hessian-vector product computations in each iteration for computing $As$ where $A:= H_{11}-H_{12}H_{22}^{-1}H_{21}$ with $H_{k\ell}=\nabla_{k\ell} f(x_t,y_{t+1}); k,\ell\in\{1,2\}$. We have removed this statement.
> >
> > **Q10:** In "[their algorithm MCN and their solver] need to additionally track the approximate objective function value of the cubic subproblem, which requires additional computation." How much more computation is needed? Is it significant?
> >
> > **A:** Good question. The approximate objective function value as the output of the cubic solver in (Luo et al. 2021) requires to compute $As$ again where $A:= H_{11}-H_{12}H_{22}^{-1}H_{21}$ with $H_{k\ell}=\nabla_{k\ell} f(x_t,y_{t+1}); k,\ell\in\{1,2\}$, which takes $\widetilde{O}(\sqrt{\kappa})$ additional Hessian-vector product computations.
> >
> > **Q11:** Left and right plots in Figures 1, 2 are very similar (time is almost proportional to samples). What may be more interesting is to plot performance per-iteration.
> >
> > **A:** Thanks for pointing that out. We have replaced samples with the number of epochs as x-axis (each update of min-player variable $x$ is considered as an epoch).
> >
> > **Q12:** Compare stochastic algorithm with a full batch version to highlight how much time/samples were actually saved by introducing stochasticity.
> >
> > **A:** Great suggestion. In Figure 1, we have added the full batch version ($B=1000$) of our algorithm and found that the full batch version takes fewer epochs but more time to converge than the stochastic versions.
> >
> > **Q13:** Footnote 2 and citation: (Luo et al., 2021) had revision on arXiv on November 9, 2022. It doesn’t have eq. (36)-(37). It has 3 authors
> > Name of algorithm Inexact Cubic-LocalMinimax has inconsistent delimiters: [space], "-", [no space].
> >
> > **A:** Thanks for your great suggestion. We have revised all these in the revision.

---

> > > ### Comment · Reviewer_VJjz · 2023-01-24
> > > **Response to rebuttal**
> > >
> > > I have limited time this week, so I cannot do a detailed editing check before the recommendation deadline.
> > > I read the paper briefly, but I haven’t checked whether all of the requested changes were implemented.
> > >
> > > ---
> > >
> > > **Re answers to _Q3_, _Q4_, and _Appendix I_.**
> > >
> > > I appreciate adding the comparison table. However, it is overcrowded and I have a very hard time understanding it. It takes significant effort to tell which rates are better than others.
> > > Even though algorithms with different settings and metrics can’t be compared directly, one can still omit small differences to make them comparable (but mention the change in a footnote).
> > >
> > >
> > > **Strong suggestions for simplification/clarification of _Appendix I_:**
> > > - re stochastic setup: mentioned papers allow different levels of heterogeneity. E.g., (Xu 2020b) allows gradient variance bounded by $\sigma$ instead of $1$, which is much better. This is an important assumption, which affects practical usability. Can you specify it in comparison? If there are algorithms having comparable rates while allowing much more heterogeneity, you may want to relegate stochastic setup.
> > > - re metrics: it is unclear what the difference between them is and how do they affect convergence. If the difference is important, it’d help to have it described. If it is not important, it may be omitted.
> > > - re gradient column: $ \widetilde {\mathcal O}$ notation hides logarithmic factors; why is it not used in all rows?
> > > - re setting column: this column groups $3$ assumptions (stochasticity, convexity, concavity). Translating text abbreviations takes effort. Why not mark triplets (stoch. variance, convexity, concavity) as ($0$/$1$/$\sigma$, ‘-‘/$0$/$L$/PL, ‘-‘/$0$/$\mu$/PL) to make it explicit?
> > > - re row split: rows are hard to navigate. Some rows include $2$ different setups (with different rates), and some rows include $2$-line algorithm names.
> > > - re rows: I find myself jumping between rows a lot. Rows are ordered by reference. Why not groups them by something more meaningful, e.g., setting or metric?
> > >
> > > **Other suggestions for _Appendix I_:**
> > > - maybe partition table to deterministic/stochastic table
> > > - maybe remove ``Hv column” and create a separate table from these rows.
> > > - Theorems and Corollary numbers in references (for other papers) add non-crucial text in the already overcrowded table. Maybe remove them to simplify.
> > > - *ZO-VRGDA*, *SREDA*, *ZO-SREDA* rows match, setting, metric, rate. Maybe put them in the same row.
> > > - you can use bold/colors to highlight things
> > > - you can use table remarks (see latex package 'threeparttable')
> > >
> > > After cleaning, I believe that this table is needed in the main paper.
> > >
> > > ---
> > > **Other comments:**
> > > - Figures: It is quite hard to distinguish lines in figures. Can you add markers?
> > > - Algorithms $1$ and $2$ have equations centered while others do not. Is it intentional?
> > > - I believe that the presentation can be further improved. E.g., Appendix I, heterogeneity assumptions in literature, etc.

---

> > > > ### Author Response · Authors · 2023-01-24
> > > > **Authors' reply to Reviewer VJjz (3)**
> > > >
> > > > We thank the reviewer for the quick response. We will update the manuscript according to the reviewer’s suggestion ASAP by this week.

---

> > > > > ### Author Response · Authors · 2023-01-29
> > > > > **Authors' reply to Reviewer VJjz (4)**
> > > > >
> > > > > We thank the reviewer VJjz for your thoughtful suggestions and we have edited the figures, the tables in Appendix I and algorithm boxes based on your suggestions.
> > > > >
> > > > > About "re gradient column": We are aware that $\widetilde{\mathcal{O}}$ hides logarithmic factors, and use $\widetilde{\mathcal{O}}$ or $\mathcal{O}$ exactly following the original convergence results.
> > > > >
> > > > > About centered equations: Those centered equations are in the align environment (centered) in order to cite them in the text. Some equations were in-line and aligned left since we did not cite them. Now we also put them in the center of the align environment to make them consistent. Thanks for your suggestion.

---

### Review · Reviewer_jvV1 · 2023-01-04

**Summary Of Contributions:**

In this two-part paper, the authors provide the analysis of a Cubic regularization approach for finding local minimax points in the general bifurcation min-max optimization problem. More precisely the starting point of the authors is the observation that standard gradient descent-ascent approach typically converges to stationary points for nonconvex minimax optimization, which are suboptimal compared with local minimax points. In order to tackle this gap, the authors turn their attention inspired by the convex-concave analogue on Cubic Regularization techniques. Their first contribution is to provide an equivalent characterization of the local minimax points as the second-order stationary points of a certain envelope function. This second-order envelope function actually is based on the standard approach of Shur's Complement of the objective bifurcation. Having established this equivalence, initially they provide a deterministic and stochastic best-iterate convergence for value/gradient/hessian oracle and finite-sum case. Following the analogue of non-convex single agent optimization, the authors describe a non-exact hessian vector approximation scheme to avoid the complete computation of Hessian matrix.



**Audience:**

No

**Broader Impact Concerns:**

Not-applicable

**Claims And Evidence:**

Yes

**Requested Changes:**

Specific suggestions for revision
Below, I am providing some specific suggestions that the authors might want to
take into account in revising the paper. In order to increase the readability of the appendices,I would strongly suggest that the authors will provide a ''foreshadowing'' paragraph before every proof. In this short period they could explain the intuition behind the proofs in order to increase the readability of them.
My expectation for a journal publication is  the manuscript which is self-contained, with correct grammar and syntax, and it offers the opportunity not only to proof-read but obtain important intuition behind the techniques. If these basics are not explained to readers unfamiliar with technique, the paper intuition is almost impossible to follow.

In the current form, experiments are more confusing than  convincing for the contribution.
In practice single loop optimization schemes work faster (ceteris paribus any system details).
So it would be beneficial to understand the applicability of such method to real practice scenario.
Such kind of experiments are more valuable than standard simulations with "almost artificial data".

Basics of convergence guarantees. Given that the basic aim of the paper is to
provide a min-max CR method for non-convex optimization, the basic notions for convergence should appear early on in the main text (not in an appendix) and be clearly explained to the reader. Also, I would suggest that the authors provide a more intuitive discussion of the characterization lemma. This is the most interesting element in the draft - in my humble opinion - and it is less developed in the current version.

Finally, a couple of comments regarding the equivalence of the two
notions of the notion of min-max optimization and the existence of duality gap
would be helpful. The authors should thoroughly proofread the manuscript for
grammatical and spelling errors. It is impossible to provide an exhaustive list
here, but one mistake/issue that occurs at almost every page of the manuscript is the usage of min-max/minimax etc alternatively without drafting the differences.

**Strengths And Weaknesses:**

Even though I like the basic premise of the paper, I cannot recommend its acceptance at this point.

My main reasons for this are:

1) Incremental extension: The current work is a slight extension of an accepted NeurIPS submission
(see arxiv https://arxiv.org/pdf/2110.04814.pdf ) this year. Since TMLR is a new approach for journal submission, I will let to the (higher-order) Editors' committee to decide about this double submission issue. I just comment this fact because NeurIPS provides its own Proceedings.

2)  Lack of clarity: The main ideas of the paper are expected given the previous literature  but the presentation is fairly confusing. The solution concept of mini-max optimization in non-convex non-concave structure is certainly well-defined. However, it is important to notice the lack of duality gap. It would be important to explain why this absence is not crucial for the important applications of this concept in ML. More importantly, from the way that the results are written using the parameters $T$ and $T'$, it is impossible to understand if the authors refer to a best-iterate result (which is the expected one given the non-convexity) or last-iterate convergence to a point which satisfies a specific stopping criterion (which is computational intractable in general case). Since the readership of TMLR cannot be assumed to be necessarily well-versed in non-convex optimization, the authors should do an effort to clearly present the basic notions needed from convergence guarantees in continuous optimization, so that readers can follow clearly the intuition behind their results.

3)

a) Having said (2), it would be more interesting given the incrementally of the current submission and the accepted NeurIPS result to provide multiple additional experimental work which would verify the practicality of the result. Especially given the best iterate approach of the result, why such kind of results are important and interesting in ML community?

b) An additional issue that I would like to raise for the experiments is the suitability of number of epochs as the correct metric. It would be important to provide system-wise information about the execution of the experiment.

Strengths:
Clear scope of focus: besides of the paper’s length (due in part to many ancillary
results), it is very easy to filter which results comprise the authors’ main contributions and which ones are secondary. I strongly appreciate the well-positioned papers and the easy flow writing style of this paper.
However,  I had to read both parts (main and appendix) two times until I could safely pin-
point the authors’ main results proof ideas – a somewhat disappointing trait for a
paper which purports to provide an intuitive optimization idea.

---

> ### Author Response · Authors · 2023-01-24
> **Authors' reply to Reviewer jvV1 (1)**
>
> Thank you very much for reviewing our manuscript and providing valuable feedback. Below is a response to the review comments. We have submitted a revised version with all revisions marked in "red". Please let us know if further clarifications are needed.
>
> **Q1:** The current work is a slight extension of an accepted NeurIPS submission (see arxiv https://arxiv.org/pdf/2110.04814.pdf)
>
> **A:** We want to point out that our work is concurrent to theirs.
>
> Compared to this concurrent work, our work is different in the following three aspects. First, our algorithm adopts a very different output rule, which leads to a different algorithm design of the cubic solver as well as different convergence proof strategies. Second, we develop a GDA-based solver for solving the cubic subproblem, whereas they use a gradient-based solver with Chebyshev polynomials. Moreover, we develop a stochastic version of Cubic-LocalMinimax and analyze its sample complexity, which to our knowledge has not been studied in the existing literature.
>
> **Q2a:** Why is lack of duality gap not crucial for ML applications? I suggest to comment on the equivalence of the two notions of min-max optimization and the existence of duality gap.
>
> **A:** Good question. Zero duality gap is required for simultaneous games where the min-player and the max-player act simultaneously [1]. In fact, [1] pointed out that most ML applications are sequential games where min-player and max-player act sequentially and the sequence (i.e., who plays first) is crucial and pre-specified. For example, in adversarial training, classifier acts first and then the adversary generates an adversarial sample. In GAN training, generator acts first followed by discriminator. Moreover, in sequential games, Nash equilibrium with zero duality gap is not necessarily a proper solution as that may not exist, and this motivates introducing the local minimax solution instead.
>
> We have added the above clarification to Section 2 in the revision. Specifically, if the parameter $(x,y)$ is a local Nash equilibrium of the simultaneous game, then it is also a local minimax solution of the sequential game [1].
>
> [1] Jin, C., Netrapalli, P., \& Jordan, M. (2020, November). What is local optimality in nonconvex-nonconcave minimax optimization? In International conference on machine learning (pp. 4880-4889). PMLR.
>
> **Q2b:** Do we refer to the best-iterate result (which is the expected one given the non-convexity) or last-iterate convergence?
>
> **A:** Good question. It is last-iterate convergence. Different from most of the existing last-iterate convergence results where the last iterate $T$ is prespecified, out last iterate $T'$ is determined by a termination rule and is proved to have a finite upper bound $T$. We have added this clarification to the remark right after Theorem 1 in the revision. Thanks for pointing this out.
>
> **Q3a:** Add experiments to compare with the accepted Neurips work [1]. Especially given the best iterate approach of the result, why such kind of results are important and interesting in ML community?
>
> **A:** Great question. We have added comparison with the inexact MCN (Algorithm 3 of [1]) to our experiments. Our convergence result is actually last-iterate as we explained in the response to Q2b.
>
> [1] Luo, L., Li, Y., \& Chen, C. (2022). Finding Second-Order Stationary Points in Nonconvex-Strongly-Concave Minimax Optimization. In Advances in Neural Information Processing Systems.
>
> **Q3b:** Number of epochs is not the correct metric for the experiment. Use system-wise information instead.
>
> **A:** Thanks for your great suggestion. In the revision, we have added performance comparison under various metrics such as the total number of oracle calls and overall time consumption. We also reported the system information such as memory, CPU, and GPU.
>
> **Q4:** Provide a "foreshadowing" paragraph before every proof, and a more intuitive discussion of the characterization lemma.
>
> **A:** Thank you for your great suggestion. We have added one paragraph to explain the intuition right before every proof that is not straightforward.

---

> > ### Author Response · Authors · 2023-01-24
> > **Authors' reply to Reviewer jvV1 (2)**
> >
> > **Q5:** In practice single loop optimization schemes work faster (ceteris paribus any system details). What's the applicability of such method to real practice scenario?
> >
> > **A:** Good question. Our algorithm has advantage in both number of oracle calls (i.e., the number of gradient and Hessian-vector evaluations) and time complexity experiment on adversarial deep learning, which a is widely used real practice scenario. [1-3]
> >
> > [1] Shafahi, A., Najibi, M., Ghiasi, M. A., Xu, Z., Dickerson, J., Studer, C., ... \& Goldstein, T. (2019). Adversarial training for free!. Advances in Neural Information Processing Systems, 32.
> >
> > [2] Wong, E., Rice, L., \& Kolter, J. Z. (2020). Fast is better than free: Revisiting adversarial training. ArXiv:2001.03994.
> >
> > [3] Bai, T., Luo, J., Zhao, J., Wen, B., \& Wang, Q. (2021). Recent advances in adversarial training for adversarial robustness. ArXiv:2102.01356.
> >
> > **Q6:** The basic notions for convergence should appear early on in the main text (not in an appendix) and be clearly explained to the reader.
> >
> > **A:** We think the reviewer was referring to the notions of last-iterate convergence and the two kinds of minimax optimization? We have added those notions to Theorem 1's remark and Section 2 respectively in the revision.
> >
> > **Q7:** There are grammatical and spelling errors, such as using min-max/minimax alternatively.
> >
> > **A:** Thanks for pointing that out. We have proofread the whole paper and corrected the errors, including replacing all "min-max" (except those in references) into "minimax".

---

> > ### Comment · Reviewer_jvV1 · 2023-02-08
> > **A quick answer**
> >
> > My apologies for the late answer.
> >
> > I am in general happy with the changes however, I certainly believe that the edit in abstract can be improved. It is reasonably awkward writing non-specific words like "some other algorithms". It is more preferable to include the citations.

---

> > > ### Author Response · Authors · 2023-02-09
> > > **Authors' reply to Reviewer jvV1 (3)**
> > >
> > > Hi Reviewer jvV1,
> > >
> > > Thanks a lot for pointing that out. We have edited the abstract.
> > >
> > > Authors

---

### Author Response · Authors · 2023-01-18
**We will upload the rebuttal and revision in 2-3 days**

Dear Reviewers and Editor,

We are almost done with our rebuttal and need a bit more time to present and organize the new experiment results. We will upload the rebuttal and revision in 2-3 days. Thank you for your patience.

Authors

---

### Decision · Action_Editors · 2023-02-21

**Recommendation:** Accept with minor revision

**Comment:**

The authors propose a new method and analysis for  the minimax optimization problem, for smooth non-convex-strongly-concave function. Their goal is to find a local minimax point that satisfies second-order non-degeneracy conditions. To achieve their goal, they
reformulate the objective as a minimization problem of an envelope function, and then develop a second-order cubically regularized Newton method method that leverages the structure of this envelope function.

There is some closely related, and almost concurrent work by Luo, L., Li, Y., and Chen, C, which does take away from some of the novelty of the paper. But the authors do a good job of comparing their work to Luo et. al. and amply cite Luo et. al, and thus I find this acceptable.

The major issue brought up by the reviewers was a lack of clarity and insufficient comparison to the literature. I can see that the authors have done much to address this, and many other questions and recommendations of the reviewers. But authors also posted their substantial revision quite late in the process (5 days after discussion phase), making it difficult for the reviewers to carefully check this new version. Furthermore, all three reviewers still see some issues with clarity, and thus am accepting *conditioned* on improving clarity and to give time for the reviewers to check again the final version.

In particular, reviewer VJjz requested:

1.  What was formally Section I in the appendix (now in the main paper), should be re-written and in it's current state was very hard to read
2. "Paper suffers from insufficient literature comparison and presentation issues. I consider claims related to paper's novelty unclear/unsupported." I see the authors have made a great effort on this, but bare in mind I will ask the reviewers to check the literature comparison again after the next re-submission.

Reviewer YWBv pointed out that

1. Table 2, some typos (additional space) in e.g. $X=N$
2. Table 3, Variance assumption should be $\sigma^2$
3. Def 1, the two parentheses should be combined

Finally, may I also request that the authors adjust the statement in the abstract "Experimental results demonstrate faster convergence of our stochastic Cubic-LocalMinimax than some existing algorithms" to qualify exactly in which experimental setting is Cubic-LocalMinimax faster.


**Audience:**

All three reviewers agreed that part of TMLR's audience would be interested in this work. In particular, quoting Reviewer VJjz:

"The proposed reformulation and the (non-stochastic) approach to tackle it should be interesting for certain people in TMLR's audience. Although, in the current version due to the presentation issues, it is hard to pinpoint what is novel and interesting."

**Claims And Evidence:**

The papers claims do seem to match the content. But there are issues of clarity, that still need to be addressed in a revision. Details below.

---

> ### Author Response · Authors · 2023-02-23
> **Reply to Action Editor**
>
> Dear Action Editor,
>
> We have prepared a final version to address your and the reviewers' comments. Specifically, regarding the literature review in section 1.2, we have added the complexity results of the existing works, and summarized how these works differ from our work. We also simplified the tables to focus on non-convex settings, as required by Reviewer VjJz.
>
> In addition, we have fixed the typos pointed out by Reviewer YWBv, and edited the abstract per editor's suggestion.
>
> Thanks a lot for your suggestions.
>
> Authors